# LoLCATs: On Low-Rank Linearizing of Large Language Models

**Michael Zhang**[*†‡], **Simran Arora**[†‡], **Rahul Chalamala**[‡§], **Alan Wu**[§],
**Benjamin Spector**[†], **Aaryan Singhal**[†], **Krithik Ramesh**[‡§§], **and Christopher Ré**[†]
[†]Stanford University, [‡]Together AI, [§]California Institute of Technology, [§§]MIT
mzhang@cs.stanford.edu

## Abstract

Recent works show we can linearize large language models (LLMs)—swapping the quadratic attentions of popular Transformer-based LLMs with subquadratic analogs, such as linear attention—avoiding the expensive pretraining costs. However, linearizing LLMs often significantly degrades model quality, still requires training over billions of tokens, and remains limited to smaller 1.3B to 7B LLMs. We thus propose Low-rank Linear Conversion via Attention Transfer (LoLCATs), a simple two-step method that improves LLM linearizing quality with orders of magnitudes less memory and compute. We base these steps on two findings. First, we can replace an LLM's softmax attentions with closely-approximating linear attentions, simply by *training* the linear attentions to match their softmax counterparts with an output MSE loss ("*attention transfer*"). Then, this enables adjusting for approximation errors and recovering LLM quality simply with *low-rank* adaptation (LoRA). LoLCATs significantly improves linearizing quality, training efficiency, and scalability. We significantly reduce the linearizing quality gap and produce state-of-the-art subquadratic LLMs from Llama 3 8B and Mistral 7B v0.1, leading to 20+ points of improvement on 5-shot MMLU. Furthermore, LoLCATs does so with only 0.2% of past methods' model parameters and 0.04-0.2% of their training tokens. Finally, we apply LoLCATs to create the first linearized 70B and 405B LLMs ($50\times$ that of prior work). When compared with prior approaches under the same compute budgets, LoLCATs significantly improves linearizing quality, closing the gap between linearized and original Llama 3.1 70B and 405B LLMs by 77.8% and 78.1% on 5-shot MMLU.

## 1 Introduction

"Linearizing" large language models (LLMs)—or converting existing Transformer-based LLMs into attention-free or subquadratic alternatives—has shown promise for scaling up efficient architectures. While many such architectures offer complexity-level efficiency gains, like *linear-time* and *constant-memory* generation, they are often limited to smaller models pretrained on academic budgets (Gu & Dao, 2023; Peng et al., 2023a; Yang et al., 2023; Arora et al., 2024; Beck et al., 2024). In a complementary direction, linearizing aims to start with openly available LLMs—*e.g.,* those with 7B+ parameters pretrained on trillions of tokens (AI, 2024; Jiang et al., 2023)—and (i) swap their softmax attentions with subquadratic analogs, before (ii) further finetuning to recover quality. This holds exciting promise for quickly scaling up subquadratic capabilities.

However, to better realize this promise and allow anyone to convert LLMs into subquadratic models, we desire methods that are (1) **quality-preserving**, *e.g.,* to recover the zero-shot abilities of modern LLMs; (2) **parameter and token efficient**, to linearize LLMs on widely accessible compute; and (3) **highly scalable**, to support linearizing 70B+ LLMs available today (Touvron et al., 2023a;b).

Existing methods present opportunities to improve all three criteria. On quality, despite using motivated subquadratic analogs such as RetNet-inspired linear attentions (Sun et al., 2023; Mercat et al., 2024) or state-space model (SSM)-based Mamba layers (Gu & Dao, 2023; Bick et al., 2024; Wang et al., 2024), prior works significantly reduce performance on popular LM Evaluation Harness tasks (LM Eval) (Gao et al., 2023) (up to 23.4-28.2 pts on 5-shot MMLU (Hendrycks et al., 2020)). On parameter and token efficiency, to adjust for architectural differences, prior methods update *all* model

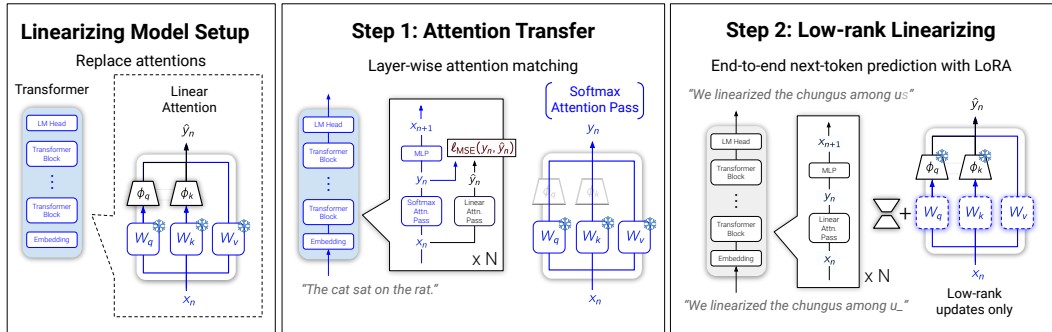

Figure 1: **LoLCATs framework**. We linearize LLMs by (1) training attention analogs to approximate softmax attentions (attention transfer), before swapping attentions and (2) minimally adjusting (with LoRA).

parameters in at least one stage of training (Mercat et al., 2024; Wang et al., 2024; Bick et al., 2024), and use 20 - 100B tokens to linearize 7B LLMs. On scalability, these training costs make linearizing larger models on academic compute more difficult; existing works only linearize up to 8B LLMs. This makes it unclear how to support linearizing 70B to 405B LLMs (Dubey et al., 2024).

In this work, we thus propose **LoLCATs** (**LO**w-rank **L**inear **C**onversion with **A**ttention **T**ran**S**fer), a simple approach to improve the quality, efficiency, and scalability of linearizing LLMs. As guiding motivation, we ask if we can linearize LLMs by simply reducing architectural differences, *i.e.,*

1. Starting with simple softmax attention analogs such as linear attention (Eq. 2), and *training* their parameterizations explicitly to approximate softmax attention ("**attention transfer**").

2. Subsequently only training with low-cost finetuning to adjust for any approximation errors, *e.g.,* with low-rank adaptation (LoRA) (Hu et al., 2021) ("**low-rank linearizing**").

In evaluating this hypothesis, we make several contributions. First, to better understand linearizing feasibility, we empirically study attention transfer and low-rank linearizing with existing linear attentions. While intuitive—by swapping in perfect subquadratic softmax attention approximators, we could get subquadratic LLMs with no additional training—prior works suggest linear attentions struggle to match softmax expressivity (Keles et al., 2023; Qin et al., 2022) or need full-model updates to recover linearizing quality (Kasai et al., 2021; Mercat et al., 2024). In contrast, we find that while *either* attention transfer or LoRA alone is insufficient, we can rapidly recover quality by simply doing *both* (Figure 3, Table 2). At the same time, we do uncover quality issues related to attention-matching architecture and training. With prior linear attentions, the best low-rank linearized LLMs still significantly degrade in quality vs. original Transformers (up to 42.4 pts on 5-shot MMLU). With prior approaches that train all attentions jointly (Zhang et al., 2024), we also find that later layers can result in $200\times$ the MSE of earlier ones (Figure 7). We later find this issue aggravated by larger LLMs; jointly training all Llama 3.1 405B's 126 attention layers fails to viably linearize.

Next, to resolve these issues and improve upon our original criteria, we detail LoLCATs' method components. For **quality**, we generalize prior notions of learnable linear attentions to sliding window + linear attention variants. These remain subquadratic to compute yet consistently yield better attention transfer via lower mean-squared error (MSE) on attention outputs. For **parameter and token efficiency**, we maintain our simple 2-step framework of (1) training subquadratic attentions to match softmax attentions, before (2) adjusting for any errors via only LoRA. For **scalability**, we use finer-grained "block-by-block" training. We split LLMs into blocks of $k$ layers before jointly training attentions only within each block to improve layer-wise attention matching. We pick $k$ to balance the speed of training blocks in parallel with the memory of saving hidden state outputs of prior blocks (as inputs for later ones). We provide a simple cost model to navigate these tradeoffs.

Finally, in experiments, we validate that LoLCATs improves on each of our desired criteria.

- On **quality**, when linearizing popular LLMs such as Mistral-7B and Llama 3 8B, LoLCATs substantially improves past linearizing methods (by $1.1-8.6$ points (pts) on zero-shot LM Eval tasks; +17.2 pts on 5-shot MMLU)). With Llama 3 8B, LoLCATs for the first time closes the zero-shot LM Eval gap between linearized and Transformer models (73.1 vs 74.2 pts), while supporting $3\times$ the throughput and $64\times$ the batch sizes vs. popular FlashAttention-2 (Dao, 2023)

| Name | Architecture | Quality Preserving | Parameter Efficient | Token Efficient | Validated at Scale |
|------|-------------|---------|----------|---------|----------|
| Pretrained | Attention | ✓✓ | ✗✗ | ✗✗ | ✓✓ |
| SUPRA | Linear Attention | ✗ | ✗ | ✓ | ✓ |
| Mohawk | Mamba (2) | ✗ | ✗ | ✓ | ✗ |
| Mamba in Llama | Mamba (2) | ✗ | ✗ | ✓ | ✓ |
| LoLCATs | Softmax-Approx. Linear Attention | ✓ | ✓ | ✓✓ | ✓✓ |

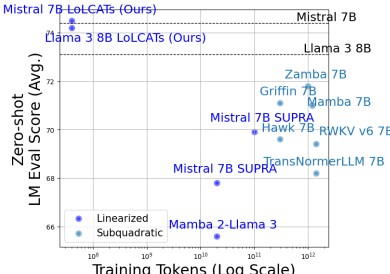

Figure 2: **Linearizing comparison**. LoLCATs significantly improves LLM linearizing quality and training efficiency. No. of ✓ or ✗ indicate relatively better or worse support.

models (generating 4096 tokens on an 80GB H100). As an alternative to pretraining, LoLCATs outperforms strong pretrained 7B subquadratic LLMs and hybrids by 1.2–9.9 pts (LM Eval avg.).

- On **parameter and token-efficiency**, by only training linear attention feature maps in Stage 1, while only using LoRA on linear attention projections in Stage 2, LoLCATs enables these gains while updating only <0.2% of past linearizing methods' model parameters, making a single 40GB GPU sufficient for 7B LLM linearizing. This also only takes 40M tokens, *i.e.,* 0.003% and 0.04% of prior pretraining and linearizing methods' token counts.

- On **scalability**, with LoLCATs we scale up linearizing to support Llama 3.1 70B and 405B LLMs (Dubey et al., 2024). LoLCATs presents the first viable approach to linearizing larger LLMs. We create the first linearized 70B LLM, taking only 18 hours on one 8×80GB H100 node, and the first linearized 405B LLM with a combination of 5 hours on 14 80GB H100 GPUs (attention transfer) + 16 hours on three 8×80GB H100 nodes (LoRA finetuning) for Llama 3.1 405B. For both models, this amount to under half the total GPU hours than prior methods reported to linearize 8B models (5 days on 8×80GB A100s) (Wang et al., 2024). Furthermore, under these computational constraints, LoLCATs significantly improves quality versus prior linearizing approaches without attention transfer. With Llama 3.1 70B and 405B, we close 77.8% and 78.1% of the 5-shot MMLU gap between Transformers and linearized variants respectively.

## 2 PRELIMINARIES

To motivate LoLCATs, we first go over Transformers, attention, and linear attention. We then briefly discuss related works on linearizing Transformers and Transformer-based LLMs.

**Transformers and Attention.** Popular LLMs such as Llama 3 8B (AI@Meta, 2024a) and Mistral 7B (Jiang et al., 2023) are decoder-only Transformers, with repeated blocks of multi-head *softmax attention* followed by MLPs (Vaswani et al., 2017). For one head, attention computes outputs $\boldsymbol{y} \in \mathbb{R}^{l \times d}$ from inputs $\boldsymbol{x} \in \mathbb{R}^{l \times d}$ (where $l$ is sequence length, $d$ is head dimension) with query, key, and value weights $\boldsymbol{W}_q, \boldsymbol{W}_k, \boldsymbol{W}_v \in \mathbb{R}^{d \times d}$. In causal language modeling, we compute $\boldsymbol{q} = \boldsymbol{x}\boldsymbol{W}_q$, $\boldsymbol{k} = \boldsymbol{x}\boldsymbol{W}_k$, $\boldsymbol{v} = \boldsymbol{x}\boldsymbol{W}_v$, before getting attention weights $\boldsymbol{a}$ and outputs $\boldsymbol{y}$ via

$$a_{n,i} = \frac{\exp(\boldsymbol{q}_n^\top \boldsymbol{k}_i / \sqrt{d})}{\sum_{i=1}^n \exp(\boldsymbol{q}_n^\top \boldsymbol{k}_i / \sqrt{d})}, \quad \boldsymbol{y}_n = \sum_{i=1}^n a_{n,i} \boldsymbol{v}_i, \quad \text{for } n \text{ in } [1, \dots, l] \quad (1)$$

Multi-head attention maintains inputs, outputs, and weights for each head, *e.g.,* $\boldsymbol{x} \in \mathbb{R}^{h \times l \times d}$ or $\boldsymbol{W}_q \in \mathbb{R}^{h \times d \times d}$ ($h$ being number of heads), and computes Eq. 1 for each head. In both cases, we compute final outputs by concatenating $\boldsymbol{y}_n$ across heads, before using output weights $\boldsymbol{W}_o \in \mathbb{R}^{hd \times hd}$ to compute $\boldsymbol{y}_n \boldsymbol{W}_o \in \mathbb{R}^{l \times hd}$. While expressive, causal softmax attention requires all $\{\boldsymbol{k}_i, \boldsymbol{v}_i\}_{i \leq n}$ to compute $\boldsymbol{y}_n$. For long context or large batch settings, this growing *KV cache* can incur prohibitive memory costs even with state-of-the-art implementations such as FlashAttention (Dao, 2023).

**Linear Attention.** To get around this, Katharopoulos et al. (2020) show a similar attention operation, but with *linear* time and *constant* memory over generation length (linear time and space when processing inputs). To see how, note that softmax attention's exponential is a kernel function $\mathcal{K}(\boldsymbol{q}_n, \boldsymbol{k}_i)$, which in general can be expressed as the dot product of feature maps $\phi : \mathbb{R}^d \mapsto \mathbb{R}^{d'}$. Swapping $\exp(\boldsymbol{q}_n^\top \boldsymbol{k}_i / \sqrt{d})$ with $\phi(\boldsymbol{q}_n)^\top \phi(\boldsymbol{k}_i)$ in Eq. 1 gives us *linear attention* weights and outputs:

$$\hat{\boldsymbol{y}}_n = \sum_{i=1}^n \frac{\phi(\boldsymbol{q}_n)^\top \phi(\boldsymbol{k}_i) \boldsymbol{v}_i}{\sum_{i=1}^n \phi(\boldsymbol{q}_n)^\top \phi(\boldsymbol{k}_i)} = \frac{\phi(\boldsymbol{q}_n)^\top \left( \sum_{i=1}^n \phi(\boldsymbol{k}_i) \boldsymbol{v}_i^\top \right)}{\phi(\boldsymbol{q}_n)^\top \sum_{i=1}^n \phi(\boldsymbol{k}_i)} \quad (2)$$

This lets us compute both the numerator $s_n = \sum_{i=1}^n \phi(k_i) v_i^\top$ and denominator $z_n = \sum_{i=1}^n \phi(k_i)$ as recurrent "KV states". With $s_0 = 0, z_0 = 0$, we recurrently compute linear attention outputs as

$$\hat{y}_n = \frac{\phi(q_n)^\top s_n}{\phi(q_n)^\top z_n} \text{ for } s_n = s_{n-1} + \phi(k_n) v_n^\top \text{ and } z_n = z_{n-1} + \phi(k_n) \tag{3}$$

Eq. 2 lets us compute attention over an input sequence of length $n$ in $\mathcal{O}(ndd')$ time and space, while Eq. 3 lets us compute $n$ new tokens in $\mathcal{O}(ndd')$ time and $\mathcal{O}(dd')$ memory. Especially during generation, when softmax attention has to compute new tokens sequentially anyway, Eq. 3 enables time and memory savings if $d' <$ (prompt length + prior generated tokens).

**Linearizing Transformers.** To combine efficiency with quality, various works propose different $\phi$, (*e.g.,* $\phi(x) = 1 + \text{ELU}(x)$ as in Katharopoulos et al. (2020)). However, they typically train linear attention Transformers from scratch. We build upon recent works that *swap* the softmax attentions of *existing* Transformers with linear attention before finetuning the modified models with next-token prediction to recover language modeling quality. These include methods proposed for LLMs (Mercat et al., 2024), and those for smaller Transformers—*e.g.,* 110M BERTs (Devlin et al., 2018))—reasonably adaptable to modern LLMs (Kasai et al., 2021; Mao, 2022; Zhang et al., 2024).

## 3 METHOD: LINEARIZING LLMS WITH LoLCATS

We now study how to build a high-quality and highly efficient linearizing method. In Section 3.1, we present our motivating framework, which aims to (1) learn good softmax attention approximators with linear attentions and (2) enable low-rank adaptation for recovering linearized quality. In Section 3.2, we find that while this attention transfer works surprisingly well for low-rank linearizing with existing linear attentions, on certain tasks, it still results in sizable quality gaps compared to prior methods. We also find that attention-transfer quality strongly corresponds with the final linearized model's performance. In Section 3.3, we use our learned findings to overcome prior issues, improving attention transfer to subsequently improve low-rank linearizing quality.

### 3.1 LoLCATS PART 1: A FRAMEWORK FOR LOW-COST LINEARIZING

In this section, we present our initial LoLCATs framework for linearizing LLMs in an effective yet efficient manner. Our main hypothesis is that by first learning linear attentions that approximate softmax, we can then swap these attentions in as drop-in subquadratic replacements. We would then only need a minimal amount of subsequent training—*e.g.,* that is supported by low-rank updates—to recover LLM quality in a cost-effective manner effectively. We thus proceed in two steps.

1. **Parameter-Efficient Attention Transfer.** For each softmax attention in an LLM, we aim to learn a closely-approximating linear attention, *i.e.,* one that computes attention outputs $\hat{y} \approx y$ for all natural inputs $x$. We call this "attention transfer", as we aim to *transfer* the attention modeling of existing softmax attentions into target linear attentions. Due to architectural similarity, we can treat this as a feature map learning problem, learning $\phi$ to approximate softmax. For each head and layer, let $\phi_q$ and $\phi_k$ be query and key feature maps. Per head, we compute:

$$y_n = \underbrace{\sum_{i=1}^n \frac{\exp(q_n^\top k_i / \sqrt{d})}{\sum_{i=1}^n \exp(q_n^\top k_i / \sqrt{d})} v_i}_{\text{Softmax Attention}}, \quad \hat{y}_n = \underbrace{\sum_{i=1}^n \frac{\phi_q(q_n)^\top \phi_k(k_i)}{\sum_{i=1}^n \phi_q(q_n)^\top \phi_k(k_i)} v_i}_{\text{Linear Attention}} \tag{4}$$

for all $n \in [l]$ with input $\in \mathbb{R}^{l \times d}$, and train $\phi_q, \phi_k$ to minimize sample mean squared error (MSE)

$$\ell_{\text{MSE}} = \frac{1}{MH} \sum_{m=1}^M \sum_{h=1}^H \ell_{\text{MSE}}^{h,m}, \quad \ell_{\text{MSE}}^{h,m} = \frac{1}{d} \sum_{n=1}^d (y_n - \hat{y}_n)^2 \tag{5}$$

*i.e.,* jointly for each head $h$ in layer $m$. Similar to past work (Kasai et al., 2021; Zhang et al., 2024), rather than manually design $\phi$, we parameterize each $\phi: \mathbb{R}^d \mapsto \mathbb{R}^{d'}$ as a *learnable* layer:

$$\phi_q(q_n) := f(q_n \tilde{W}_{(q)} + \tilde{b}_{(q)}), \quad \phi_k(k_i) := f(k_i \tilde{W}_{(k)} + \tilde{b}_{(k)})$$

Here $\tilde{W} \in \mathbb{R}^{d \times d'}$ and $\tilde{b} \in \mathbb{R}^{d'}$ are trainable weights and optional biases, $f(\cdot)$ is a nonlinear activation, and $d'$ is an arbitrary feature dimension (set to equal head dimension $d$ in practice).

2. **Low-rank Adjusting.** After training the linearizing layers, we replace the full-parameter training of prior work with low-rank adaptation (LoRA) (Hu et al., 2021). Like prior work, to adjust for the modifying layers and recover language modeling quality, we now train the modified LLM end-to-end over tokens to minimize a sample next-token prediction loss $\ell_{\text{xent}} = -\sum \log P_\Theta(\boldsymbol{u}_{t+1} \mid \boldsymbol{u}_{1:t})$. Here $P_\Theta$ is the modified LLM, $\Theta$ is the set of LLM parameters, and we aim to maximize the probability of true $\boldsymbol{u}_{t+1}$ given past tokens $\boldsymbol{u}_{1:t}$ (Fig. 1 right). However, rather than train all LLM parameters, we only train the swapped linear attention $\boldsymbol{W}_q, \boldsymbol{W}_k, \boldsymbol{W}_v, \boldsymbol{W}_o$ with LoRA, updating $\boldsymbol{W}' \leftarrow \boldsymbol{W} + \Delta\boldsymbol{W}$ with $\Delta\boldsymbol{W}$ as the product of two low-rank matrices $\boldsymbol{BA}$, $\boldsymbol{B} \in \mathbb{R}^{d \times r}$, $\boldsymbol{A} \in \mathbb{R}^{r \times d}$. This enables parameter efficiency for $r \ll d$.

**Training footprint and efficiency**. Both steps remain parameter-efficient. For Step 1, optimizing Eq. 5 is similar to a layer-by-layer cross-architecture distillation. We compute layer-wise $(\boldsymbol{x}, \boldsymbol{y})$ as pretrained attention inputs and outputs, using an LLM forward pass over natural language samples (Fig. 1 middle). However, to keep our training footprint low, we freeze the original pretrained attention layer's parameters and simply *insert* new $\phi_q, \phi_k$ after $\boldsymbol{W}_q, \boldsymbol{W}_k$ in each softmax attention (Fig. 1 left). We compute outputs $\boldsymbol{y}, \hat{\boldsymbol{y}}$ with the same attention weights in separate passes (choosing either Eq. 1 or Eq. 2; "teacher-forcing" by only sending softmax outputs $\boldsymbol{y}$ to future layers and preventing error propagation, Fig. 1 middle). For Llama 3 8B or Mistral 7B, training $\phi_q, \phi_k$ with $d' = 64$ then only takes 32 layers $\times$ 32 heads $\times$ 2 feature maps $\times$ $(128 \times 64)$ weights $\approx$ 16.8M trainable weights (0.2% of LLM sizes). Furthermore, Eq. 5 keeps attention transfer memory-efficient. While Zhang et al. (2024) also train layer-wise $\phi$ to approximate softmax attention, they supervise by matching on the $n^2$ *attention weights* computed for $n$-token inputs (App. D, Eq. 12). This scales poorly for large $n$. Instead, by only needing attention outputs, we can reduce memory from $\mathcal{O}(n^2)$ to $\mathcal{O}(n)$, computing softmax attention $\boldsymbol{y}$ with FlashAttention (Dao et al., 2022; Dao, 2023) and linear attention $\hat{\boldsymbol{y}}$ with Eq. 2 (or 6) both in $\mathcal{O}(n)$ memory by not materializing attention weights. Despite these savings, we show similar attention weight recovery in App. E.2. In Step 2, LoRA with $r = 8$ on all attention projections suffices for state-of-the-art quality. This updates just <0.09% of 7B parameters.

## 3.2 BASELINE STUDY: ATTENTION TRANSFER AND LOW-RANK LINEARIZING

As a first step, we aim to understand if attention transfer and low-rank adjusting as proposed are sufficient for linearizing LLMs. While simple, it is unclear whether these steps can lead to high-quality linearizing, as all prior works default to more involved approaches (Mercat et al., 2024; Bick et al., 2024; Wang et al., 2024). They use linearizing layers featuring GroupNorms (Wu & He, 2018) and decay factors (Sun et al., 2023), or alternate SSM-based architectures (Gu & Dao, 2023; Dao & Gu, 2024). They also all use full-LLM training after swapping in the subquadratic layers. In contrast, as a first contribution we find that simple linear attentions *can* lead to viable linearizing, with attention transfer + LoRA obtaining competitive quality on 4 / 6 popular LM Eval tasks.

**Experimental Setup.** We test the LoLCATs framework by linearizing two popular base LLMs, Llama 3 8B (AI, 2024) and Mistral 7B v0.1 (Jiang et al., 2023). For linearizing layers, we study two feature maps used in prior work (Table 1). To support the rotary positional embeddings (RoPE) (Su et al., 2024) in these LLMs, we apply the feature maps $\phi$ after

| Feature Map | $\phi(\boldsymbol{q})$ (same for $\boldsymbol{k}$) | Weight Shapes |
|---|---|---|
| T2R | $\text{ReLU}(\boldsymbol{q}\bar{\boldsymbol{W}} + \bar{\boldsymbol{b}})$ | $\bar{\boldsymbol{W}}$: (128, 128), $\bar{\boldsymbol{b}}$: (128,) |
| Hedgehog | $[\text{SM}_d(\boldsymbol{q}\tilde{\boldsymbol{W}}) \oplus \text{SM}_d(-\boldsymbol{q}\tilde{\boldsymbol{W}})]$ | $\tilde{\boldsymbol{W}}$ : (128, 64) |

Table 1: **Learnable feature maps**. Transformer to RNN (T2R) from Kasai et al. (2021), Hedgehog from Zhang et al. (2024), both $\oplus$ (concat) and $\text{SM}_d$ (softmax) apply over feature dimension.

RoPE,[1] *i.e.,* computing query features $\phi_q(\boldsymbol{q}) = f(\text{RoPE}(\boldsymbol{q})\tilde{\boldsymbol{W}}_q + \tilde{\boldsymbol{b}})$. For linearizing data, we wish to see if LoLCATs with a small amount of data can recover general zero-shot and instruction-following LLM abilities. We use the 50K samples of a cleaned Alpaca dataset[2], due to its ability to improve general instruction-following in 7B LLMs despite its relatively small size (Taori et al., 2023). We train all feature maps jointly. Training code and implementation details are in App. C.

To study the effects of attention transfer and low-rank linearizing across LLMs and linear attention architectures, we evaluate their validation set perplexity (Table 2, Fig. 3) and downstream LM Eval

---

[1] Unlike prior works that apply $\phi$ before RoPE (Mercat et al., 2024; Su et al., 2024), our choice preserves the linear attention kernel connection, where we can hope to learn $\phi_q, \phi_k$ for $\exp(\boldsymbol{q}_n^\top \boldsymbol{k}_i / \sqrt{d}) \approx \phi_q(\boldsymbol{q}_n)^\top \phi_k(\boldsymbol{k}_i)$.

[2] https://huggingface.co/datasets/yahma/alpaca-cleaned

| Llama 3 8B, T2R | Llama 3 8B, Hedgehog | Mistral 7B, T2R | Mistral 7B, Hedgehog |
|---|---|---|---|

Figure 3: **Attention transfer training efficiency**. Even accounting for initial training steps, low-rank linearizing with attention transfer still consistently achieves lower perplexity faster across feature maps and LLMs.

zero-shot quality (Table 4). We use the same data for both stages, early stopping, and either 2 epochs for attention transfer and LoRA adjusting or 4 epochs for either alone ($\approx$ 40M total training tokens). We use LoRA $r = 8$ by popular default (Hu et al., 2021), training 0.2% of LLM parameter counts.

| Attention | Llama 3 8B | | | | Mistral 7B | | | |
|---|---|---|---|---|---|---|---|---|
| | T2R | | Hedgehog | | T2R | | Hedgehog | |
| Transfer? | PPL@0 | PPL@2/4 | PPL@0 | PPL@2/4 | PPL@0 | PPL@2/4 | PPL@0 | PPL@2/4 |
| No ✗ | 1539.39 | 16.05 | 2448.01 | 9.02 | 2497.13 | 8.85 | 561.47 | 4.87 |
| Yes ✓ | **79.33** | **4.11** | **60.86** | **3.90** | **32.78** | **3.29** | **18.94** | **3.04** |

Table 2: Alpaca validation set perplexity (PPL) of linearized LLMs, comparing attention transfer, no LoRA adjusting (PPL@0) and PPL after training (PPL@2/4; 2 with attention transfer, 4 without, for equal total steps).

**Attention Transfer + LoRA Enables Fast LLM Linearizing.** In Table 2 and Fig. 3, we report the validation PPL of linearized LLMs, ablating attention transfer and LoRA adjusting. We find that while attention transfer alone is often insufficient (*c.f.,* PPL@0, Table 2), a single low-rank update rapidly recovers performance by 15–75 PPL (Fig. 3), where training to approximate softmax leads to up to 11.9 lower PPL than no attention transfer. Somewhat surprisingly, this translates to performing competitively with prior linearizing methods that train *all* model parameters (Mercat et al., 2024; Wang et al., 2024) (within 5 accuracy points on 4 / 6 popular LM Eval tasks; Table 4), while *only* training with 0.04% of their token counts and 0.2% of their parameter counts. The results suggest we can linearize 7B LLMs at orders-of-magnitude less training costs than previously shown.

**LoL SAD: Limitations of Low-Rank Linearizing.** At the same time, we note quality limitations with the present framework. While sometimes close, low-rank linearized LLMs perform worse than full-parameter alternatives and original Transformers on 5 / 6 LM Eval tasks (up to 42.4 points on 5-shot MMLU; Table 4). To understand the issue, we study if the attention transfer stage can produce close linear attention approximations of LLM softmax attentions. We note three observations:

1. Attention transfer quality (via output MSE) strongly ties to final linearized LLM quality (via PPL) (Fig. 5), suggesting we can improve quality by reducing MSE with softmax attentions.

2. However, larger MSEs coincide with lower softmax attention weight entropies (Fig. 6a). Zhang et al. (2024) find linear attentions struggle to approximate such "spikier" distributions, suggesting we may need better attention-matching layers to reduce MSE and improve final linearized quality.

3. When training layers jointly like in prior work (Zhang et al., 2024), larger MSEs also heavily concentrate in later layers (Fig. 6b). To bring the MSE in these layers down, we may thus need more fine-grained layer-wise supervision, rather than the objective over all layers in Eq. 5.

| Model | Tokens (B) | PiQA | ARC-E | ARC-C | HS | WG | MMLU |
|---|---|---|---|---|---|---|---|
| Llama 3 8B | - | 79.9 | 80.1 | 53.3 | 79.1 | 73.1 | 66.6 |
| → Mamba2 | 100 | 76.8 | **74.1** | **48.0** | **70.8** | **58.6** | **43.2** |
| → LoRA Hedgehog | 0.04 | **77.4** | 71.1 | 40.6 | 66.5 | 54.3 | 24.2 |
| Mistral 7B | - | 82.1 | 80.9 | 53.8 | 81.0 | 74.0 | 62.4 |
| → SUPRA | 100 | **80.4** | 75.9 | **45.8** | **77.1** | **70.3** | **34.2** |
| → LoRA Hedgehog | 0.04 | 79.3 | **76.4** | 45.1 | 73.1 | 57.5 | 28.2 |

Figure 4: **Linearizing comparison on LM Eval**. Task names in Table 4. Acc. norm: ARC-C, HS. Acc. otherwise. 5-shot MMLU. 0-shot otherwise.

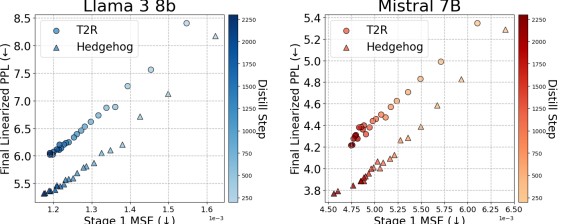

Figure 5: **Attention MSE vs. PPL**. Across feature maps, LLMs; lower MSE coincides with better linearized quality.

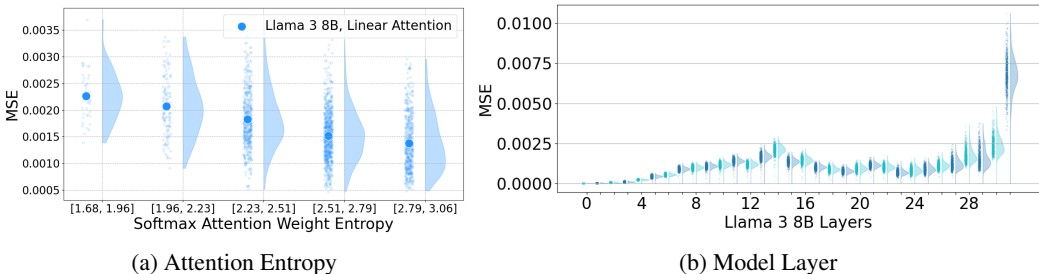

(a) Attention Entropy
(b) Model Layer

Figure 6: **Sources of Attention Transfer Error** with Llama 3 8B. We find two potential sources of attention transfer difficulty: (a) low softmax attention entropy and (b) attentions in later layers.

### 3.3 LoLCATs PART 2: COMPONENTS TO IMPROVE LOW-RANK LINEARIZING

Following our motivating hypotheses, framework, and observations, we now introduce two simple improvements to improve linearized LLM quality by reducing MSE: (1) better attention-matching architectures (Section 3.3.1), and (2) finer-grained layer-wise attention transfer (Section 3.3.2).

#### 3.3.1 ARCHITECTURE: GENERALIZING LEARNABLE LINEAR ATTENTIONS

As described, we can apply our framework with any linear attentions with learnable $\phi$ (*e.g.*, T2R and Hedgehog, Figure 3). However, to improve attention-matching quality, we introduce a hybrid $\phi$ parameterization combining linear attention and *sliding window* attention. Motivated by prior works that show quality improvements when combining attention layers with linear attentions (Arora et al., 2024; Munkhdalai et al., 2024), we combine short sliding windows of softmax attention (Beltagy et al., 2020; Zhu et al., 2021) (size 64 in experiments) followed by linear attention in a single layer. This allows attending to all prior tokens for each layer while keeping the entire LLM subquadratic. For window size $w$ and token indices $[1, \ldots, n-w, \ldots, n]$, we apply the softmax attention over the $w$ most recent tokens, and compute attention outputs $\hat{\boldsymbol{y}}_n$ as

$$\hat{\boldsymbol{y}}_n = \frac{\sum_{i=n-w+1}^{n} \gamma \exp(\boldsymbol{q}_n^\top \boldsymbol{k}_i/\sqrt{d} - \boldsymbol{c}_n)\boldsymbol{v}_i + \phi_q(\boldsymbol{q}_n)^\top \left( \sum_{j=1}^{n-w} \phi_k(\boldsymbol{k}_j)\boldsymbol{v}_j^\top \right)}{\sum_{i=n-w+1}^{n} \gamma \exp(\boldsymbol{q}_n^\top \boldsymbol{k}_i/\sqrt{d} - \boldsymbol{c}_n) + \phi_q(\boldsymbol{q}_n)^\top \left( \sum_{j=1}^{n-w} \phi_k(\boldsymbol{k}_j)^\top \right)} \quad (6)$$

$\gamma$ is a learnable mixing term, and $\boldsymbol{c}_n$ is a stabilizing constant as in log-sum-exp calculations ($\boldsymbol{c}_n = \max_i \left\{ \boldsymbol{q}_n^\top \boldsymbol{k}_i/\sqrt{d} : i \in [n-w+1, \ldots, n] \right\}$). Like before, we can pick any learnable $\phi$.

**Subquadratic efficiency.** The hybrid layer retains linear time and constant memory generation. For $n$-token prompts, we initially require $\mathcal{O}(w^2 d)$ and $\mathcal{O}((n-w)dd')$ time and space for window and linear attention respectively, attending over a $w$-sized KV-cache and computing KV and K-states (Eq. 3). For generation, we only need $\mathcal{O}(w^2 d + dd')$ time and space for every token. We evict the KV-cache's first $\boldsymbol{k}, \boldsymbol{v}$, compute $\phi_k(\boldsymbol{k})$, and add $\phi_k(\boldsymbol{k})\boldsymbol{v}^\top$ and $\phi_k(\boldsymbol{k})$ to KV and K-states respectively.

**Hardware-aware implementation.** To make Eq. 6 competitive with modern softmax attentions like FlashAttention-2 (Dao, 2023), we provide a "hardware-aware" Eq. 6 with the Hedgehog feature map. For space, we defer implementation details to App. A, C.2. We evaluate this version by default.

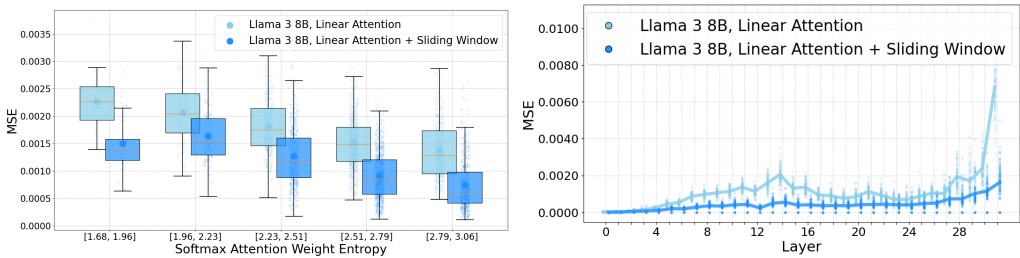

Figure 7: **Improving Attention matching MSE**. Linearizing with linear + sliding window attention better matches LLM softmax attentions (lower MSE) over attention entropy values and LLM layers.

### 3.3.2 TRAINING: LAYER (OR BLOCK)-WISE ATTENTION TRANSFER

We describe our training approach and provide a simplified model to show its cost-quality tradeoffs. Based on the layer-wise MSE differences in Sec. 3.2 from training all layers jointly (Eq. 5), we instead generalize to training over finer-grained $b$-layer blocks, and train each block independently:

$$\ell_{\text{MSE}}^{\text{block}} = \frac{1}{bH} \sum_{m=i}^{i+b} \sum_{h=1}^{H} \ell_{\text{MSE}}^{h,m} \quad \text{(for blocks starting at layers } i = 0, b, 2b, \ldots) \tag{7}$$

We choose $b$ to balance quality and cost in both memory and training time, which we find particularly helpful for linearizing larger LLMs. Several block-wise training approaches exist, including joint training with separate optimizer groups per block, sequentially training separate blocks, or precomputing hidden states and training blocks in parallel across GPUs. For space, we report results on LLM quality trade-offs in App. B.7.3, where we find smaller $b$ improves LoLCATs Llama 3.1 405B PPL by 1.02 points (Table 24). Below we discuss primary cost trade-offs:

- **Compute**: While the joint training of Llama 3.1 405B in 16-bit precision uses *multiple nodes* (e.g., NVIDIA H100 $8 \times 80$GB nodes), an individual block of $b = 9$ or fewer layers can be trained on a *single GPU* (e.g., H100 80GB GPU) at sequence length 1024.
- **Time**: However, doing so on a single GPU multiples total training time by $M/b$. With multiple GPUs, we can distribute training different blocks on difference devices in parallel. To train blocks at layers $b, 2b, \ldots$, we need the outputs from each prior block. As we teacher-force with the "true" softmax attention outputs, we can simply precompute these with the original Transformer.
- **Memory**: Lastly, we need to save each precomputed block's outputs to disk. The total disk space required is $2 \times T \times d \times \frac{L}{k}$ for total training tokens $T$, model dimension $d$, number of layers $L$ and 2-byte (16-bit) precision. For Llama 3.1 405B, saving states per-layer ($b = 1$) for just 50M tokens requires over 200TB of disk space. Larger $b$ divides this storage, potentially at the cost of quality.

For each target LLM, we thus aim to make low-rank linearizing feasible by first doing block-wise attention transfer, adjusting parameters based on linearizing quality and what is feasible in compute, time, and memory. We summarize LoLCATs with Alg. 1, 2, providing pseudocode in App. C.1.

---

**Algorithm 1** LoLCATs Step 1: Attn. Transfer

**Input:** Pretrained Transformer with $M$ attn. layers; input tokens $\boldsymbol{u}$
**Input:** Linear attn. feature map params. $\{\phi_q^m, \phi_k^m, \gamma^m : m \in [M]\}$, window size $w$; layer block-size $b$, learning rate $\alpha$
1: Freeze all Transformer parameters
2: **Initialize** block-wise losses $\{\ell_i \leftarrow 0 : i \in [M/\!/b]\}$
3: Compute initial attn. input $\boldsymbol{x}^1 \leftarrow \text{embed}(\boldsymbol{u})$
4: **for** attn. layer $m \in [M]$ **do** $\quad \triangleright$ **Compute attentions**
5: $\quad \boldsymbol{q}, \boldsymbol{k}, \boldsymbol{v} = \boldsymbol{x}^m \boldsymbol{W}_q^m, \boldsymbol{x}^m \boldsymbol{W}_k^m, \boldsymbol{x}^m \boldsymbol{W}_v^m$
6: $\quad \boldsymbol{y}^m = \text{softmax\_attn}(\boldsymbol{q}, \boldsymbol{k}, \boldsymbol{v})$ (Eq. 1) $\quad \triangleright$ (No grad)
7: $\quad \hat{\boldsymbol{y}}^m = \text{linear\_attn}(\boldsymbol{q}, \boldsymbol{k}, \boldsymbol{v}, w, \phi_q^m, \phi_k^m, \gamma^m)$ (Eq. 6)
8: $\quad \ell_{m/\!/b} \leftarrow \ell_{m/\!/b} + \ell_{\text{MSE}}(\hat{\boldsymbol{y}}^m, \boldsymbol{y}^m)$ $\quad$ (Eq. 7)
9: $\quad \boldsymbol{x}^{m+1} = \text{mlp}^m(\boldsymbol{y}^m \boldsymbol{W}_o^m)$ $\quad \triangleright$ Teacher-force next layer
10: **for** attn. layer $m \in [M]$ **do** $\quad \triangleright$ **Update feature maps**
11: $\quad$ **for** weights $\theta \in \{\phi_q^m, \phi_k^m, \gamma^m\}$ **do**
12: $\quad\quad$ Update $\theta \leftarrow \theta - \alpha \frac{\partial}{\partial \theta} \ell_{m/\!/b}$

**Algorithm 2** LoLCATs Step 2: LoRA Adjust

**Input:** *Linearized* Transformer with $M$ attn. layers; input tokens $\boldsymbol{u}$
**Input:** *Trained* feature map params. $\{\phi_q^m, \phi_k^m, \gamma^m : m \in [M]\}$, window size $w$; learning rate $\alpha$
1: **for** attn. layer $m \in [M]$ **do** $\quad \triangleright$ (Add LoRA weights)
$\quad$ **Init.** LoRA $\boldsymbol{A}^m, \boldsymbol{B}^m$ for each $\boldsymbol{W}_q^m, \boldsymbol{W}_k^m, \boldsymbol{W}_v^m, \boldsymbol{W}_o^m$
2: Compute initial attn. input $\boldsymbol{x}^1 \leftarrow \text{embed}(\boldsymbol{u})$
3: **for** attn. layer $m \in [M]$ **do** $\quad \triangleright$ **Compute attentions**
4: $\quad \boldsymbol{q} = \boldsymbol{x}^m (\boldsymbol{W}_q^m + \boldsymbol{B}_q^m \boldsymbol{A}_q^m)$ $\quad \triangleright$ (LoRA forward pass)
5: $\quad \boldsymbol{k} = \boldsymbol{x}^m (\boldsymbol{W}_k^m + \boldsymbol{B}_k^m \boldsymbol{A}_k^m)$
6: $\quad \boldsymbol{v} = \boldsymbol{x}^m (\boldsymbol{W}_v^m + \boldsymbol{B}_v^m \boldsymbol{A}_v^m)$
7: $\quad \hat{\boldsymbol{y}}^m = \text{linear\_attn}(\boldsymbol{q}, \boldsymbol{k}, \boldsymbol{v}, w, \phi_q^m, \phi_k^m, \gamma^m)$ (Eq. 6)
8: $\quad \boldsymbol{x}^{m+1} = \text{mlp}^m(\boldsymbol{y}^m (\boldsymbol{W}_o^m + \boldsymbol{B}_o^m \boldsymbol{A}_o^m))$
9: Compute next-token pred $\hat{\boldsymbol{u}} = \text{lm\_head}(\boldsymbol{x}^M)$ $\quad \triangleright$ **Train LoRA**
10: Compute sample loss $\ell = \ell_{\text{CrossEnt}}(\hat{\boldsymbol{u}}_{0:n-1}, \boldsymbol{u}_{1:n})$
11: Update $\theta \leftarrow \theta - \alpha \frac{\partial \ell}{\partial \theta}$ for all $\theta$ in all $\boldsymbol{A}, \boldsymbol{B}$

---

## 4 EXPERIMENTS

Through experiments, we study: (1) if LoLCATs linearizes LLMs with higher quality than existing subquadratic alternatives and linearizations, and higher generation efficiency than original Transformers (Sec. 4.1); (2) how ablations on attention transfer loss, subquadratic architecture, and parameter and token counts impact LLM quality (Sec. 4.2); and (3) how LoLCATs' quality and efficiency holds up to 70B and 405B LLMs by linearizing the complete Llama 3.1 family (Sec. 4.3).

### 4.1 MAIN RESULTS: LoLCATS EFFICIENTLY RECOVERS QUALITY IN LINEARIZED LLMS

In our main evaluation, we linearize the popular base Llama 3 8B (AI, 2024) and Mistral 7B (Jiang et al., 2023) LLMs. We first test if LoLCATs can efficiently create high-quality subquadratic LLMs from strong base Transformers, comparing to existing linearized LLMs from prior methods. We also

| Model | Training Tokens (B) | PiQA | ARC-e | ARC-c (norm) | HellaSwag (norm) | Wino-grande | MMLU (5-shot) | Avg. | Avg. (no MMLU) |
|---|---|---|---|---|---|---|---|---|---|
| Mistral 7B | - | 82.1 | 80.9 | 53.8 | 81.0 | 74.0 | 62.4 | 72.4 | 74.4 |
| Mistral 7B SUPRA | 100 | 80.4 | 75.9 | 45.8 | 77.1 | 70.3 | 34.2 | 64.0 | 69.9 |
| Mistral 7B LoLCATs (Ours) | **0.04** | **81.5** | **81.7** | **54.9** | **80.7** | **74.0** | **51.4** | **70.7** | **74.5** |
| Llama 3 8B | - | 79.9 | 80.1 | 53.3 | 79.1 | 73.1 | 66.6 | 72.0 | 73.1 |
| Mamba2-Llama 3 | 20 | 76.8 | 74.1 | 48.0 | 70.8 | 58.6 | 43.2 | 61.9 | 65.6 |
| Mamba2-Llama 3, 50% Attn. | 20 | **81.5** | 78.8 | **58.2** | 79.5 | 71.5 | **56.7** | **71.0** | 73.9 |
| Llama 3 8B Hedgehog | **0.04** | 77.4 | 71.1 | 40.6 | 66.5 | 54.3 | 24.2 | 55.7 | 62.0 |
| Llama 3 8B LoLCATs (Ours) | **0.04** | 80.9 | **81.7** | 54.9 | **79.7** | **74.1** | 52.8 | 70.7 | **74.2** |

Table 3: **LoLCATs comparison among linearized 7B+ LLMs**. Among linearized 7B+ LLMs, LoLCATs-linearized Mistral 7B and Llama 3 8B consistently achieve best or 2nd-best performance on LM Eval tasks. LoLCATs closes the Transformer quality gap by 79.8% (Mistral 7B) and 86.6% (Llama 3 8B) (average over all tasks; numbers except Hedgehog cited from original works), despite only using 40M tokens to linearize.

| Model | Tokens (B) | PiQA | ARC-e | ARC-c (acc. norm) | HellaSwag (acc. norm) | Winogrande | MMLU (5-shot) | Avg. (w MMLU) | Avg. (no MMLU) |
|---|---|---|---|---|---|---|---|---|---|
| **Transformer** | | | | | | | | | |
| Gemma 7B | 6000 | 81.9 | 81.1 | 53.2 | 80.7 | 73.7 | 62.9 | 72.3 | 74.1 |
| Mistral 7B | 8000* | 82.1 | 80.9 | 53.8 | 81.0 | 74.0 | 62.4 | 72.4 | 74.4 |
| Llama 3 8B | 15000 | 79.9 | 80.1 | 53.3 | 79.1 | 73.1 | 66.6 | 72.0 | 73.1 |
| **Subquadratic** | | | | | | | | | |
| Mamba 7B | 1200 | 81.0 | 77.5 | 46.7 | 77.9 | 71.8 | 33.3 | 64.7 | 71.0 |
| RWKV-6 World v2.1 7B | 1420 | 78.7 | 76.8 | 46.3 | 75.1 | 70.0 | - | 69.4 | 69.4 |
| TransNormerLLM 7B | 1400 | 80.1 | 75.4 | 44.4 | 75.2 | 66.1 | 43.1 | 64.1 | 68.2 |
| Hawk 7B | 300 | 80.0 | 74.4 | 45.9 | 77.6 | 69.9 | 35.0 | 63.8 | 69.6 |
| Griffin 7B | 300 | 81.0 | 75.4 | 47.9 | 78.6 | 72.6 | 39.3 | 65.8 | 71.1 |
| **Hybrid Softmax** | | | | | | | | | |
| StripedHyena-Nous-7B | - | 78.8 | 77.2 | 40.0 | 76.4 | 66.4 | 26.0 | 60.8 | 67.8 |
| Zamba 7B | 1000 | 81.4 | 74.5 | 46.6 | 80.2 | **76.4** | 57.7 | 69.5 | 71.8 |
| **Linearized** | | | | | | | | | |
| Mistral 7B LoLCATs (Ours) | **0.04** | **81.5** | **81.7** | **54.9** | **80.7** | 74.0 | 51.4 | **70.7** | **74.5** |
| Llama 3 8B LoLCATs (Ours) | **0.04** | 80.9 | **81.7** | **54.9** | 79.7 | 74.1 | 52.8 | **70.7** | 74.2 |

Table 4: **LoLCATs comparison to pretrained subquadratic LLMs**. LoLCATs-linearized Mistral 7B and Llama 3 8B outperform pretrained Transformer alternatives by 1.2 to 9.9 points (Avg.), only training 0.2% of their parameter counts on 0.013 to 0.003% of their training token counts. *Reported in Mercat et al. (2024).

test if LoLCATs can create subquadratic LLMs that outperform modern Transformer alternatives pretrained from scratch. For space, we defer linearizing training details to Appendix A.

In Table 4, we report results on six popular LM Evaluation Harness (LM Eval) tasks (Gao et al., 2023). Compared to recent linearizing methods, LoLCATs significantly improves quality and training efficiency across tasks and LLMs. On quality, LoLCATs closes 79.8% and 86.6% of the Transformer-linearizing gap for Mistral 7B and Llama 3 8B respectively, notably improving 5-shot MMLU by 60.9% and 40.9% over next best fully subquadratic models (17.2 and 9.6 points). On efficiency, we achieve these results while only training <0.2% of model parameters via LoRA versus prior full-parameter training. We also only use 40M tokens versus the prior 20 – 100B (a 500 – 2500× improvement in "tokens-to-model" efficiency). Among all 7B LLMs, LoLCATs-linearized LLMs further outperform strong subquadratic Transformer alternatives, representing RNNs or linear attentions (RWKV-v6 (Peng et al., 2024), Hawk, Griffin (De et al., 2024), TransNormer (Qin et al., 2023)), state-space models (Mamba (Gu & Dao, 2023)), and hybrid architectures with some full attention (StripedHyena (Poli et al., 2023b), Zamba (Glorioso et al., 2024)).

### 4.2 LoLCATs COMPONENT PROPERTIES AND ABLATIONS

We next validate that LoLCATs enables subquadratic efficiency, and study how each of LoLCATs' components contribute to these quality gains. We include additional ablations in Appendix B.

**Subquadratic Generation Throughput and Memory.** We measure generation throughput and memory of LoLCATs LLMs, validating that linearizing LLMs can significantly improve their generation efficiency. We use the popular Llama 3 8B HuggingFace model[3], and compare LoL-

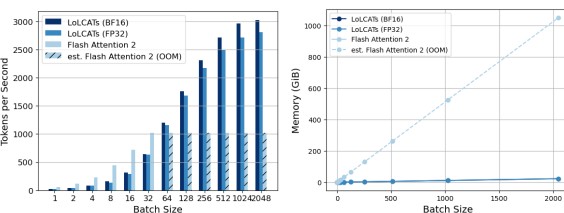

Figure 8: **Generation Efficiency**, Llama 3 8B.

[3] https://huggingface.co/meta-llama/Meta-Llama-3-8B

| Feature Map | LM Eval Metric | Swap & Finetune | +Attention Transfer | +Sliding Window, +Attention Transfer | + Sliding Window, No Attention Transfer |
|---|---|---|---|---|---|
| Hedgehog | Average | 44.20 | 55.32 | **70.66** | 68.78 |
| | MMLU (5-shot) | 23.80 | 23.80 | **52.77** | 45.80 |
| T2R | Average | 38.84 | 54.83 | **68.28** | 39.52 |
| | MMLU (5-shot) | 23.20 | 23.10 | **40.70** | 23.80 |

Table 5: **LoLCATs component ablations**, linearizing Llama 3 8B over 1024-token sequences. LoLCATs default shaded. Across Hedgehog and T2R feature maps, LoLCATs' attention transfer and sliding window increasingly improve linearized LLM quality. Full task results in App. B.1.1.

| | PiQA acc | ARC Easy acc | ARC Challenge (acc norm) | HellaSwag (acc norm) | WinoGrande acc | MMLU (5-shot) acc |
|---|---|---|---|---|---|---|
| Llama 3.1 8B | 79.87 | 81.52 | 53.58 | 79.01 | 73.48 | 66.11 |
| Linearized, no attn. transfer | 78.67 | 78.11 | 49.83 | 77.83 | 68.51 | 51.44 |
| LoLCATs (Ours) | **80.96** | **82.37** | **54.44** | **79.07** | 69.69 | **54.88** |
| Llama 3.1 70B | 83.10 | 87.30 | 60.60 | 85.00 | 79.60 | 78.80 |
| Linearized, no attn. transfer | 81.99 | 80.89 | 54.44 | 82.29 | 71.19 | 28.74 |
| LoLCATs (Ours) | 82.10 | **84.98** | 60.50 | **84.62** | 73.72 | **67.70** |
| Llama 3.1 405B | 85.58 | 87.58 | 66.21 | 87.13 | 79.40 | 82.98 |
| Linearized, no attn. transfer | 84.44 | 86.62 | 64.33 | 86.19 | 79.87 | 33.86 |
| LoLCATs (Ours) | 85.58 | **88.80** | **67.75** | **87.41** | **80.35** | 72.20 |

Table 6: **Linearizing Llama 3.1 8B, 70B, and 405B**. Among the first linearized 70B and 405B LLMs (via low-rank linearizing), LoLCATs significantly improves zero- and few-shot quality.

CATs implemented in HuggingFace Transformers with the supported FlashAttention-2 (FA2) (Dao, 2023). We benchmark LoLCATs with Hedgehog feature map and linear + sliding window attention in FP32 and BF16 on one 80GB H100. Fig. 8 reports scaling batch size on throughput (left) and memory (right). We measure throughput as (generated tokens × batch size / total time), with 128-token prompts and 4096-token generations. With larger batch size, LoLCATs-linearized LLMs achieve higher throughput than FA2. This corresponds with lower memory, where FA2 exceeds memory at batch size 64. Meanwhile, LoLCATs supports 3000 tokens / second at batch size 2048 (Fig. 8 left), with fixed "KV state" vs. softmax attention's growing KV cache (Fig. 8 right).

**Ablations.** We study how attention transfer and linear + sliding window attention in LoLCATs contribute to downstream linearized Llama 3 8B performance (Table 5). We start with prior linear attentions (Hedgehog, Zhang et al. (2024); T2R, Kasai et al. (2021)), using the prior linearizing approach that swaps attentions and finetunes the model to predict next tokens (Mercat et al., 2024). We then add (i) attention transfer, (ii) sliding window attentions, or (iii) both. On average LM Eval score and 5-shot MMLU accuracy, LoLCATs' default performs best across feature maps.

## 4.3 SCALING UP LINEARIZING TO 70B AND 405B LLMS

We finally use LoLCATs to scale up linearizing to Llama 3.1 70B and 405B models. In Table 6, LoLCATs provides the first practical solution for linearizing larger LLMs, achieving significant quality improvements over prior linearizing approaches of swapping in attentions and finetuning (Mercat et al., 2024). With the same linear + sliding window layer, LoLCATs gets +39.0 points in 5-shot MMLU accuracy on Llama 3.1 70B, and +38.3 on Llama 3.1 405B. These results highlight LoLCATs' ability to linearize large-scale models with greater efficiency and improved performance, showing for the first time that we can scale up linearizing to 70B+ LLMs.

## 5 CONCLUSION

We propose LoLCATs, an efficient LLM linearizing method that (1) trains linear attentions to match an LLM's self-attentions, before (2) swapping the attentions and only finetuning the replacing attentions with LoRA. This reduces linearizing to learning good softmax attention approximations, and we study how to do so. On popular LM Eval tasks, LoLCATs enables LLMs with high quality and inference efficiency, outperforming prior Transformers alternatives while only updating 0.2% of model parameters and requiring 0.003% of LLM pretraining tokens. Our findings substantially improve linearizing quality and accessibility, enabling the first linearized 70B and 405B LLMs.

ETHICS STATEMENT

Our work deals with improving the efficiency of open-weight models. While promising for beneficial applications, increasing their accessibility also raises concerns about potential misuse. Bad actors could leverage our technique to develop LLMs capable of generating harmful content, spreading misinformation, or enabling other malicious activities. We focus primarily on base models, but acknowledge that linearizing could also be used on instruction-tuned LLMs; research on whether linearizing preserves guardrails is still an open question. We acknowledge the risks and believe in the responsible development and deployment of efficient and widely accessible models.

REPRODUCIBILITY

We include experimental details in Appendix A, and further implementation details with sample code for linearizing architectures and training in Appendix C.1. Our code is also available at https://github.com/HazyResearch/lolcats

ACKNOWLEDGEMENTS

We thank Mayee Chen, Ben Viggiano, Gautam Machiraju, Dan Fu, Sabri Eyuboglu, Tri Dao, and anonymous reviewers for helpful discussions on linear attention and paper feedback. We gratefully acknowledge the support of NIH under No. U54EB020405 (Mobilize), NSF under Nos. CCF2247015 (Hardware-Aware), CCF1763315 (Beyond Sparsity), CCF1563078 (Volume to Velocity), and 1937301 (RTML); US DEVCOM ARL under Nos. W911NF-23-2-0184 (Long-context) and W911NF-21-2-0251 (Interactive Human-AI Teaming); ONR under Nos. N000142312633 (Deep Signal Processing); Stanford HAI under No. 247183; NXP, Xilinx, LETI-CEA, Intel, IBM, Microsoft, NEC, Toshiba, TSMC, ARM, Hitachi, BASF, Accenture, Ericsson, Qualcomm, Analog Devices, Google Cloud, Salesforce, Total, the HAI-GCP Cloud Credits for Research program, the Stanford Data Science Initiative (SDSI), and members of the Stanford DAWN project: Meta, Google, and VMWare. The U.S. Government is authorized to reproduce and distribute reprints for Governmental purposes notwithstanding any copyright notation thereon. Any opinions, findings, and conclusions or recommendations expressed in this material are those of the authors and do not necessarily reflect the views, policies, or endorsements, either expressed or implied, of NIH, ONR, or the U.S. Government.

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

## A EXPERIMENTAL DETAILS

### A.1 MAIN RESULTS, LINEARIZING 7B AND 8B LLMS

**Setup.** We describe our setup for linearizing Mistral 7B (v0.1) (Jiang et al., 2023), Llama 3 8B (AI@Meta, 2024a), and Llama 3.1 8B (Dubey et al., 2024).

For linearizing layers, we replace softmax attentions with hybrid linear + sliding window analogs (Section 3.3.1), using Hedgehog's feature map for its prior quality (Zhang et al., 2024).

For the sliding window implementation, we considered two options: a standard sliding window where $w$ is the same for all tokens, and a "terraced" window where $w$ changes based on token index (Figure 9). While we found both comparable in quality (Table 18), the latter lets us exploit the new ThunderKittens (TK) DSL's (Spector et al., 2024) primitives for implementing fast CUDA kernels. Here we prefer contiguous blocks of size $w = 64$, which can quickly be computed in parallel on modern GPUs. We use this "terrace" implementation in our main results, and include further implementation details in Appendix C.2.

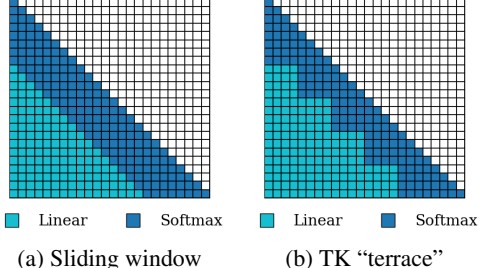

(a) Sliding window     (b) TK "terrace"

Figure 9: We apply softmax attention locally and attend to all past tokens with linear attention.

For linearizing data, we use the Alpaca linearizing data setup in Section 3.2 unless otherwise noted. We also tried a more typical pretraining corpus (a subset[4] of RedPajama (Computer, 2023)), but found comparable performance when controlling for number of token updates (Appendix B.5.1). To linearize, we simply train all feature maps in parallel for two epochs with learning rate 1e-2, before applying LoRA on the attention projection layers for two epochs with learning rate 1e-4. By default, we use LoRA rank $r = 8$, and scale LoRA updates by 2 ($\alpha = 16$ in HuggingFace PEFT[5]), amounting to training <0.09% of all model parameters. For both stages, we train with early stopping, AdamW optimizer (Loshchilov & Hutter, 2017), and packing into 1024-token sequences with batch size 8. We evaluate the best checkpoints based on validation set perplexity.

**Hyperparameters.** We list all model and training hyperparameters in Table 7. For learning rates, we did an initial sweep over {1e-2, 1e-3, 1e-4}, choosing the best based on final validation set perplexity during step 2: low-rank adjusting, and checkpointing with early stopping. We did not tune batch size or choice of optimizer, and used default values informed by prior work for other design parameters such as sliding window size (Arora et al., 2024), LoRA rank, and LoRA projection layers (Hu et al., 2021). In Appendix B, we study the effect of sweeping various values such as window sizes, ranks, and LoRA modules as ablations.

**Compute Resources.** For each linearizing run we use one NVIDIA 40GB A100 GPU. With batch size 1 and gradient accumulation over 8 batches, attention transfer takes $\approx 2$ hours and post-swap finetuning takes $\approx 4.5$ hours, *i.e.,* 6.5 total GPU hours to linearize an 8B LLM.

### A.2 LINEARIZING LLAMA 3.1 70B

We provide experimental details corresponding to the 70B parameter results reported in Table 6.

**Setup.** We compare the quality of two linearization approaches to the quality of the original Llama 3.1 70B model, including (1) the baseline linearization *without* attention transfer, which is representative of the approach used in prior work (Mercat et al., 2024; Yang et al., 2024; Wang et al., 2024) and (2) our approach, LOLCATs. For both the baseline and LOLCATs, we start with Llama 3.1

---

[4]https://huggingface.co/datasets/togethercomputer/
RedPajama-Data-1T-Sample
[5]https://huggingface.co/docs/peft/en/index

|  | Hedgehog | LoLCATs |
|---|---|---|
| **Model** | | |
| Precision | 16-bit (bfloat16) | |
| Sequence length | 1024 | |
| Linearizing attention | (Linear) | (Linear + Sliding Window) |
| Linear attn feature map | Hedgehog | |
| Linear attn feature dimension | 64 (effectively 128, see Table 1) | |
| Linear attn feature activation | Softmax (across feature dim) | |
| Sliding window implementation | N/A | Terrace |
| Sliding window attn size | N/A | 64 |
| **Optimizer and LR Schedule** | | |
| Optimizer | AdamW | |
| Global batch size | 8 | |
| Gradient accumulation | 8 | |
| Gradient clipping threshold | 1.0 | |
| Learning rate schedule | Reduce LR on Plateau | |
| **Step 1: Attention Transfer** | | |
| Number of epochs | 2 | |
| Tokens per epoch | 10M | |
| Learning rate | 0.01 | |
| **Step 2: Low-rank Adjusting** | | |
| Number of epochs | 2 | |
| Tokens per epoch | 20M | |
| Learning rate | 1e-4 | |
| LoRA rank and alpha | $r = 8, \alpha = 16$ | |
| LoRA dropout | 0.0 | |
| LoRA projections | $\boldsymbol{W}_q, \boldsymbol{W}_k, \boldsymbol{W}_v, \boldsymbol{W}_o$ | |

Table 7: Hyperparameters for Mistral 7B, Llama 3 8B, and Llama 3.1 8B experiments.

70B and replace the softmax attentions with the linear attention architecture defined in Section 3.3.1, Equation (6). The training procedure involves:

- Baseline: We introduce LoRA parameters to the attention $\boldsymbol{W}_q, \boldsymbol{W}_k, \boldsymbol{W}_v, \boldsymbol{W}_o$ projection matrices. We train the linear attention feature maps, learnable mixing term $\gamma$, and the LoRA parameters during the fine-tuning adjustment stage of the linearization process.

- LoLCATs: We first perform layer-wise attention transfer following Equation (7), with $k = 80$ (i.e., we optimize over all layers together). We then introduce LoRA parameters to the attention $\boldsymbol{W}_q, \boldsymbol{W}_k, \boldsymbol{W}_v, \boldsymbol{W}_o$ projection matrices. During fine-tuning we only train the LoRA parameters, freezing the linear attention map and $\gamma$ weights.

We use an MSE loss for the layer-wise attention transfer stage and cross-entropy loss for the next-token prediction fine-tuning (Low-rank Finetuning) stage.

**Hyperparameters.** We include the hyperparameters for the baseline and LoLCATs approaches in Table 8, following the same sweep as the 8B models. Since the baseline does not use attention transfer, we mark these values with "N/A". We linearize each model using the same randomly sampled 20M tokens of the RedPajama pre-training corpus (Computer, 2023). We pack the sequences to fill full context length, and evaluate the best checkpoints based on validation set perplexity.

**Compute Resources.** We linearize using a single NVIDIA $8 \times 80$GB H100 node. Attention transfer takes 4 hours and fine-tuning takes 14 hours. We use PyTorch FSDP with activation checkpointing for distributed training.

|  | Baseline | LoLCATs |
|---|---|---|
| **Model** | | |
| Precision | | 16-bit (bfloat16) |
| Sequence length | | 1024 |
| Linearizing attention | | Linear + Sliding Window |
| Linear attn feature map | | Hedgehog |
| Linear attn feature dimension | | 64 (effectively 128, see Table 1) |
| Linear attn feature activation | | Softmax (across feature dim) |
| Sliding window implementation | | Terrace |
| Sliding window attn size | | 64 |
| **Optimizer and LR Schedule** | | |
| Optimizer | | AdamW |
| Global batch size | | 8 |
| Gradient accumulation | | 8 |
| Gradient clipping threshold | | 1.0 |
| Learning rate schedule | | Reduce LR on Plateau |
| **Stage 1: Attention Transfer** | | |
| Number of epochs | N/A | 1 |
| Tokens per epoch | N/A | 20M |
| Learning rate | N/A | 0.01 |
| **Stage 2: Low-rank Adjusting** | | |
| Number of epochs | | 1 |
| Tokens per epoch | 20M | 20M |
| Learning rate | | 1e-4 |
| LoRA rank and alpha | | $r = 8, \alpha = 16$ |
| LoRA dropout | | 0.0 |
| LoRA projections | | $\boldsymbol{W}_q, \boldsymbol{W}_k, \boldsymbol{W}_v, \boldsymbol{W}_o$ |

Table 8: Hyperparameters for Llama 3.1 70B experiments.

## A.3 LINEARIZING LLAMA 3.1 405B

**Setup.** We compare the quality of two linearization approaches to the quality of the original Llama 3.1 405B model, including (1) the baseline linearization *without* attention transfer, which is representative of the approach used in prior work (Mercat et al., 2024) and (2) our approach, LoLCATs. For both the baseline and LoLCATs, we start with Llama 3.1 405B and replace the softmax attentions with the linear attention architecture defined in Section 3.3.1, Equation (6). The training procedure involves:

- Baseline: We introduce LoRA parameters to the attention $\boldsymbol{W}_q, \boldsymbol{W}_k, \boldsymbol{W}_v, \boldsymbol{W}_o$ projection matrices. We train the linear attention feature maps, learnable mixing term $\gamma$, and the LoRA parameters during the fine-tuning adjustment stage of the linearization process.

- LoLCATs: We first perform block-wise attention transfer following Equation (7), with $k = 9$ as the block size. To perform attention transfer for block $i$, we save the hidden states outputted by block $i - 1$ to disk and then use this as training data for block $i$. We then introduce LoRA parameters to the attention $\boldsymbol{W}_q, \boldsymbol{W}_k, \boldsymbol{W}_v, \boldsymbol{W}_o$ projection matrices. During fine-tuning we only train the LoRA parameters, freezing the linear attention map and $\gamma$ weights.

For the reported checkpoints, we train the layer-wise attention transfer stage using use a weighted combination of the MSE loss on attention outputs plus a cross-entropy loss between the softmax and linear attention maps. We use cross-entropy loss for Stage 2 Low-rank Linearizing.

**Hyperparameters.** We include the hyperparameters for the baseline and LoLCATs approaches in Table 9, following the same sweep as the 8B models. Since the baseline does not use attention transfer, we mark it with "N/A". We linearize each model using the same randomly sampled 20M

tokens of the RedPajama pre-training corpus (Computer, 2023). We pack the sequences to fill the full context length, and evaluate the best checkpoints based on validation set perplexity.

| | Baseline | LoLCATs |
|---|---|---|
| **Model** | | |
| Precision | | 16-bit (FP16) |
| Sequence length | | 1024 |
| Linearizing attention | | Linear + Sliding Window |
| Linear attn feature map | | Hedgehog |
| Linear attn feature dimension | | 64 (effectively 128, see Table 1) |
| Linear attn feature activation | | Softmax (across feature dim) |
| Sliding window implementation | | Terrace |
| Sliding window attn size | | 64 |
| **Optimizer and LR Schedule** | | |
| Optimizer | | AdamW |
| Global batch size | | 8 |
| Gradient accumulation | | 8 |
| Gradient clipping threshold | | 1.0 |
| Learning rate schedule | | Reduce LR on Plateau |
| **Stage 1: Attention Transfer** | | |
| Number of epochs | N/A | 1 |
| Tokens per epoch | N/A | 20M |
| Learning rate | N/A | 0.01 |
| MSE and X-ent weights | N/A | 1000, 1 |
| **Stage 2: Low-rank Adjusting** | | |
| Number of epochs | | 1 |
| Tokens per epoch | 20M | 20M |
| Learning rate | | 1e-4 |
| LoRA rank and alpha | | $r = 4, \alpha = 8$ |
| LoRA dropout | | 0.5 |
| LoRA projections | | $\boldsymbol{W}_q, \boldsymbol{W}_k, \boldsymbol{W}_v, \boldsymbol{W}_o$ |

Table 9: Hyperparameters for Llama 3.1 405B experiments.

**Compute Resources.** We linearize the baseline using three NVIDIA $8 \times 80$GB H100 nodes, evaluating the best validation checkpoint after 19.5 hours. For LoLCATs, we perform attention transfer with 1 80GB H100 GPU for 5 hours per block, and we finetune with 3 NVIDIA $8 \times 80$GB H100 nodes for 16 hours. We use PyTorch FSDP with activation checkpointing for distributed training.

## B    ADDITIONAL EXPERIMENTS AND RESULTS

To better understand LoLCATs' properties and performance, we now report extended results on LoLCATs. We first report expanded results of our main paper, including task-specific ablation numbers, multiple seeds, and additional comparison at the 1B LLM scale. We then extend our ablations by studying how different amounts of parameter updates, different data sources, and different amounts of training data affect LoLCATs quality for various tasks such as zero-shot LM Eval, 5-shot MMLU, and passkey retrieval. We finally expand on the layer-wise training dynamics of LoLCATs: how layer-wise MSE between learned and softmax attentions changes during LoRA adjusting (App. B.7.1), how LoRA weights update over time depending on attention transfer and softmax attention-matching quality (App. B.7.2), and how block-wise attention transfer both is motivated by layer-wise MSE and improves linearized LLM quality for larger LLMs (App. B.7.3).

## B.1 Expanded Results

### B.1.1 Task-specific Results for LoLCATs Component Ablations

We report the task-specific LM Eval results when ablating the attention transfer and linear + sliding window attention in LoLCATs, expanding on Table 5 in Table 10. Across all but one task (ARC-easy), we validate that the LoLCATs proposed combination leads to best performance.

| Feature Map | +Attention Transfer | +Sliding Window | PiQA | ARC-e | ARC-c (acc. norm) | HellaSwag (acc. norm) | Winogrande | MMLU (5-shot) | Average |
|---|---|---|---|---|---|---|---|---|---|
| Hedgehog | ✗ | ✗ | 67.8 | 58.1 | 28.9 | 35.8 | 50.8 | 23.8 | 44.2 |
| | ✓ | ✗ | 76.5 | 72.6 | 40.1 | 65.6 | 53.3 | 23.8 | 55.3 |
| | ✗ | ✓ | 80.5 | 80.6 | 53.4 | 78.8 | 73.6 | 45.8 | 68.8 |
| | ✓ | ✓ | **80.9** | 81.7 | **54.9** | **79.7** | **74.1** | **52.8** | **70.7** |
| T2R | ✗ | ✗ | 62.0 | 42.1 | 24.7 | 32.7 | 48.3 | 23.2 | 38.8 |
| | ✓ | ✗ | 76.1 | 72.8 | 40.8 | 63.6 | 52.6 | 23.1 | 54.8 |
| | ✗ | ✓ | 54.8 | 26.3 | 26.2 | 56.4 | 49.6 | 23.8 | 39.5 |
| | ✓ | ✓ | 80.7 | **82.0** | 54.6 | 79.5 | 72.2 | 40.7 | 68.3 |

Table 10: **LoLCATs component ablations on individual LM Eval tasks**, linearizing Llama 3 8B, expanded view of Table 5. LoLCATs default shaded. Across Hedgehog and T2R feature maps, LoLCATs' attention transfer and sliding window increasingly improve linearized LLM quality on popular LM Eval tasks both on average and specifically for 5-shot MMLU scores.

### B.1.2 Llama 3 8B LoLCATs Results Across Multiple Runs

In Table 11, we report the LM Eval results for Llama 3 8B models linearized with two linear attention approaches, Hedgehog (Zhang et al., 2024) and LoLCATs (ours), after doing linearizing across three seeds (0, 1, 2). We report means and standard deviations (in parentheses). Across all tasks, variation across seeds is low (under 1 point, other than Winogrande), with the difference in task performance across methods being much higher (greater than 10 points).

| Llama 3 8B | PiQA | ARC-e | ARC-c (norm) | HellaSwag (norm) | Winogrande | MMLU (5-shot) | Average | Average (no MMLU) |
|---|---|---|---|---|---|---|---|---|
| Hedgehog | 76.86 (0.32) | 73.27 (0.67) | 40.76 (0.69) | 65.77 (0.38) | 53.42 (0.22) | 24.22 (0.62) | 55.72 (0.35) | 62.02 (0.35) |
| LoLCATs (Ours) | **80.79 (0.11)** | **81.62 (0.41)** | **54.73 (0.41)** | **79.48 (0.07)** | **72.92 (1.02)** | **52.74 (0.64)** | **70.38 (0.33)** | **73.91 (0.29)** |

Table 11: **LoLCATs comparison with prior linearizing methods, multiple runs**; expanded view of linear attention Llama 3 8B results in Table 3. We report mean and (standard deviation) across three seeds, comparing LoLCATs and prior linear attention Hedgehog linearizing method.

## B.2 Closing Performance Gaps with Hybridizing LoLCATs across Layers

We now study how LoLCATs models perform while keeping some layers as full softmax attention. For Llama 3 8B, we keep the first 16 layers (half of all layers) as full softmax attention and only linearize the last half with the sliding window + linear attention layer for Llama 3 8B. We then just do attention transfer over Alpaca samples, instead of the two-stage attention transfer and subsequent LoRA finetuning as default in Table 7 (keeping Stage 1 details the same). In Table 12, we find that keeping just 50% softmax attention, while only doing Stage 1 transfer on 20 million tokens, substantially closes the MMLU performance gap by 94.2% (65.8% hybrid versus 52.8% default LoLCATs, compared to Transformer 66.6% accuracy). Furthermore, while just training on 20 million tokens (half that of the default protocol), we recover performance on all other tasks.

## B.3 LoLCATs Evaluation for Linearizing 1B+ LLMs

We now study LoLCATs' performance when linearizing $\sim$ 1B parameter LLMs. We choose two popular models, Llama 3.2 1B (AI@Meta, 2024b) and Phi 1.5 1.3B (Li et al., 2023), and linearize with LoLCATs using the same hyperparameters, architectures, and training details (*e.g.,* 40M tokens over Alpaca) as Llama 3 8B (Table 7) except for linear attention feature dimension, instead adjusting this to the 1B model head dimensions (*i.e.,* 0.5 × head dimension 64 = 32). To evaluate

| Softmax Attn. Layers | Tokens (B) | PiQA | ARC-E | ARC-C | HellaSwag | WinoGrande | MMLU |
|---|---|---|---|---|---|---|---|
| All (Llama 3 8B baseline) | 0 | 79.9 | 80.1 | 53.3 | 79.1 | 73.1 | 66.6 |
| 0-15 (LoLCATS 50%, Just Attn Transfer) | **0.02** | 79.5 | 80.2 | 53.4 | **79.2** | 73.6 | **65.8** |
| None (LoLCATS, Attn Transfer + LoRA) | 0.04 | **80.9** | **81.7** | **54.9** | 79.0 | **74.1** | 52.8 |

Table 12: **Hybrid LoLCATS, Llama 3 8B, LM Eval**. With just 50% softmax attention, training on 20 million tokens (half that of the default protocol) results in substantially closing the gap to full softmax attention Transformers on MMLU (+13 points absolute), and maintaining performance on other LM Eval tasks.

| Model | PiQA | ARC-e | ARC-c (acc. norm) | HellaSwag (acc. norm) | Winogrande | MMLU (5-shot) | Avg. | Avg. (No MMLU) | Avg. % of Transformer |
|---|---|---|---|---|---|---|---|---|---|
| Llama 3.2 1B | 74.4 | 65.5 | 35.8 | 63.7 | 60.5 | 31.9 | 55.3 | 60.0 | - |
| → T2R | 69.2 | 58.2 | 29.9 | 42.6 | 54.1 | 23.3 | 46.2 | 50.8 | 84.7 |
| → Hedgehog | 70.1 | 55.8 | 29.8 | 47.7 | 50.7 | 23.0 | 46.2 | 50.8 | 84.7 |
| → LoLCATs (Ours) | **74.6** | **63.0** | **35.1** | **63.7** | **61.5** | **27.3** | **54.2** | **59.6** | **99.3** |

Table 13: **LoLCATs comparison with linearizing methods, Llama 3.2 1B, LM Eval**. Following the two-step linearizing procedure in LoLCATS, we compare the LoLCATS architecture against the prior Transformer-to-RNN (T2R) and Hedgehog linear attentions. At the 1B scale, LoLCATS substantially closes the performance gap with original Llama 3.2 1B Transformer.

LoLCATs quality, we compare the LM Eval Harness performance of LoLCATS with other available linearizing methods on each Transformer LLM (Table 13, Table 14), and against competitive 1B subquadratic LLMs pretrained from scratch (Table 15), reporting results from Bick et al. (2024).

In all evaluation settings, LoLCATs demonstrates competitive or state-of-the-art linearizing quality. When first controlling for the original pretrained Transformer, LoLCATS is able to outperform both prior linear attentions, as well as pure and hybrid Phi-Mamba models created with MOHAWK, another method to distill Transformers into SSM-based Mamba architectures (Bick et al., 2024). Notably, LoLCATS makes this possible while only using 1.33% of the training tokens in MOHAWK (Table 14), and again only using parameter-efficient updates. Furthermore, by linearizing Llama 3.2 1B and Phi 1.5 1.3B, LoLCATS efficiently creates subquadratic 1B+ LLMs that outperform various LLMs pretrained from scratch (Table 15). These results suggest LoLCATS remains a competitive option for linearizing LLMs at multiple parameter counts.

## B.4 STUDY ON PARAMETER-EFFICIENT TRAINING

In our main results, we found that simple default initializations (*e.g.,* rank 8, applied to all attention projection layers) could recover high quality linearizing while only updating <0.2% of model parameters. In this section, we study how changing different aspects of low-rank adaptation and sliding window size impact linearizing performance.

### B.4.1 EFFECT OF LoRA RANK

We study the effect of LoRA rank in post-swap finetuning for zero-shot linearized LLM performance. Following standard implementations (Hu et al., 2021), we consider two factors. First: rank $r$, which determines the rank of the low-rank matrices $A$, $B$ we decompose the weight deltas into. Second: alpha $\alpha$, where $\alpha/r$ is a scaling factor that controls the degree to which $BA$ affect the output (*i.e.,* LoRA output $y = Wx + \frac{\alpha}{r}BAx$).

**Setup.** We sweep over ranks {4, 8, 16, 32, 64, 128, 256}, and adjust $\alpha$ such that $\alpha/r = 2$ as a default scaling factor (Raschka, 2023). For comparison, we also do a full finetuning run, where after attention transfer we do the stage 2 adjustment but training all Llama 3 8B parameters. For all runs, we linearize Llama 3 8B. For LoRA runs, we use the same default experimental setup as our main results, while for the full finetuning we decreased learning rate to 1e-5 due to training instability. We start with the linear + "terrace" window attentions, training the feature maps via attention transfer over 20M tokens over Alpaca (2 full epochs). We then freeze feature maps and apply LoRA with the above ranks on all attention weight projections (freezing all other parameters such as those in MLPs or GroupNorms), finetuning for two more epochs over Alpaca using the hyperparameters in Table 7. We evaluate with LM Eval tasks.

| Model | Training Tokens (B) | PiQA | ARC-e | ARC-c (acc. norm) | HellaSwag (acc. norm) | Winogrande | MMLU (5-shot) | Avg. (No MMLU) | Avg. % of Transformer |
|---|---|---|---|---|---|---|---|---|---|
| Phi 1.5 1.3B (MOHAWK) | 150 | 76.6 | 75.6 | 48.0 | 62.6 | 73.4 | - | 67.2 | - |
| Phi 1.5 1.3B (Our run) | 150 | 76.6 | 76.1 | 47.6 | 62.6 | 72.8 | 43.6 | 67.1 | - |
| Phi-Mamba 1.5 | 3 | 75.5 | 74.0 | 44.1 | 60.2 | 71.7 | - | 65.1 | 96.8 |
| Hybrid Phi-Mamba 1.5 | 3 | 76.5 | 75.3 | 45.8 | 60.6 | 72.0 | - | 66.0 | 98.2 |
| Phi 1.5 1.3B T2R | **0.04** | 71.0 | 69.1 | 36.6 | 46.2 | 53.6 | 24.3 | 55.3 | 82.4 |
| Phi 1.5 1.3B Hedgehog | **0.04** | 72.7 | 70.9 | 38.0 | 49.4 | 54.1 | 23.5 | 57.0 | 85.0 |
| Phi 1.5 1.3B LoLCATs (Ours) | **0.04** | **76.9** | **77.0** | **46.9** | **62.3** | **72.7** | **39.2** | **67.2** | 100.1 |

Table 14: **LoLCATs comparison with linearizing methods, Phi 1.5 1.3B, LM Eval**. We compare LoLCATs with linearizing with prior linear attentions, and available results from Bick et al. (2024), who distill Phi 1.5B into Mamba and hybrid Mamba-Transformer architectures with their MOHAWK method (Phi-Mamba 1.5, Hybrid Phi-Mamba 1.5). LoLCATs similarly outperforms prior linearizing methods, closing the gap to the Transformer Phi 1.5 1.3B on PiQA, ARC-Easy and average zero-shot LM Eval (no MMLU).

| Model | Training Tokens (B) | PiQA | ARC-e | ARC-c (acc. norm) | HellaSwag (acc. norm) | Winogrande | MMLU (5-shot) | Avg. (No MMLU) |
|---|---|---|---|---|---|---|---|---|
| **Transformer** | | | | | | | | |
| Pythia 1.4B | 300 | 71.1 | 60.6 | 26.0 | 52.1 | 57.3 | 26.6 | 53.4 |
| Llama 3.2 1B | 9000 | 74.4 | 65.5 | 35.8 | 63.7 | 60.5 | 31.9 | 60.0 |
| Phi 1.5 1B | 150 | 76.6 | 76.1 | 47.6 | 62.6 | 72.8 | 43.6 | 67.1 |
| **Subquadratic** | | | | | | | | |
| xLSTM 1.4B | 300 | 74.6 | 64.3 | 32.6 | 60.9 | 60.6 | - | 58.6 |
| Finch 1.6B (RWKV-v6) | 1100 | 72.6 | 64.2 | 34.1 | 57.3 | 59.4 | - | 57.5 |
| DeltaNet 1.3B | 100 | 71.2 | 57.2 | 28.3 | 50.2 | 53.6 | - | 52.1 |
| GLA 1.3B | 100 | 71.8 | 57.2 | 26.6 | 49.8 | 53.9 | - | 51.9 |
| Mamba 1 1.4B | 315 | 74.2 | 65.5 | 32.8 | 59.1 | 61.5 | - | 58.6 |
| Mamba 2 1.3B | 315 | 73.2 | 64.3 | 33.3 | 59.9 | 60.9 | - | 58.3 |
| Llama 3.2 1B LoLCATs (Ours) | **0.04** | 74.6 | 63.0 | 35.1 | **63.7** | 61.5 | 27.3 | 59.6 |
| Phi 1.5 1.3B LoLCATs (Ours) | **0.04** | **76.9** | **77.0** | **46.9** | 62.3 | **72.7** | 39.2 | **67.2** |

Table 15: **LoLCATs comparison to pretrained 1B LLMs**. LoLCATs-linearized Llama 3.2 1B and Phi 1.5 1.3B consistently outperform strong subquadratic 1B+ LLMs pretrained from scratch, achieving **best** or second-best accuracy on all tasks other than ARC-easy. Subquadratic results reported from Bick et al. (2024).

**Results.** We report results in Table 16. We find that when applying LoRA to adjust in linearizing after attention transfer, larger rank updates, *e.g.,* $r = 128$ or $256$, do not necessarily lead to improved zero or few-shot downstream performance. Full finetuning also does not improve performance. Somewhat surprisingly, using just $r = 4$ leads to overall best performance, while $r = 128$ and $r = 8$ achieve best and second-best 5-shot MMLU accuracy. While we leave further exploration for future work, we hypothesize that low-rank updates may improve quality by preventing large and potentially harmful updates with the linearizing data to pretrained weights. During the second step of adjusting, if the linearizing data is not particularly diverse or large (*e.g.,* for efficient linearizing), LoRA can then reduce the risk of overfitting to the linearizing data and losing LLM pretrained generalization.

### B.4.2 EFFECT OF LoRA PROJECTION LAYER

We next compare performance when applying LoRA to different weight matrices of the linear attention layers. With the same training and evaluation setup as Appendix B.4.1, but fixing $r = 8, \alpha = 16$, we now apply LoRA to different combinations of $W_q, W_k, W_v, W_o$ weights after swapping in attention-transferred linear attentions. We use the same combination for each layer.

We report results in Table 17. Interestingly, when isolating for projections updated, LoRA on projections *not involved* in computing layer-wise attention weights (value $W_v$ and output $W_o$ projections) improves quality compared to query $W_q$ or key projections $W_k$. Somewhat surprisingly, updating just $W_v$ or $W_o$ achieves comparable performance to updating all projections (*c.f.,* average zeros-shot accuracy of 74.19% when updating just $\{W_v\}$, 74.09% for $\{W_o\}$ versus 74.24% updating $\{W_q, W_k, W_v, W_o\}$). Meanwhile, updating just $W_q$ or $W_k$ performs significantly worse (72.68%, 72.29%; versus 74.24%). This suggests much of the quality recovery in Stage-2 low-rank finetuning comes from adjusting values and outputs to the learned attention weights—as opposed to further refining attention weight computation. While best results do come from a combination of adapting either $W_q$ or $W_k$ with value and output projections, we may be able to achieve even more parameter-efficient linearizing—with comparable quality—by focusing on subsets with the latter.

| LoRA Rank | PiQA | ARC-e | ARC-c (norm) | HellaSwag (norm) | Wino-grande | MMLU (5-shot) | Avg. | Avg. (no MMLU) |
|---|---|---|---|---|---|---|---|---|
| 4 | 80.7 | **82.5** | **56.0** | 79.6 | 73.6 | 50.1 | **71.3** | **74.5** |
| 8 | 80.9 | 81.7 | 54.9 | **79.7** | **74.1** | 52.8 | 70.7 | 74.2 |
| 16 | 80.7 | 81.9 | 54.5 | **79.7** | 73.8 | 48.9 | 69.9 | 74.1 |
| 32 | **81.1** | 81.5 | 54.5 | **79.7** | 72.8 | 51.0 | 70.1 | 73.9 |
| 64 | 80.9 | 81.9 | 54.5 | 79.3 | 72.1 | 51.7 | 71.1 | 73.8 |
| 128 | 80.5 | 80.9 | 52.8 | 78.4 | 72.2 | **53.4** | 69.7 | 73.0 |
| 256 | 80.7 | 80.2 | 52.1 | 78.8 | 71.4 | 52.1 | 69.2 | 72.7 |
| Full Finetune | 80.6 | 81.9 | 54.3 | 79.4 | 72.4 | 52.1 | 70.1 | 73.7 |

Table 16: **LoRA rank $r$ comparison**. Evaluation on LM Evaluation Harness tasks. When adapting all projections $W_q, W_k, W_v, W_o$ after attention transfer, we find smaller ranks $r = 4, 8$ are surprisingly sufficient. Larger ranks with more parameter-heavy updates do not necessarily improve downstream performance.

| LoRA Projection | PiQA | ARC-e | ARC-c (norm) | HellaSwag (norm) | Winogrande | Average |
|---|---|---|---|---|---|---|
| $W_q$ | 79.49 | 81.06 | 51.45 | 79.18 | 72.22 | 72.68 |
| $W_k$ | 79.82 | 79.80 | 50.68 | 78.54 | 72.61 | 72.29 |
| $W_v$ | 80.69 | 82.49 | 56.66 | 79.28 | 71.82 | 74.19 |
| $W_o$ | 80.09 | 81.65 | 55.63 | 79.35 | 73.72 | 74.09 |
| $W_q, W_k$ | 79.76 | 81.19 | 51.02 | 79.29 | 72.22 | 72.70 |
| $W_q, W_v$ | 80.96 | 81.90 | 54.35 | 79.01 | 72.30 | 73.70 |
| $W_q, W_o$ | **81.28** | 81.86 | 54.78 | 79.20 | 73.72 | 74.17 |
| $W_k, W_v$ | 80.74 | 82.62 | **56.83** | 79.26 | 71.98 | 74.28 |
| $W_k, W_o$ | 80.69 | 82.15 | 55.46 | 79.53 | 73.01 | 69.33 |
| $W_v, W_o$ | 80.69 | **82.79** | 55.89 | 79.57 | 71.59 | 74.10 |
| $W_q, W_k, W_v$ | 80.90 | 82.20 | 56.48 | 79.13 | 73.95 | **74.53** |
| $W_q, W_k, W_o$ | 80.41 | 80.89 | 54.18 | 79.18 | **74.35** | 73.80 |
| $W_k, W_v, W_o$ | 80.36 | 82.32 | 54.61 | 79.13 | 73.56 | 74.00 |
| $W_q, W_k, W_v, W_o$ | 80.85 | 81.73 | 54.86 | **79.65** | 74.11 | 74.24 |

Table 17: **LoRA projection comparison**. Evaluation on zero-shot LM Evaluation Harness tasks. We apply LoRA with the same rank $r = 8$ to different combinations of the attention projections, shading scores by increasing quality (darker is better). When isolating for projections updated, LoRA on projections *not involved* in layer-wise attention weight computations (value $W_v$ and output $W_o$ projections) achieves higher quality over query $W_q$ or key projections $W_k$. This suggests $W_v, W_o$ may be more important to adapt after attention transfer, although best results involve a combination across attention weight and output projections.

### B.4.3 EFFECT OF WINDOW SIZE

We now compare model performance using different window sizes in the LoLCATS linear + sliding window attention layer. With the standard sliding window implementation (Fig 9), we compare LM Eval performance after linearizing with window sizes $w \in \{4, 16, 64, 256\}$. We also compare against the ThunderKittens-motivated "terraced" implementation used in our main experiments with window size 64. In Table 18, we find that in each of these settings, having more softmax attention generally improves performance (*c.f.*, $w = 16, 64$ versus $w = 4$). However, more softmax attention does not always lead to better quality. Window size 256 results in up to an 8.8 point drop in 5-shot MMLU accuracy, suggesting we may not necessarily trade-off more softmax attention for higher quality. Comparing the standard sliding window and terracing implementations, we find similar performance (70.3 versus 70.7 average accuracy across tasks).

### B.5 STUDY ON LINEARIZING DATA

While most of our work focuses on architecture and training procedure for improving LLM linearizing quality, we now study how data selection affects LoLCATS performance.

| Window Size | PiQA | ARC-e | ARC-c (norm) | HellaSwag (norm) | Winogrande | MMLU (5-shot) | Avg. | Avg. (no MMLU) |
|---|---|---|---|---|---|---|---|---|
| 4 | 80.7 | 81.4 | 55.8 | 76.6 | 72.1 | 40.8 | 67.9 | 73.3 |
| 16 | 80.5 | **82.2** | **56.0** | 78.2 | 73.9 | **50.3** | 70.2 | 74.1 |
| 64 | **80.7** | 81.7 | 54.7 | **79.1** | **75.3** | **50.3** | **70.3** | **74.3** |
| 256 | 80.6 | 81.8 | 55.0 | 75.6 | 74.9 | 41.5 | 68.3 | 73.6 |
| TK 64 | 80.9 | 81.7 | 54.9 | 79.7 | 74.1 | 52.8 | 70.7 | 74.2 |

Table 18: **Window size comparison**. We ablate the window size in LoLCATs linear + sliding window attention. For each window size $w$, the layer applies softmax attention to the $w$-most recent positions, combined with Hedgehog linear attention applied for all prior positions (Eq. 6, Figure 9). We compare $w \in \{4, 16, 64, 256\}$ using the standard sliding window implementation with our default $w = 64$ terraced window setup motivated by ThunderKittens (TK 64). Window size 64 performs best, where both implementations perform comparably.

| Model | Linearizing Dataset | PiQA | ARC-e | ARC-c (acc. norm) | HellaSwag (acc. norm) | Wino-grande | MMLU (5-shot) | Avg. | Avg. (no MMLU) |
|---|---|---|---|---|---|---|---|---|---|
| Mistral 7B (v0.1) | - | 82.1 | 80.9 | 53.8 | 81.0 | 74.0 | 62.4 | 72.4 | 74.4 |
| → LoLCATs | Alpaca Clean | **81.5** | **81.7** | **54.9** | **80.7** | **74.0** | 51.4 | **70.7** | **74.5** |
| → LoLCATs | RedPajama | 80.1 | 77.6 | 49.0 | 80.3 | 71.7 | **53.2** | 68.6 | 71.7 |
| Llama 3 8B | - | 79.9 | 80.1 | 53.3 | 79.1 | 73.1 | 66.6 | 72.0 | 73.1 |
| → LoLCATs | Alpaca Clean | **80.9** | **81.7** | **54.9** | **79.7** | **74.1** | 52.8 | **70.7** | **74.2** |
| → LoLCATs | RedPajama | 78.9 | 79.0 | 52.0 | 78.1 | 72.6 | **55.2** | 69.3 | 72.1 |

Table 19: **Linearizing data comparison**. For linearizing Mistral 7B (v0.1) and Llama 3 8B, LoLCATs with Alpaca and RedPajama subsets perform comparably (c.f. prior methods, Table 3), though we find that Alpaca actually leads to higher accuracy for most tasks other than 5-shot MMLU.

### B.5.1 DATA SOURCE: ALPACA VERSUS REDPAJAMA

We study the effect of linearizing data for downstream LLM performance. While we initially found that just using the $\sim 50K$ samples of a cleaned Alpaca dataset[6] (Taori et al., 2023) could lead to surprisingly high performance on popular zero-shot LM Eval tasks, prior linearizing works (Mercat et al., 2024) use more typical pretraining datasets to linearize such as RefinedWeb (Penedo et al., 2023). We thus also try linearizing with a random subset of RedPajama (Computer, 2023) to evaluate how LoLCATs works with pretraining data, albeit without any special curation. For both setups, we pack samples into 1024 token sequences and randomly subsample the RedPajama data so that we use the same number of training tokens (20M) for both attention transfer and finetune stages (40M tokens overall). We use the setup as described in Appendix A.1 for all other hyperparameters.

In Table 19, we find that across Mistral 7B (v0.1) and Llama 3 8B, using the Alpaca cleaned dataset actually leads to better downstream task quality for all tasks except for 5-shot MMLU, where linearizing with RedPajama consistently leads to $\sim 2$ percentage point improvements. LoLCATs with both of these datasets leads to comparable or higher performance than prior methods trained on $2500\times$ the data (*c.f.*, Table 3; SUPRA trained on 100B tokens gets Avg. accuracy of 64.0%), suggesting that LoLCATs can robustly improve linearizing quality over different data sources.

### B.5.2 MATCHING LINEARIZING DATA TO DOWNSTREAM (RETRIEVAL) TASK

Among LM Eval tasks, we note a sizable gap between linearized and Transformer-based LLMs on MMLU. We hypothesize one source of this gap is due to MMLU's evaluation setup, which not only tests for knowledge recall in pretrained weights, but also *retrieval* over the input context (Hendrycks et al., 2020). In the default 5-shot multiple choice setup, models must be able to retrieve and produce the letter associated with the right answer choice in context. However, prior works have shown that linear attentions and non-softmax attention models perform worse on retrieval, both explicitly on MMLU (Waleffe et al., 2024) and in retrieval tasks at large (Waleffe et al., 2024; Shen et al., 2024).

To counteract these effects, we study if linearizing LLMs with data that explicitly reflects the target downstream task can improve performance. We report two such settings next.

---

[6]https://huggingface.co/datasets/yahma/alpaca-cleaned

**Improving MMLU.** First, to test improving performance on MMLU, we linearize with additional data from CommonsenseQA (Talmor et al., 2019) (CQA), another multiple-choice dataset. We construct a linearizing dataset by using the same 5-shot in-context template as in MMLU, *i.e.,* `<bos><question 1><answer choices 1><answer 1>, <question 2><answer choices 2><answer 2>, ..., <question 5><answer choices 5>___<eos>`, where the next token to predict ___ corresponds to the correct answer among `<answer choices 5>`, using the ~10k samples of the CQA training set. For both stages, we linearize Llama 3 8B over the combined set of 5-shot CQA samples and our default ~50k Alpaca samples, comparing against just Alpaca or CQA samples. We pack all samples to context length 1024.

In Table 20, we report the MMLU scores following each linearizing data choice. We find that adding a small amount of multiple-choice samples can substantially improve MMLU accuracy ($\sim 2$ points). However, CQA alone performs $\sim 10$ points worse than Alpaca. This suggests quantity and diversity of samples may still be necessary for linearizing; we break this down further and study the amount of linearizing needed during both attention transfer and low-rank adjusting in Appendix B.6.

| Model | Llama 3 8B LoLCATs | | | Llama 3 8B |
|---|---|---|---|---|
| Linearizing Data | Alpaca | Alpaca + CQA | CQA | N/A |
| MMLU (5-shot) Acc. | 52.8 | **54.5** | 43.9 | 66.6 |

Table 20: **MMLU comparison with task-specific linearizing data**. Adding multiple-choice samples (CommonsenseQA; CQA) to linearizing training data modestly improves downstream MMLU performance. However, CQA alone is insufficient to achieve competitive quality.

**Evaluating and improving needle-in-a-haystack retrieval.** Next, to further test linearizing performance on downstream tasks, we evaluate the LoLCATs-linearized Llama 3 8B LLM on "needle-in-a-haystack" tasks. We use the passkey-retrieval task setup introduced in Mohtashami & Jaggi (2023) and evaluated in various prior works (Chen et al., 2023a;b; Tworkowski et al., 2024), where a model must retrieve a hidden passkey uniform randomly placed inside a text span. We use randomly generated prompts and 5-digit passkeys (Listing 1), and test correctness by whether the model outputs the passkey exactly (exact-match). We evaluate over various text lengths (2048 to 10240 tokens), and report accuracy binned by which decile the passkey occurs in.

Like in our MMLU study, we evaluate LoLCATs with two linearizing data setups. First, we use the default Alpaca data, but this time test linearizing Llama 3 8B at its max context length by packing the 50K Alpaca samples into 8192-token chunks. Second, to see if linearizing explicitly with retrieval data helps, we generate 10K passkey retrieval samples with ~8192 tokens, and use these samples for both stages of LoLCATs linearizing. For both, we use the same hyperparameters as Table 7.

In Table 21, we first report the results for Llama models evaluated on 8192-token prompts. We compare the LoLCATs Llama models against the base Llama 3 8B, the instruction-tuned version (Llama 3 8B Instruct), and a Transformer Llama 3 8B LoRA-finetuned on Alpaca data (Llama 3 8B (Alpaca); using the same LoRA parameters and second stage training procedure as in Table 7). We find that with LoLCATs models, linearizing data plays a particularly strong role in downstream performance. As a potential drawback of LoLCATs, when just using packed Alpaca samples, the linearized Llama 3 8B fails to get even a single passkey retrieval sample correct (LoLCATs Llama 3 8B (Alpaca)). Meanwhile, LoRA-finetuning the non-linearized Llama 3 8B maintains high passkey retrieval accuracy. However, linearizing with passkey samples (LoLCATs Llama 3 8B (Passkey)) recovers 100% accuracy. This suggests that with LoLCATs linearizing, the linear + sliding window attention is able to do passkey retrieval with similar performance to full softmax attention models.

| Passkey Placement | 0-10% | 10-20% | 20-30% | 30-40% | 40-50% | 50-60% | 60-70% | 70-80% | 80-90% | 90-100% |
|---|---|---|---|---|---|---|---|---|---|---|
| Llama 3 8B | **100.00** | **100.00** | **100.00** | **100.00** | 92.86 | **100.00** | 94.44 | 94.12 | 92.31 | **100.00** |
| Llama 3 8B Instruct | **100.00** | **100.00** | **100.00** | **100.00** | **100.00** | **100.00** | **100.00** | **100.00** | **100.00** | **100.00** |
| Llama 3 8B (Alpaca) | **100.00** | **100.00** | **100.00** | **100.00** | **100.00** | **100.00** | **100.00** | **100.00** | **100.00** | **100.00** |
| LoLCATs Llama 3 8B (Alpaca) | 0.00 | 0.00 | 0.00 | 0.00 | 0.00 | 0.00 | 0.00 | 0.00 | 0.00 | 0.00 |
| LoLCATs Llama 3 8B (Passkey) | **100.00** | **100.00** | **100.00** | **100.00** | **100.00** | **100.00** | **100.00** | **100.00** | **100.00** | **100.00** |

Table 21: **8K Context Passkey Retrieval**. We report passkey retrieval accuracy for various Llama 3 8B models, using 8192-token samples and binning by passkey placement decile. Linearizing with just packed Alpaca samples fails, but using passkey samples as the linearizing data recovers 100% retrieval performance.

```
1  There is an important piece of info hidden inside a lot of irrelevant
2  text. Find it and memorize it. I will quiz you about the important
3  information there.
4  The grass is green. The sky is blue. The sun is yellow. Here we go. There
5  and back again. The grass is green. The sky is blue. The sun is yellow.
6  Here we go. There and back again. The grass is green. The sky is blue.
7  The sun is yellow. Here we go. There and back again...
8  ...  <EVEN MORE FILLER>
9  The pass key is <PASSKEY>. Remember it. <PASSKEY> is the pass key.
10 The grass is green. The sky is blue. The sun is yellow. Here we go. There
11 and back again. The grass is green. The sky is blue. The sun is yellow.
12 Here we go. There and back again...
13 ...  <MORE FILLER>
14 What is the pass key? The pass key is
```

Listing 1: Passkey Retrieval Prompt Template

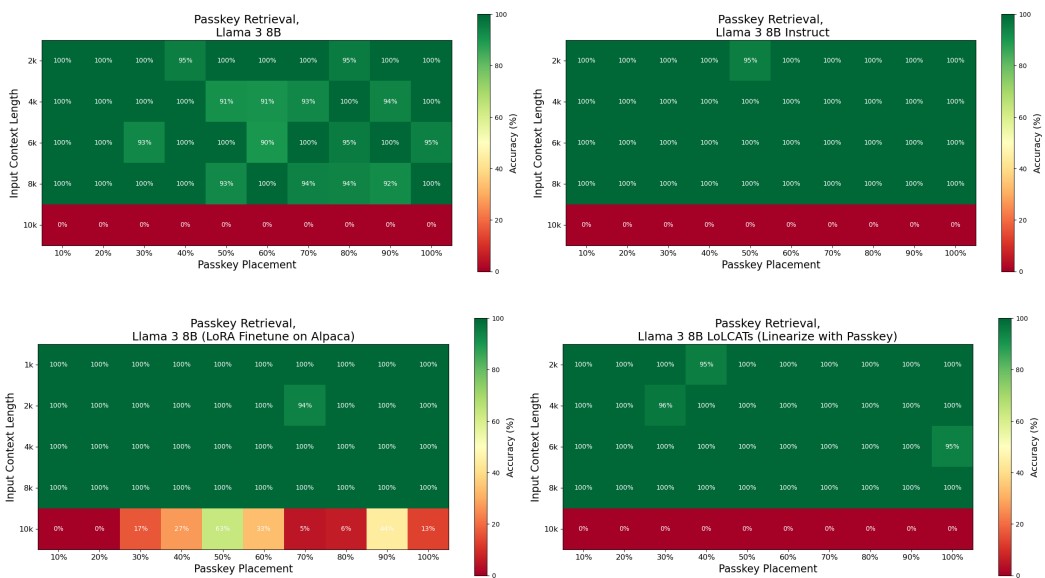

Figure 10: **Needle-in-a-haystack with passkey retrieval**. LOLCATS-linearized Llama 3 8B with passkey retrieval linearizing data results in comparable retrieval to original Llama 3 8B and instruction-tuned variants.

We further test the robustness of this retrieval across various context lengths (2048 to 10240 tokens) in Figure 10. We report accuracies across input context lengths and passkey placements, finding similar strong retrieval performance for samples under Llama 3 8B's 8192 context length. Interestingly, only the Transformer Llama 3 8B LoRA-finetuned on Alpaca is able to do retrieval over longer context samples (10240 tokens).

### B.5.3  SAMPLE LENGTHS: EFFECT OF EFFECTIVE SEQUENCE LENGTHS

We further study the impact of sample sequence length for linearizing quality. By default, for linearizing data we pack original data samples into sequences of consistent length, *e.g.,* 1024 tokens. As done in prior work (Raffel et al., 2020), this allows us to pack multiple short data samples together into longer training sequences, improving training efficiency and removing any padding tokens. However, it may also introduce situations where our linearizing sequences only carry short-context dependencies, *i.e.,* because we pack together many samples with few tokens, or split longer samples into multiple sequences. Especially with attention transfer, linearized LLMs may model longer samples less well (*e.g.,* the 5-example in-context samples in 5-shot MMLU) because we never learn to approximate attentions over "long enough" sequence lengths.

**Effective sequence length.**   To study this data effect, we define an "effective sequence length" (ESL) metric. This roughly captures for each query how far back a layer needs to attend to capture all non-zero softmax attention weights. For query at position $i$, we define the ESL per query as

$$\mathrm{ESL}(\boldsymbol{q}_i) := \sum_{j=1}^{i} (i - j) \frac{\exp(\boldsymbol{q}_i^\top \boldsymbol{k}_j / \sqrt{d})}{\sum_{m \leq j} \exp(\boldsymbol{q}_i^\top \boldsymbol{k}_m / \sqrt{d})} \tag{8}$$

We compute a sample's ESL per head as the sum over all query ESLs, *i.e.*, $\sum_{i=1}^{n} \mathrm{ESL}(\boldsymbol{q}_i)$ for a sample with $n$ tokens. We average this over all heads and layers to measure a sample's overall ESL.

We hypothesize that if our linearizing data only has samples with shorter ESL than those encountered at test time, then we would poorly model these test samples. Conversely, we may be able to improve linearizing quality by specifically filtering for samples with longer ESL. We report two findings next.

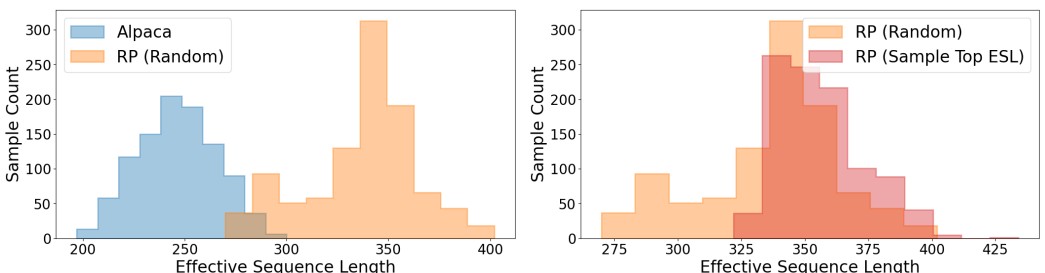

Figure 11: **Effective sequence length distributions**. Although we pack data samples into 1024-token training sequences, different data sources can vary in effective sequence lengths (left). Furthermore, we can selectively filter for longer ESL samples (right). We find linearizing with longer ESLs coincides with improved MMLU scores, albeit at the cost of other LM Eval tasks (Table 22).

**Finding 1: ESL corresponds with RedPajama versus Alpaca performance.**   In Figure 11, we first plot the distribution of sample ESLs computed with Llama 3 8B on Alpaca and RedPajama linearizing data subsets. We find RedPajama samples on average display longer ESLs, which conincides with improved MMLU score (*c.f.*, Table B.5.1).

**Finding 2: Filtering for higher ESL improves MMLU.**   Furthermore, we can increase the ESLs in linearizing data by actively filtering for high ESL samples (Figure 11 right). Here we actively filter for the top 20,000 packed RedPajama samples with the highest ESLs, amounting to 20M tokens. When doing one epoch of attention transfer and low-rank linearizing with this subset, we further improve MMLU accuracy by 1.3 points (Table 22). However, this comes at a cost for all other LM Eval tasks, dropping quality compared to random RedPajama packing by 0.2 to 2.1 points.

## B.6   STUDY ON LINEARIZING TOKEN BUDGET

We further study how varying the number of tokens used for both attention transfer and low-rank adaptation impacts LOLCATs linearizing quality.

**Impact of minimal tokens.** To first test how efficient we can be with attention transfer, we linearize Llama 3 8B with varying numbers of attention transfer steps (0 - 1800), before low-rank adjusting for up to 2000 steps. We use the Alpaca dataset and the same packed random sampling as our main experiments, and measure evaluation perplexity on validation samples both in-distribution (held-out

| Linearizing Data | PiQA | ARC-easy | ARC-challenge | HellaSwag (acc. norm) | Winogrande (acc. norm) | MMLU (5-shot) | Avg. (no MMLU) |
|---|---|---|---|---|---|---|---|
| Alpaca | **80.9** | **81.7** | **54.9** | **79.7** | **74.1** | 52.8 | **74.2** |
| RedPajama | 78.9 | 79.0 | 52.0 | 78.1 | 72.6 | 55.2 | 72.1 |
| RedPajama (Sample Top ESL) | 78.4 | 77.0 | 49.8 | 78.0 | 71.4 | **56.5** | 70.9 |

Table 22: **Effect of ESL on linearized LLM quality**, Llama 3 8B. While linearizing with longer ESLs—*e.g.,* RedPajama samples or specifically filtering for top ESLs in RedPajama (Sample Top ESL)—improves 5-shot MMLU accuracy up to 3.7 points, it reduces quality on all other evaluated LM Eval tasks by 1.7 to 5.0 points.

Alpaca samples) and out-of-distribution (RedPajama validation samples) over different combinations of steps (Figure 13). Without attention transfer, low-rank adaptation converges significantly higher on in-distribution samples (Figure 12a), suggesting poorer quality linearizing. However, we find similar held-out perplexities after relatively few attention transfer steps (*c.f.*, 1000 - 1800 updates, the former amounting to just 8 million tokens for attention transfer), where all runs improve in-distribution PPL by ~0.23 points after LoRA finetuning for 2000 steps.

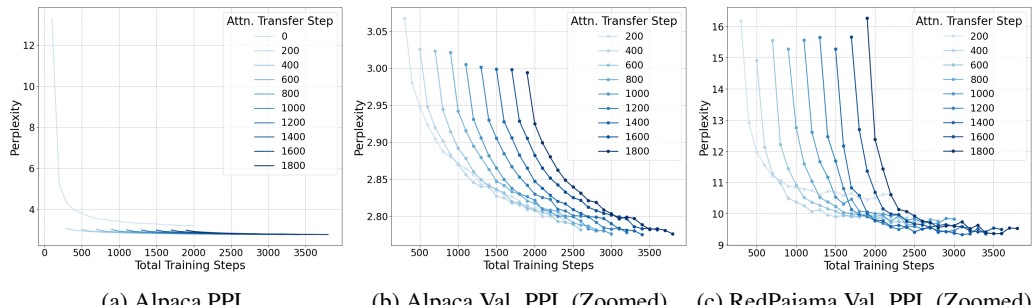

| (a) Alpaca PPL | (b) Alpaca Val. PPL (Zoomed) | (c) RedPajama Val. PPL (Zoomed) |

Figure 12: **Evaluation curves over number of training steps**. Llama 3 8B linearized with Alpaca. We report the impact of steps allotted to attention transfer versus LoRA linearizing, using validation set perplexity (PPL) over both in-distribution (held-out Alpaca samples) and out-of-distribution (RedPajama validation samples). We first run linearizing with 0 - 1800 attention transfer steps, before LoRA-finetuning for up to 2000 steps. Without any attention transfer (0 steps), linearized LLMs get much higher perplexity (Fig 12a). On the other hand, we observe similar convergence after attention transfer over only 1000 steps.

In Table 23, we report the numerical values for held-out perplexities at the end of linearizing (1800 attention transfer steps + 2000 low-rank adaptation steps), as well as the average LM Eval score over zero-shot tasks. We similarly find competitive generalized zero-shot quality with relatively few attention transfer steps (200 steps), all achieving 7.70–8.16 higher points than the next best Mamba-Llama model (0.16–0.62 higher points than the 50% softmax attention variant, *c.f.*, Table 3). Without any attention transfer, linearized LLMs perform drastically worse on out-of-distribution samples (Table 23, RedPajama and LM Eval metrics).

**Impact of more pretraining data.** We next study how linearizing over larger amounts of pretraining data impacts quality. We randomly sample a larger set of unique RedPajama training sequences (1024-token packed; 72,000 such samples overall), allowing us to linearize Llama 3 8B with different combinations of up to 9,000 attention transfer updates and 2,000 low-rank linearizing updates. To test language modeling recovery, we report both held-out validation sample perplexity (Table 13a) and general zero-shot LM Evaluation Harness quality (Table 13b). Increasing both Stage 1 attention transfer steps and Stage 2 low-rank adjusting steps notably improves validation perplexity. However, we similarly find competitive zero-shot LM scores across all evaluated attention transfer steps. Across checkpoints at different numbers of low-rank updates, attention transfer with up to 9× more unique tokens does not seem to monotonically improve downstream quality (Table 13b). Meanwhile, we do find that across various amounts of attention transfer steps, subsequent low-rank adaptation consistently improves average zero-shot LM score by >1 points.

## B.7 Layer-wise Training Dynamics and Analysis

Finally, we further study LoLCATs layer-wise training dynamics, such as the resulting linear attention approximation quality to softmax attention at every layer (App. B.7.1), how this corresponds

| Attn. Transfer Steps | 0 | 200 | 400 | 600 | 800 | 1000 | 1200 | 1400 | 1600 | 1800 |
|---|---|---|---|---|---|---|---|---|---|---|
| Alpaca Eval PPL | 3.211 | 2.802 | 2.792 | 2.782 | 2.782 | 2.776 | 2.778 | **2.775** | 2.782 | 2.776 |
| RedPajama Eval PPL | 61.305 | 10.459 | 9.799 | 9.699 | 9.628 | 9.679 | 9.420 | **9.328** | 9.413 | 9.358 |
| Avg. Zero-shot LM Eval Acc. | 56.86 | 73.68 | 73.34 | 73.70 | 73.66 | 73.74 | 73.47 | 73.70 | **73.80** | 73.66 |

Table 23: **Effect of attention transfer steps**. With Llama 3 8B linearized on Alpaca data, we report the final evaluation perplexities after 2000 LoRA steps in Fig 13, as well as downstream LM Eval performance averaged over zero-shot tasks. We again find competitive quality with relatively few attention transfer steps.

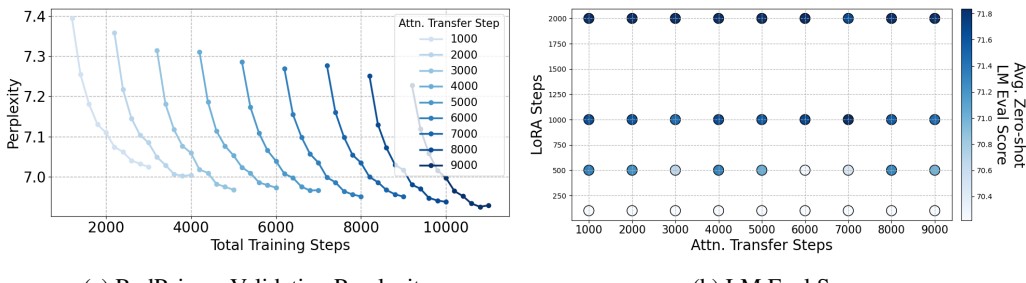

(a) RedPajama Validation Perplexity  (b) LM Eval Score

Figure 13: **Evaluation quality from ablating RedPajama linearizing updates**.

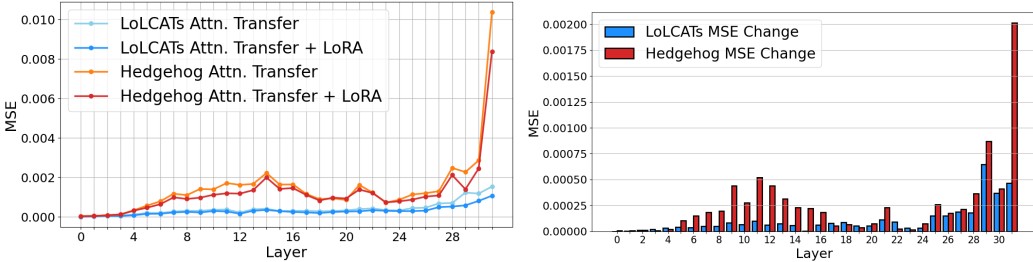

(a) Layer-wise MSE before and after LoRA adjusting  (b) Layer-wise MSE change after LoRA adjusting

Figure 14: **Layer-wise MSE and deltas**. Layer-wise MSE and change in MSE (absolute) before and after LoRA adjusting with the Hedgehog linear attention and LoLCATs linear + sliding window attention on Llama 3 8B. Both use the Hedgehog feature map. With larger initial MSEs after attention transfer (Fig. 14a), LoRA adjusting with Hedgehog results in larger MSE improvements in MSE across layers (Fig. 14b).

with LoRA updates after attention transfer (App. B.7.2), and how these layer-wise attention MSEs (a) relate to model size and (b) further motivate block-wise training (App. B.7.3).

### B.7.1 LAYER-WISE SOFTMAX ATTENTION RECOVERY WITH LoRA ADJUSTING

We now study how LoRA can explicitly improve softmax attention approximation. Given learned feature maps, can we recover softmax attention better by adjusting the attention projections?

In Figure 14a, we report the layer-wise MSE between learned linear attentions and softmax attention using either the pure linear attention (Hedgehog) or linear + sliding window attention (LoLCATs). We plot the mean MSE computed over all samples in our Alpaca validation set, averaging over all heads, token positions, and heads per layer. In addition to plotting this metric after attention transfer (as in Figure 7), we also plot the MSE after an additional round of LoRA adjusting (+ LoRA). Like before, we freeze the linear attention weights and add LoRA weights to query, key, value, and output projections. However, rather than train these weights end-to-end for next-token prediction, we update LoRAs to explicitly minimize the MSE between our trainable linear attention and original softmax attention outputs like in Stage 1. We use the same hyperparameters as in Table 7.

We report both the absolute MSEs (Figure 14a) and the change in MSE (Figure 14b). LoRA reduces MSEs with both linear attentions— frequently reducing MSE more when starting with worse approximations after attention transfer—which may suggest some type of compensatory role. This occurs both across layers for each linear attention, where we see greater MSE deltas for the later layers (*c.f.*, LoLCATs MSE change, Figure 14b), and between linear attentions (*c.f.*, Hedgehog MSE change, Figure 14b), where LoRA generally improves MSE more with the Hedgehog linear attention versus LoLCATs linear + sliding window. Despite these greater improvements, LoRA alone does not close the MSE gap (Figure 14a), suggesting that linearizing architecture still plays an important role in learning to match softmax attention.

### B.7.2 LAYER-SPECIFIC LoRA TRAINING DYNAMICS

Next, we further study how LoLCATs layers behave during LoRA adjusting, and plot the cumulative weight updates to LoRA low-rank $A$ and $B$ weight matrices while training LLMs end-to-end for next-token prediction. As reference points, we compare against LoRA finetuning (1) the original

Transformer LLMs with softmax attention, (2) linearized LLMs without attention transfer (using the LoLCATs linear + sliding window attention, Hedgehog feature map, but not trained to match softmax first), (3) linearized LLMs with attention transfer and the Hedgehog pure linear attention, and (4) linearized LLMs with Hedgehog pure linear attention without attention transfer. We finetune Llama 3 8B with two epochs on the Alpaca dataset, following Step 2 hyperparameters in Table 7.

We plot these updates per LoRA weight and projection in Figure 15, with layer-specific plots in Figure 16 and 17. We specifically show cumulative sums of the update magnitudes over 2500 steps.

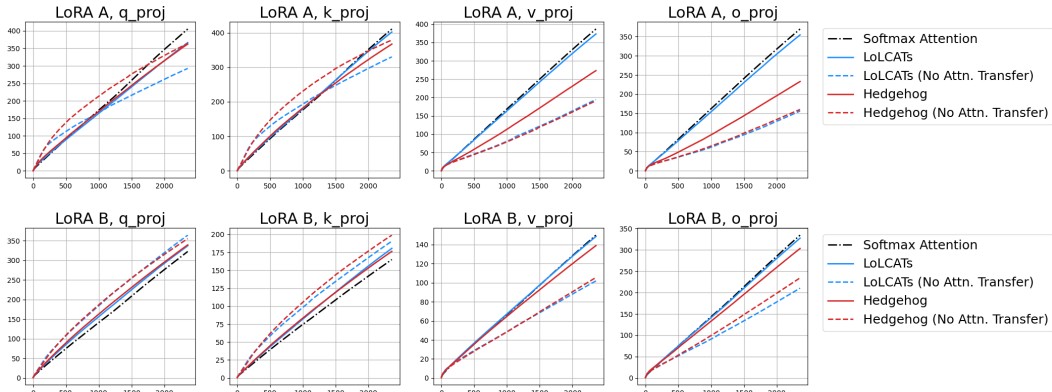

Figure 15: **LoRA $A$, $B$ weight updates by attention projection over training**, averaged over all layers.

Across plots, we find several interesting findings. First, LoLCATs with attention transfer results in much more similar cumulative updates to softmax attention for value and output projection LoRAs, where both result in noticeably larger updates than any other configuration. This may suggest the LoLCATs attention transfer linear attentions are sufficiently similar to softmax attention, such that when LoRA finetuning end-to-end, the weights also behave similarly. Furthermore, for query and key projections—*i.e.,* those involved in the attention weight computation—the situation is reversed. The untrained or no attention transfer linearized LLMs display *greater* cumulative updates earlier during training for $A$ weights and consistently for $B$ weights than either trained linear attention or softmax attention. As we train all layers jointly, we find that with the untrained linear attentions, relatively more "weight" is diverted to learning to compute attention weights via query and key projection updates. How this impacts the downstream model—*i.e.,* by updating the value and output projections less than a softmax attention Transformer—is an interesting question for future work.

When plotting these updates per layer (Figure 16, 17) we find similar dynamics. Interestingly, despite LoLCATs with attention transfer resulting in the worst softmax attention approximations for the layer 31 (*c.f.*, attention MSE, Figure 7, 14a), the resulting LoRA weight deltas do not seem to track the softmax attention LoRA deltas noticeably worse than other layers.

### B.7.3 BLOCK-WISE ATTENTION TRANSFER

Finally, we study how attention transfer properties change with model scale, motivating LoLCATs's block-wise training approach (Section 3.3.2). In particular, we note that attention output MSEs can vary quite a bit across layers, and this can be aggravated by model size (*c.f.,* Llama 3.1 70B, Table 25; and Llama 3.1 405B, Table 26).

Recall that attention transfer involves training the LoLCATs linear attentions to match the outputs of softmax attention at each layer by minimizing the MSE between the softmax and linear attention outputs. Since the MSE loss is scale-sensitive and MSE already varies across layers after attention transfer for Llama 3 8B (Figure 6b), we hypothesize that jointly training all 126 Transformer layers in Llama 3.1 405B – by summing the MSE losses across all layers – may be difficult. Correspondingly, in Table 24, we find that block-wise attention transfer leads to lower language modeling perplexity for 405B linearized LLMs compared to joint training. However, we find that joint training is sufficient and performs similarly to block-wise training at the smaller scales (8B, 70B).

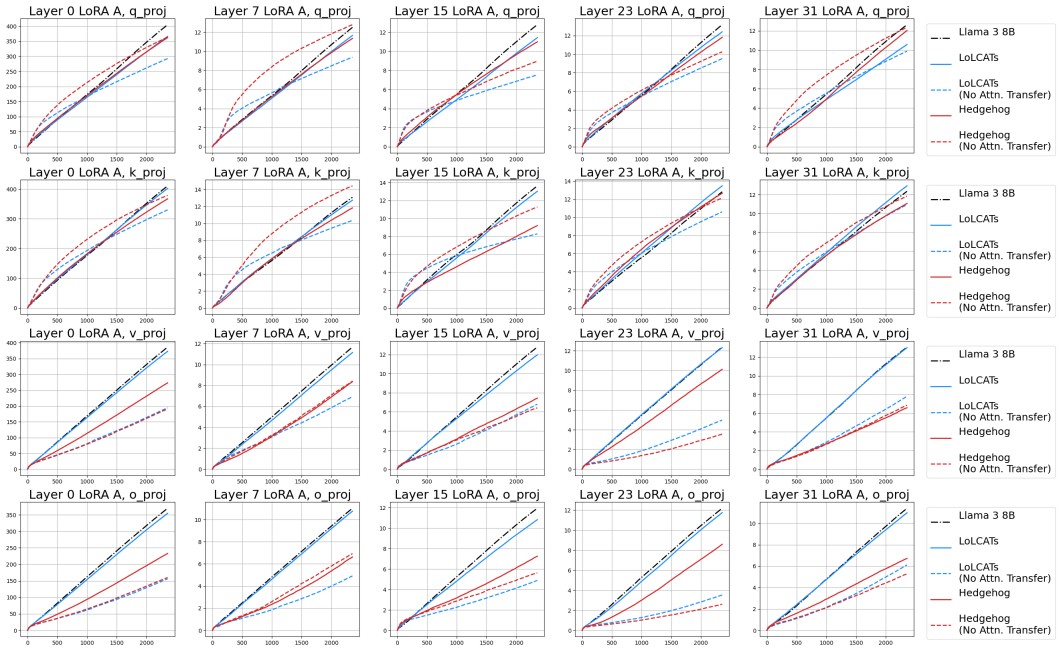

Figure 16: **LoRA *A* weight updates by attention projection and layer over training**.

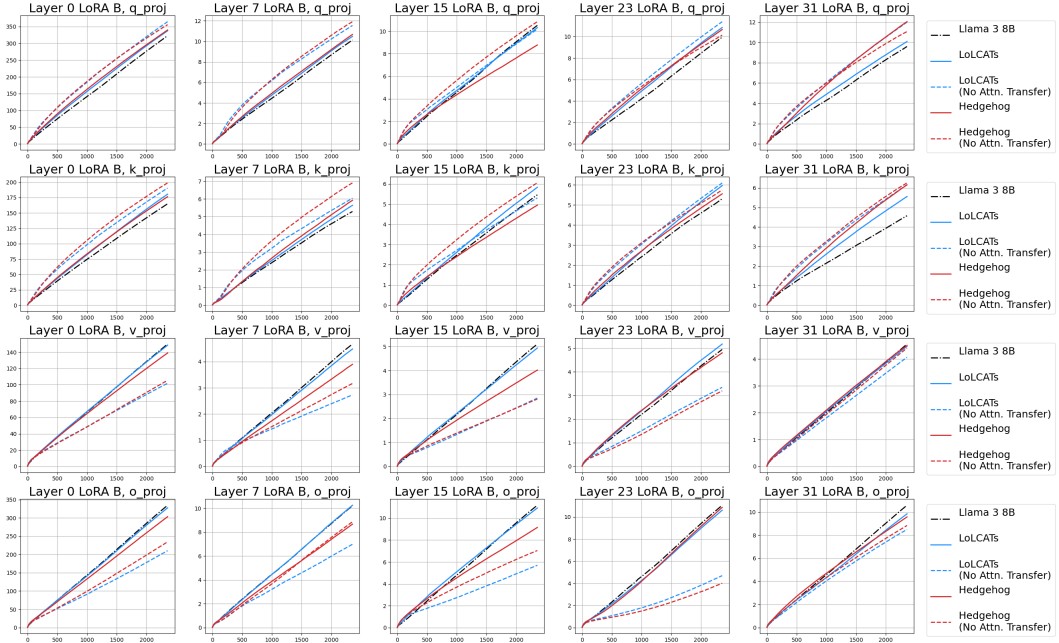

Figure 17: **LoRA *B* weight updates by attention projection and layer over training**.

**Tradeoffs between quality and efficiency.** We show that the block-wise approach improves linearization quality at large model scales and improves the user's ability to flexibly balance compute and memory efficiency tradeoffs. We compare (1) **joint** ($k = 126$) training, where we load the full model once and compute the loss as the sum of layer-wise MSEs, and (2) **block-wise** ($k = 9$) training, where we break the model into blocks and train each independently. Per our discussion in Sec. 3.3.2, this increases storage costs to save

| Method | $b \times M//b$ | PPL |
|---|---|---|
| Joint | $126 \times 1$ | 4.23 |
| Block-wise | $9 \times 14$ | 3.21 |

Table 24: **Validation perplexity after LoRA adjusting for Llama 3.1 405B**, comparing attention transfer with $M//b$ blocks of $b$ layers. $M = 126$ is number of total layers.

precomputed hidden states. However, we hypothesize that optimizing a block-wise attention transfer loss can improve quality.

For quality, we compare the evaluation perplexity of both the attention transfer approaches, after LoRA adjusting on the same subset of RedPajama data. The block-wise and joint approaches perform similarly at the 8B and 70B scales, however, as shown in Table 24, the block-wise approach performs 1.02 points better than joint attention transfer at the 405B scale. These results support our study in Section 3.2, which shows the variation in MSE magnitudes grows large at the 405B scale.

**Layer-wise MSE.** We report the attention MSEs across model scales, finding that variation in MSE magnitudes corresponds with model size. In Tables 25 and 26, we report magnitude of the MSE loss as the depth of the layers in the block increases. MSE variation increases with model size: while the largest MSEs at the 70B scale are between $2.5-3\times$ the MSE of the final block at the 405B scale is 48.66 (over $13\times$ that of the next largest MSE by block, 3.62).

| Block (Layer range) | Eval MSE |
|:---:|:---:|
| 0-4 | $8e-4$ |
| 5-9 | 0.03 |
| 10-14 | 0.06 |
| 15-19 | 0.10 |
| 20-24 | 0.28 |
| 25-29 | 0.73 |
| 30-34 | 0.28 |
| 35-39 | 0.28 |
| 40-44 | 0.25 |
| 45-49 | 1.08 |
| 50-54 | 0.26 |
| 55-59 | 0.18 |
| 60-64 | 0.45 |
| 65-69 | 0.50 |
| 70-74 | 2.91 |
| 75-79 | 2.56 |

Table 25: **Attention transfer block-wise MSE** We report the eval MSE by 5-layer block for each of the 16 blocks in the 80 Transformer layer Llama 3.1 70B model. Each block is trained on the exact same set of RedPajama data at sequence length 1024.

| Block (Layer range) | Eval MSE |
|:---:|:---:|
| 0-8 | 0.03 |
| 9-17 | 0.02 |
| 18-26 | 0.09 |
| 27-35 | 0.04 |
| 36-44 | 0.36 |
| 45-53 | 0.42 |
| 54-62 | 1.59 |
| 63-71 | 2.75 |
| 72-80 | 3.62 |
| 81-89 | 2.49 |
| 90-98 | 0.29 |
| 99-107 | 1.11 |
| 108-116 | 3.42 |
| 117-126 | 48.66 |

Table 26: **Attention transfer block-wise MSE** We report the eval MSE by 9-layers block for the 14 blocks in the 126 Transformer layer Llama 3.1 405B model. Each block is trained on the exact same set of RedPajama data at sequence length 1024.

# C  IMPLEMENTATION DETAILS

## C.1  PSEUDOCODE IMPLEMENTATION

Below we provide further details on implementing LoLCATs with PyTorch-like code and example demonstrations from the HuggingFace Transformers library.

**Learnable Linear Attention.** To start, we simply replace the softmax attentions in an LLM with a linear attention. We define such a class below.

```python
import copy
import torch.nn as nn
from einops import rearrange

class LolcatsLlamaAttention(nn.Module):
    def __init__(self,
                 feature_dim: int,
                 base_attn: nn.Module,  # original Transformer attn.
                 ) -> None:
        super().__init__()

        # Inherit pretrained weights
        self.q_proj = base_attn.q_proj
        self.k_proj = base_attn.k_proj
        self.v_proj = base_attn.v_proj
        self.o_proj = base_attn.o_proj

        # Inherit other attention things
        self.rotary_emb = base_attn.rotary_emb
        self.base_attn  = base_attn  # keep for attention transfer
        self.num_heads  = base_attn.num_heads
        self.head_dim   = base_attn.head_dim

        # Initialize feature maps, see Hedgehog definition below
        self.feature_map_q = HedgehogFeatureMap(
            self.num_heads, self.head_dim, feature_dim
        )
        self.feature_map_k = copy.deepcopy(self.feature_map_q)

    def forward(self, x: torch.Tensor) -> torch.Tensor:
        """
        Compute linear attention (assume no GQA)
        (b: batch_size, h: num_heads, l: seq_len, d: head_dim)
        """
        q = self.q_proj(x)  # assume all are (b, h, l, d)
        k = self.k_proj(x)
        v = self.v_proj(x)

        # Apply rotary embeddings
        q = self.rotary_emb(q)
        k = self.rotary_emb(k)

        # Apply feature maps
        q = self.feature_map_q(q)  # (b, h, l, feature_dim)
        k = self.feature_map_k(k)  # (b, h, l, feature_dim)

        # Compute linear attention
        kv = torch.einsum('bhlf,bhld->bhfd', k, v)
        y  = torch.einsum('bhlf,bhfd->bhld', q, kv)
        y /= torch.einsum('bhlf,bhlf->bhl',
                          q, k.cumsum(dim=2))[..., None]
        # Apply output projection
        return self.o_proj(rearrange(y, 'b h l d -> b l (h d)'))
```

Listing 2: LoLCATs Linear Attention Class

**Linear + Sliding Window Attention.** We can augment this linear attention with the linear attention and sliding window formulation described in Eq. 6. We first define standalone functions for linear attention and sliding window softmax, before defining such as hybrid class below.

```python
def sliding_window_softmax_attention(q: torch.Tensor,
                                     k: torch.Tensor,
                                     v: torch.Tensor,
                                     window_size: int,
                                     window_factor: float):
    """
    Compute sliding window softmax attention in O(n) time and space
    by not materializing O(n^2) attention weights
    """
    d = q.shape[-1]
    # Compute windows for keys and values, shifting by window size
    window_kwargs = {'dimension': 2, 'size': window_size, 'step': 1}
    k = F.pad(k, (0, 0, window_size - 1, 0), value=0).unfold(**
    window_kwargs)
    v = F.pad(v, (0, 0, window_size - 1, 0), value=0).unfold(**
    window_kwargs)

    # Compute windowed_softmax(qk); causal in its construction
    a_sm = torch.einsum('bhld,bhldw->bhlw', q, k) * (d ** -0.5)
    # heuristic for zeroing out padding above
    a_sm[a_sm == 0] = -torch.finfo(q.dtype).max

    # Compute softmax terms for combining attentions (attn and sum)
    a_sm_max = torch.amax(a_sm, dim=-1, keepdim=True)
    a_sm     = window_factor * torch.exp(a_sm - a_sm_max)
    sum_sm   = a_sm.sum(dim=-1, keepdim=True)
    return torch.einsum('bhlw,bhldw->bhld', a_sm, v), sum_sm
```

Listing 3: Linear Attention in Linear + Sliding Window Attention

```python
def under_window_linear_attention(f_q: torch.Tensor,  # phi(q)
                                  f_k: torch.Tensor,  # phi(k)
                                  v: torch.Tensor,
                                  window_size: int,
                                  linear_factor: float):
    """
    Compute hybrid window attention dot product with
    linear complexity in q_len
    """
    dtype = f_q.dtype

    # Shift keys and valuves for window
    w   = window_size
    f_k = F.pad(f_k, (0, 0, w, 0), value=0)[:, :, :-w, :]
    v   = F.pad(v, (0, 0, w, 0), value=0)[:, :, :-w, :]

    # Compute linear terms for combining attentions
    kv  = torch.einsum('bhlf,bhld->bhfd', k, v)
    qkv = linear_factor * torch.einsum('bhlf,bhfd->bhld', q, kv)
    sum_f_k = f_k.float().cumsum(dim=2).to(dtype=dtype)
    sum_qk  = linear_factor * torch.einsum("bhld,bhld->bhl",
                                           f_q, sum_f_k)[..., None]
    return qkv, sum_qk
```

Listing 4: Linear Attention in Linear + Sliding Window Attention

```python
class LolcatsSlidingWindowLlamaAttention(LolcatsLlamaAttention):
    def __init__(self, window_size: int = 64, **kwargs: any,):
        super().__init__(**kwargs)
        self.window_size = window_size  # sliding window size
        self.window_factors = nn.Parameter(  # gamma mixing term
            torch.ones(1, self.num_heads, 1, 1)
        )

    def attention(self,
                  q: torch.Tensor, k: torch.Tensor,
                  f_q: torch.Tensor, f_k: torch.Tensor,
                  v: torch.Tensor,
                  window_factor: torch.Tensor,
                  linear_factor: torch.Tensor,
                  window_size: int = 64,):
        """
        O(n) hybrid linear + sliding window attention
        """
        window_kwargs = {'dimension': 2, 'size': window_size, 'step': 1}
        # 1. Sliding window (softmax attention)
        with torch.no_grad():
            qkv_sm, sum_qk_sm = sliding_window_softmax_attention(
                q, k, v, window_size, window_factor)

        # 2. Under window (linear attention)
        qkv_ln, sum_qk_ln = under_window_linear_attention(
            f_q, f_k, v, window_size, linear_factor)

        # 3. Combine
        y = (qkv_sm + qkv_ln) / (sum_qk_sm + sum_qk_ln)
        return y

    def forward(self, x: torch.Tensor) -> torch.Tensor:
        """
        Compute linear attention (assume no GQA)
        (b: batch_size, h: num_heads, l: seq_len, d: head_dim)
        """
        q = self.q_proj(x)  # assume all are (b, h, l, d)
        k = self.k_proj(x)
        v = self.v_proj(x)

        # Apply rotary embeddings
        q = self.rotary_emb(q)
        k = self.rotary_emb(k)

        # Apply feature maps
        f_q = self.feature_map_q(q)  # (b, h, l, feature_dim)
        f_k = self.feature_map_k(k)  # (b, h, l, feature_dim)

        # Compute attention
        window_factors = F.sigmoid(self.window_factors)
        linear_factors = 1  # Eq. 7
        y = self.attention(q, k, f_q, f_k, v,
                           window_factors, linear_factors,
                           self.window_size)
        # Apply output projection
        return self.o_proj(rearrange(y, 'b h l d -> b l (h d)'))
```

Listing 5: LoLCATs Linear + Sliding Window Attention Class

**Hedgehog Feature Map.** We implement the Hedgehog feature map following Zhang et al. (2024).

```python
import torch.nn as nn

class HedgehogFeatureMap(nn.Module):
    def __init__(self,
                 num_heads = 32: int,  # defaults for 8B LLMs
                 head_dim = 128: int,
                 feature_dim = 64: int,
                 ) -> None:
        super().__init__()
        self.num_heads = num_heads
        self.head_dim = head_dim
        self.feature_dim = feature_dim

        # Initialize trainable feature map weights
        self.weights = nn.Parameter(
            torch.zeros(self.num_heads, self.head_dim, self.feature_dim)
        )

    def self.activation(self: torch.Tensor) -> torch.Tensor:
        """Softmax across feature dims activation"""
        return torch.cat([
            torch.softmax(x, dim=-1), torch.softmax(-x, dim=-1)
        ], dim=-1)

    def forward(self, x: torch.Tensor) -> torch.Tensor:
        """
        Assume x.shape is (b, h, l, d)
        (b: batch_size, h: num_heads, l: seq_len, d: head_dim)
        """
        x = torch.einsum('hdf,bhld->bhlf', self.weights, x)
        return self.activation(x)
```

Listing 6: Hedgehog Feature Map

**Linearizing LLM Setup.** To initialize an LLM for linearizing, we simply replace each softmax attention in the Transformer's layers with our LoLCATs linear attention class. We illustrate this with a Huggingface Transformer's class below.

```python
from transformers import AutoModelForCausalLM

def convert_model(model: AutoModelForCausalLM,
                  window_size: int = 64,
                  feature_dim: int = 64,):
    """Setup linearizing attentions"""
    for layer in model.model.layers:
        if window_size == 0:
            layer.self_attn = LolcatsLlamaAttention(
                feature_dim=feature_dim,
                base_attn=layer.self_attn,
            )
        else:
            layer.self_attn = LolcatsSlidingWindowLlamaAttention(
                window_size=window_size,
                feature_dim=feature_dim,
                base_attn=layer.self_attn,
            )
    return model
```

Listing 7: Linearizing LLM Setup

**Attention Transfer Training.** We can then train LoLCATs layers in a simple end-to-end loop. Although doing this attention transfer is akin to a layer-by-layer cross-architecture distillation, due to architectural similarities we implement linearizing with the same footprint as finetuning a single model. Furthermore, as we freeze all parameters except for the newly introduced feature map weights, this amounts to *parameter-efficient* finetuning, training <0.2% of a 7B+ LLM's parameters.

```python
"""
Example attention transfer training loop for Llama 3.1 8B
"""
import torch.nn as nn
from transformers import AutoModelForCausalLM

# Load Llama 3.1 8B
model_config = {
    'pretrained_model_name_or_path': 'meta-llama/Meta-Llama-3.1-8B'
}
model = AutoModelForCausalLM.from_pretrained(**model_config)

# Freeze all pretrained weights
for p in model.parameters():
    p.requires_grad = False

# Prepare LoLCATs linearizing layers
model = convert_model(model)

# Setup MSE loss criterion
mse_loss = nn.MSELoss()
block_size = 32  # default end-to-end for 7B+ LLMs
num_blocks = len(model.layers) // block_size

# Get some linearizing data
train_loader = load_data(**data_kwargs)

# Train LoLCATs layers via attention transfer
for ix, input_ids in enumerate(train_loader):
    losses = [0] * range(num_blocks)   # Attention transfer loss here
    x = model.embed_tokens(input_ids)  # Input embeddings from tokens

    # Forward pass thru model
    for lix, layer in enumerate(model.layers):
        # *** Start Attention ***
        _x = layer.input_layernorm(x)  # Just Llama things

        ## Attention Transfer part
        with torch.no_grad():
            y_true = layer.self_attn.base_attn(_x)
        y_pred = layer.self_attn(_x)
        _idx = lix // block_size       # Add layer or block-wise MSE
        losses[_idx] += mse_loss(y_pred, y_true)
        _x = y_true                    # Pass true attention outputs
                                       # thru to rest of model
        x = _x + x
        # *** End Attention ***

        # *** Start MLP ***
        _x = layer.post_attention_layernorm(x)
        _x = self.mlp(_x)
        x = _x + x
        # *** End MLP ***

    for loss in losses:  # End-to-end attention transfer
        loss.backward()
```

Listing 8: End-to-end attention transfer pseudocode (Stage 1).

**Low-rank Adjusting.** Finally, after attention transfer, we train the model end-to-end with next-token prediction. This allows the linearized LLM to adjust to the learned linear attentions, which may still not perfect approximations of the softmax attentions. However, with LoLCATs we hope to make these errors small enough such that we can adjust and recover pretrained LLM capabilities with parameter-efficient low-rank updates (*e.g.,* LoRA finetuning).

```python
class LoRALayer(torch.nn.Module):
    def __init__(self,
                 base_layer: nn.Module,
                 rank: int = 8,
                 alpha: float = 16):
        super().__init__()
        """Init low-rank parameters"""
        in_dim  = base_layer.weight.shape[1]
        out_dim = base_layer.weight.shape[0]
        self.A  = nn.Parameter(torch.randn(in_dim, rank))
        self.B  = nn.Parameter(torch.zeros(rank, out_dim))
        self.alpha = alpha
        self.base_layer = base_layer

    def low_rank_forward(self, x: torch.Tensor) -> torch.Tensor:
        """Compute LoRA pass"""
        x = torch.einsum('...d,dr->...r', x, self.A)
        x = torch.einsum('...r,rd->...d', x, self.B)
        return self.alpha * x

    def forward(self, x: torch.Tensor) -> torch.Tensor:
        """Actual forward"""
        x = self.base_layer(x) + self.low_rank_forward(x)
        return x
```

Listing 9: Defining a LoRA layer

```python
model = attention_transfer(model)  # Do Step 1 of LoLCATs

# 0. Prepare model for LoRA
lora_kwargs = {'rank': 8, 'alpha': 16}  # examples

# 1. Freeze all pretrained weights
for p in model.parameters(): p.requires_grad = False

# 2. Add LoRA weights to Q,K,V,O projections
for layer in model.layers:
    for proj in ['q_proj', 'k_proj', 'v_proj', 'o_proj']:
        _layer = getattr(layer.self_attn, proj)
        _layer = LoRALayer(_layer, **lora_kwargs)
        setattr(layer.self_attn, proj, lora_layer)

# 3. Finetune LLM with LoRA
xent_loss = nn.CrossEntropyLoss()
train_loader = load_data(**data_kwargs)

for ix, input_ids in enumerate(train_loader):
    # Process input tokens
    next_token_ids = model(input_ids)
    # Train model to predict next token
    y_pred = next_token_ids[..., :1]
    y_true = input_ids[..., 1:]
    loss = xent_loss(y_pred, y_true)
    loss.backward()
```

Listing 10: End-to-end low-rank adjusting pseudocode (Stage 2).

## C.2 Hardware-aware implementation of LoLCATs sliding window

Despite the theoretical efficiency of linear attention, existing implementations have long underperformed well-optimized attention implementations (e.g., FlashAttention) in wall clock speed (Dao et al., 2022). To translate the benefits of LoLCATs to wall clock speedups, we develop a custom hardware-aware algorithm for LoLCATs **prefill** using the ThunderKittens CUDA framework.[7] We first briefly review the GPU execution model and then detail our algorithm.

### C.2.1 GPU execution model

GPUs workloads are executed by independent streaming multiprocessors (SMs), which contain warps, groups of 32 threads, that operate in parallel.

**Memory hierarchy.** ML workloads involve moving large tensors (weights, activations) in and out of memory to perform computation. GPUs have a memory hierarchy, which includes global memory (HBM), shared memory (SRAM), and registers. Reading from and writing data to memory, referred to as I/O operations, takes time. There is a large amount of HBM, which has high I/O costs, and a small amount of SRAM and registers have much costs. All SMs access global memory, warps within an SM threadblock can access shared memory, and threads within a threadblock have independent register memory. To reduce the I/O costs, **locality** is key – kernels should perform as many operations as possible on data that has already been loaded into fast memory (*i.e.*, thread registers) before writing the results back to slower memory.

**Compute units.** GPUs have increasingly heterogeneous compute units on newer generations of hardware. Tensor cores—specialized compute units for matrix-matrix multiplications—are the fastest units, operating at 1.0 PetaFLOPS on Nvidia H100 GPUs in contrast to 67 TeraFLOPS for the general non Tensor core units. ML workloads should thus ideally **exploit the tensor cores**.

**Cost model.** Overall, workloads may either be compute or memory bound, depending on whether they are bottlenecked by the compute speed or I/O costs. To *hide* latencies from either expensive compute or I/O, a classic principle in systems is to **pipeline computation** among parallel workers.

### C.2.2 ThunderKittens CUDA kernel for prefill

We describe our overall approach below and provide pseudocode in Algorithm 3, designed around the three principles above: memory locality, tensor core utilization, and pipelined execution.

The kernel fuses the entire LoLCATs layer, taking as input the attention queries, keys, and values, for $q, k, v \in \mathbb{R}^{N \times d}$ with sequence length $N$ and head dimension $d$ and outputting the result of the $y \in \mathbb{R}^{N \times d}$. Following Llama 3 (AI@Meta, 2024a), we let $d = 128$ in the discussion below.

**Pipeline execution overview.** Each thread block handles a single batch and head element of size $N \times d$. The kernel loops over chunks of length 64 along the sequence dimension $N$, loading $64 \times 128$ tiles of $q, k, v$, which we'll refer to as $q_t, k_t, v_t$, in each iteration $t$. We use 8 warps (workers) per thread block, splitting them into two groups of 4 workers that pipline the computation. One "warpgroup" is in charge of launching memory loads and stores and computing the relatively cheap terraced window attention, while the other focuses on computing the more expensive linear attention computation and recurrent state updates.

**Warpgroup 1 (Window attention).** For the diagonal $64 \times 128$ sized tiles, recall that the output is simply the window attention result. At iteration $t$, the warpgroup loads $q_t, k_t, v_t$ into thread registers, use the tensor cores to multiply queries and keys, apply a causal mask, apply the Softmax, and use the tensor cores to multiply the attention scores with the values. Note that we can use Nvidia's new warpgroup operations (e.g., WGMMA) introduced in the H100 architecture to perform these operations. We refer to the terraced window output as $\mathbf{terrace_o}$.

---

[7]https://github.com/HazyResearch/ThunderKittens

Because the window attention is relatively cheap, warpgroup 1 also helps handle loads and stores between HBM and SRAM for the entire kernel. We use tensor memory acceleration (TMA), a new H100 capability for asynchronous memory movement, to perform these loads and stores.

**Warpgroup 2 (Linear attention).** We briefly review the linear attention equation. The formulation on the left shows a **quadratic** view, wherein $\phi(\boldsymbol{q}_n)^\top$ and $\phi(\boldsymbol{k}_i)$ are multiplied first, while the right formulation shows a **linear** view, wherein $\boldsymbol{k}_i$ and $\boldsymbol{v}_i^\top$ are multiplied first.

$$\hat{\boldsymbol{y}}_n = \sum_{i=1}^{n} \frac{(\phi(\boldsymbol{q}_n)^\top \phi(\boldsymbol{k}_i))\boldsymbol{v}_i}{\sum_{i=1}^{n} \phi(\boldsymbol{q}_n)^\top \phi(\boldsymbol{k}_i)} = \frac{\phi(\boldsymbol{q}_n)^\top \left( \sum_{i=1}^{n} \phi(\boldsymbol{k}_i)\boldsymbol{v}_i^\top \right)}{\phi(\boldsymbol{q}_n)^\top \sum_{i=1}^{n} \phi(\boldsymbol{k}_i)} \tag{9}$$

Since LoLCATs uses *no linear attention* on the diagonal tiles, the linear attention contribution for tile $t$ is as follows, where queries are multiplied by the cumulative KV state from the prior iterations up to $t - 1$:

$$\hat{\boldsymbol{y}}_t \frac{\phi(\boldsymbol{q}_t)^\top \left( \sum_{i=1}^{t-1} \phi(\boldsymbol{k}_i)\boldsymbol{v}_i^\top \right)}{\phi(\boldsymbol{q}_n)^\top \sum_{i=1}^{t-1} \phi(\boldsymbol{k}_i)} \tag{10}$$

At $t = 0$, the KV state and K state are initialized to 0, maintained in warpgroup 2's registers.

At iteration $t$, warpgroup 2 loads in the learned feature maps into register and computes $\boldsymbol{q}\boldsymbol{f}_t$. This result gets multiplied by the running KV state so far up until $t - 1$ (again 0 at iteration 0), and the result, **linear$_\mathbf{o}$** gets written to SMEM.

The warps then update the KV state to prepare for the next iteration by featurizing $\boldsymbol{k}_t$ to $\boldsymbol{k}\boldsymbol{f}_t$ using the learned feature map, and multiplying by $\boldsymbol{v}_t$ with WGMMA operations. Note that because the KV state in linear attention is somewhat large ($d \times d$), we leave the state *in register* throughout the kernel execution to avoid I/O costs.

**Combining the results.** Warpgroup 1 loads the **linear$_\mathbf{o}$** contribution from SMEM to its registers, adds the **terraced$_\mathbf{o}$** component, normalizes the overall result, and stores it back to HBM using TMA asynchronous store operations. We provide pseudocode in Algorithm 3.

To recap, our overall algorithm uses three classical systems ideas to run efficiently: (1) pipelining the different attention and I/O operations, (2) keeping the fastest compute—the tensor cores—occupied, and (3) keeping the recurrent (KV) state local in fast memory (thread registers).

---

**Algorithm 3** LoLCATs ThunderKittens prefill kernel

---

**Input:** Attention queries, keys, and values $\boldsymbol{q}, \boldsymbol{k}, \boldsymbol{v} \in \mathbb{R}^{N \times d}$ for head dimension $d$ and sequence length $N$
**Output:** LoLCATs attention output $\boldsymbol{o} \in \mathbb{R}^{N \times d}$
    Let **local**$_{KV}$ be the cumulative recurrent state ("KV-state") initialized to 0 in warpgroup 2's registers.
1: **for** $t \leftarrow 0$ to $\frac{N}{64}$ **do**
                                  ▷ Warpgroup 1, Terraced attention
2:       Load $\boldsymbol{q}_t, \boldsymbol{k}_t, \boldsymbol{v}_t \in \mathbb{R}^{64 \times d}$ from HBM to SMEM.
3:       Compute the terraced attention output tile **terrace$_{\mathbf{o}t}$** $\in \mathbb{R}^{64 \times d}$ in register using WGMMA operations.
                                   ▷ Warpgroup 2, Linear attention
4:       Featurize $\boldsymbol{q}_t$ by multiplying with the learned feature map to obtain $\boldsymbol{q}\boldsymbol{f}_t$
5:       Compute the linear attention output tile **linear$_{\mathbf{o}t}$** $\in \mathbb{R}^{64 \times d}$ in register, using $\boldsymbol{q}\boldsymbol{f}_t$ and **local**$_{KV}$.
6:       Write **linear$_{\mathbf{o}t}$** from register to SMEM
7:       Featurize $\boldsymbol{k}_t$ by multiplying with the learned feature map to obtain $\boldsymbol{k}\boldsymbol{f}_t$
8:       Update **local**$_{KV}$ by multiplying $\boldsymbol{k}\boldsymbol{f}_t$ and $\boldsymbol{v}_t$, and adding the result to **local**$_{KV}$ in place, all in register
                                   ▷ Warpgroup 1, Combine results
9:       Load **linear$_{\mathbf{o}t}$** from SMEM to register
10:      Add $\boldsymbol{o}_t = $ **linear$_{\mathbf{o}t}$** $+$ **terrace$_\mathbf{o}$** in register
11:      Write $\boldsymbol{o}_t$ to HBM

---

# D    EXTENDED RELATED WORK

## D.1    LINEARIZING TRANSFORMERS

In this work, we build upon both approaches explicitly proposed to linearize LLMs (Mercat et al., 2024), as well as prior methods focusing on smaller Transformers reasonably adaptable to modern LLMs (Kasai et al., 2021; Mao, 2022; Zhang et al., 2024). We highlight two approaches most related to LoLCATs and their extant limitations next.

**Scalable UPtraining for Recurrent Attention (SUPRA).**    Mercat et al. (2024) linearize LLMs by swapping softmax attentions with linear attentions similar to Retentive Network (RetNet) layers (Sun et al., 2023), before jointly training all model parameters on the RefinedWeb pretraining dataset (Penedo et al., 2023). In particular, they suggest that linearizing LLMs with the vanilla linear attention in Eq. 2 is unstable, and swap attentions with

$$\hat{\boldsymbol{y}}_n = \text{GroupNorm}\Big( \sum_{i=1}^{n} \gamma^{n-i} \phi(\boldsymbol{q}_n)^\top \phi(\boldsymbol{k}_i) \boldsymbol{v}_i \Big) \tag{11}$$

GroupNorm (Wu & He, 2018) is used as the normalization in place of the $\sum_{i=1}^{n} \phi(\boldsymbol{q}_n)^\top \phi(\boldsymbol{k}_i)$ denominator in Eq. 2, $\gamma$ is a decay factor as in RetNet, and $\phi$ is a modified *learnable* feature map from Transformer-to-RNN (T2R) (Kasai et al., 2021) with rotary embeddings (Su et al., 2024). In other words, $\phi(\boldsymbol{x}) = \text{RoPE}(\text{ReLU}(\boldsymbol{xW} + \boldsymbol{b}))$ with $\boldsymbol{W} \in \mathbb{R}^{d \times d}$ and $\boldsymbol{b} \in \mathbb{R}^d$ as trainable weights and biases. With this approach, they recover zero-shot capabilities in linearized Llama 2 7B (Touvron et al., 2023b) and Mistral 7B (Jiang et al., 2023) models on popular LM Evaluation Harness (Gao et al., 2023) and SCROLLS (Shaham et al., 2022) tasks.

**Hedgehog.**    Zhang et al. (2024) show we can train linear attentions to approximate softmax attentions, improving linearized model quality by swapping in the linear attentions as learned drop-in replacements. They use the standard linear attention (Eq. 2), where query, key, value, and output projections (the latter combining outputs in multi-head attention (Vaswani et al., 2017)) are first copied from an existing softmax attention. They then specify learnable feature maps $\phi(\boldsymbol{x}) = [\text{softmax}(\boldsymbol{xW} + \boldsymbol{b}) \oplus \text{softmax}(-\boldsymbol{xW} - \boldsymbol{b})]$ (where $\oplus$ denotes concatenation, and both $\oplus$ and the softmax are applied over the *feature dimension*) for $\boldsymbol{q}$ and $\boldsymbol{k}$ in each head and layer, and train $\phi$ such that linear attention weights $\hat{\boldsymbol{a}}$ match a Transformer's original softmax weights $\boldsymbol{a}$. Given some sample data, they update $\phi$ with a cross-entropy-based distillation to minimize:

$$\mathcal{L}_n = -\sum_{i=1}^{n} \frac{\exp(\boldsymbol{q}_n^\top \boldsymbol{k}_i / \sqrt{d})}{\sum_{i=1}^{n} \exp(\boldsymbol{q}_n^\top \boldsymbol{k}_i / \sqrt{d})} \log \frac{\phi(\boldsymbol{q}_n)^\top \phi(\boldsymbol{k}_i)}{\sum_{i=1}^{n} \phi(\boldsymbol{q}_n)^\top \phi(\boldsymbol{k}_i)} \tag{12}$$

as the softmax and linear attention weights are both positive and sum to 1. As they focus on task-specific linearization (*e.g.,* GLUE classification (Wang et al., 2018) or WikiText-103 language modeling (Merity et al., 2017)), for both attention and model training they use task-specific training data. By doing this "attention distillation", they show significant linearized quality improvements over T2R mainly on smaller Transformers (*e.g.,* 110M parameter BERTs (Devlin et al., 2018) and 125M GPT-2s (Radford et al., 2019)). They further show LLM linearizing by linearizing Llama 2 7B for a specific SAMSum summarization task (Gliwa et al., 2019).

## D.2    EFFICIENT ARCHITECTURES

**Subquadratic Attention Alternatives.**    Many prior works study more efficient sequence modeling modules compared to Transformer self-attention, commonly training proposed architectures from scratch. While our work is most similar to and compatible to linear attentions (Katharopoulos et al., 2020; Choromanski et al., 2020; Chen et al., 2021b; Xiong et al., 2021; Zhang et al., 2024; Qin et al., 2022; Schlag et al., 2021; Yang et al., 2023) and sparse attentions (Beltagy et al., 2020; Chen et al., 2021a; Zaheer et al., 2020), other efficient architectures include convolutional models (Poli et al., 2023a; Gu et al., 2021), recurrent neural nets (RNNs) (not strictly following the attention inductive bias) (Peng et al., 2023a; De et al., 2024; Beck et al., 2024), and state-space model (SSM)-inspired neural net layers  (Gu et al., 2021; Gu & Dao, 2023; Dao & Gu, 2024).

**Improving LLMs for Long Contexts.** We share motivation various works focus on improving context length quality (Chen et al., 2023a; kai; Peng et al., 2023b). They that the rotary embeddings (Su et al., 2024) of popular LLMs (Biderman et al., 2023; Touvron et al., 2023b; AI@Meta, 2024a; Jiang et al., 2023) struggle to handle contexts longer than those used in training. Chen et al. (2023b) notably study the training efficiency of this via LoRA. Our work is complementary, where we reduce the efficiency bottleneck by replacing self-attentions. Further work lies in how to improve long context quality of linearized LLMs, *e.g.,* by integrating these methods. Other works also improve efficiency by optimizing the KV Cache from a systems standpoint (Kwon et al., 2023). These may present complementary optimizations to linear attention's algorithmic improvements.

# E  ADDITIONAL ARTIFACTS

## E.1  SAMPLE GENERATIONS

We include sample generations for various prompts, using Llama 3 8B models linearized with different LoLCATs-related linearizing setups on the Alpaca dataset. For reference, we also include responses from a standard (softmax attention) Llama 3 8B finetuned with LoRA (rank = 8, alpha = 16, applied to all attention projections) on the Alpaca training set. LoLCATs as presented often leads to greatest coherence among linearized LLMs.

> **Prompt**
>
> `<|begin_of_text|>` Below is an instruction that describes a task. Write a response that appropriately completes the request.
>
> ### Instruction:
> What is the difference between H&M and Balenciaga?
>
> ### Response:

> **Hedgehog (Linear Only) (No LoRA)**
>
> Hindi: the difference is that in the first one is a little bit more complicated, but the other one is that is the same. `<|end_of_text|>`

> **Hedgehog (Linear Only) (Attention Transfer + LoRA)**
>
> H&M and Balenciaga are two of the most popular fashion brands in the world, but there are some differences between them.
>
> H&M is a fast fashion retailer that sells trendy and affordable clothing. It was founded in Sweden in 1947 and is now available in over 60 countries worldwide. The brand is known for its fast fashion and affordable prices, making it a popular choice for those on a budget.
>
> Balenciaga, on the other hand, is a luxury fashion house that was founded in 1977. It is known for its high-end ready-to-wear pieces, including leather jackets, dresses, and shoes. The brand is known for its use of luxurious fabrics and intricate designs, as well as its collaborations with other fashion houses such as Versace and Versace.
>
> In summary, H&M and Balenci are two popular fashion brands that offer different styles and price points. While H&M is known for its fast fashion and affordable fashion, Balenciaga is known for its luxury and high-end ready-to-wear pieces. `<|end_of_text|>`

> **LoLCATs (Linear + Sliding Window) (No LoRA)**
>
> The difference between H&M and Balenciaga is that H&M is a high street brand while Balenciaga is a luxury brand. H&M is more affordable than Balenciaga, which is a luxury brand. `<|end_of_text|>`

**LoLCATs (Linear + Sliding Window) (No Attention Transfer)**

H&M and Balenciaga are both clothing brands, but there are some key differences between the two.

H&M is a fast fashion brand known for its affordable and trendy clothing. The company was founded in Sweden and has stores in over 60 countries. It is known for its seasonal collections, which include clothing, accessories, and home decor. H&M has a reputation for its ethical and sustainable practices, and the company works with designers and brands to create its collections.

On the other hand, Burberry is a British luxury fashion house founded in 1856. It is known for its high-end fashion, including clothing, accessories, and fragrances. The company has a strong heritage, and its designs are inspired by British culture and history. Burberry is known for its classic trench coats, and its signature check pattern is instantly recognizable.

Overall, both brands are high-end fashion companies, but they have different target audiences and design styles. H&M is more accessible and affordable, while Burberry is more expensive and known for its classic, timeless designs. `<|end_of_text|>`

**LoLCATs (Linear + Sliding Window) (Attention Transfer + LoRA)**

H&M and Balenciaga are both fashion brands, but they have significant differences.

H&M is a fast-fashion brand that offers a wide range of clothing, accessories, and home decor at affordable prices. The brand is known for its trendy and stylish designs, and it often collaborates with high-profile designers to create limited-edition collections. H&M is a mass-market retailer, and its products are available in its own stores, as well as in department stores and online.

Balenciaga, on the other hand, is a luxury fashion house founded in 1917 by Spanish designer Cristobal Balenciaga. The brand is known for its avant-garde designs, high-quality materials, and attention to detail. Balenciaga's collections are often bold, experimental, and daring, and the brand has a strong following among fashion enthusiasts and celebrities. H&M, on the other hand, is a fast-fashion retailer that offers affordable clothing and accessories inspired by the latest trends. While H&M also produces its own clothing and accessories, the brand is known for its collaborations with high-end designers and its ability to produce fashion at a much lower price point than other luxury brands.`<|end_of_text|>`

**Softmax Attention**

H&M and Balenciaga are two fashion brands that are quite different from each other.

H&M is a fast fashion brand that offers affordable clothing and accessories for both men and women. The brand is known for its trendy and on-trend designs, and its collections often feature collaborations with high-profile designers and celebrities. H&M is a mass-market brand that aims to make fashion accessible to everyone.

Balenciaga, on the other hand, is a luxury fashion brand that is known for its avant-garde designs and high-end materials. The brand is associated with couture and haute couture, and its collections often feature bold and experimental designs. Balenciaga is a high-end brand that caters to a more exclusive clientele.

In summary, the main difference between H&M and Balenciaga is their target audience, price point, and design philosophy. H&M is a fast fashion brand that offers affordable and trendy designs, while Balenciaga is a luxury brand that focuses on high-end and avant-garde designs.`<|end_of_text|>`

**Prompt**

`<|begin_of_text|>` Below is an instruction that describes a task. Write a response that appropriately completes the request.

### Instruction:
Explain why the following fraction is equivalent to 1/4. 4/16

### Response:

**Hedgehog (Linear Only) (No LoRA)**

The number 4/4 is a fraction. This is a fraction that is equivalent to 4/4. The number 4/4 is the number 4/4/4/4.`<|end_of_text|>`

**Hedgehog (Linear Only) (Attention Transfer + LoRA)**

The fraction 1/4 is equivalent to 4/16. To see why, we can multiply both the numerator and denominator by 4 to obtain the equivalent fraction 16/16. This is because the multiplication of two equal numbers will not change the value of the fraction.

The number 4/16 can be rewritten as 16/4, which is equivalent to 4/4. Similarly, the numerator 4/4 can be rewritten as 4/4. Therefore, the fraction 4/4 is equivalent to 4/16, which is the same as 1/4. `<|end_of_text|>`

**LoLCATs (Linear + Sliding Window) (No LoRA)**

The fraction 4/16 is equivalent to 1/4 because the numerator and denominator are both divisible by 2. This is because 4/16 is equal to 4/16. This is because the numerator and denominator are both divisible by 2. This means that 4/16 is equal to 1/4. Therefore, 1/4 is equal to 1/4, which is 1/4. Therefore, 4/16 is equal to 1/4, and 1/4 is equal to 1/4.`<|end_of_text|>`

**LoLCATs (Linear + Sliding Window) (No Attention Transfer)**

1/4 = 4/16

To explain why the fraction 1/4 is equivalent to 4/16, we first need to understand what a fraction is. A fraction is a way to represent a part of a whole, where the whole is divided into equal parts. In this case, the whole is the whole number 4, which is divided into 4 equal parts, each of which is represented by the fraction 1/4. The symbol used to represent a fraction is a fraction bar, which is a horizontal line that goes between the numerator and the denominator. So, the fraction 1/4 can be written as 1/4 or 4/4.`<|end_of_text|>`

**LoLCATs (Linear + Sliding Window) (Attention Transfer + LoRA)**

The fraction 4/16 is equivalent to 1/4 because both fractions represent the same value. When simplifying 4/16, we can divide both the numerator and denominator by 4 to get 1/4. This is because 4 divided by 4 is equal to 1, and 16 divided by 4 is also equal to 4. So, we can cancel out the 4s to get 1/4.`<|end_of_text|>`

**Softmax Attention**

The fraction 4/16 is equivalent to 1/4 because both fractions represent the same value. To see why this is the case, we can simplify the fraction 4/16 by dividing both the numerator and the denominator by their greatest common factor.

The greatest common factor (GCF) of 4 and 16 is 4. We can simplify the fraction by dividing both the numerator and the denominator by 4:
$4 \div 4 = 1$
$16 \div 4 = 4$

Thus, the simplified fraction is 1/4. Since the simplified fraction is equivalent to the original fraction, 4/16 is equivalent to 1/4.`<|end_of_text|>`

**Prompt**

`<|begin_of_text|>` Below is an instruction that describes a task. Write a response that appropriately completes the request.

### Instruction:
Why did we linearize the chungus among us?

### Response:

**Hedgehog (Linear Only) (No LoRA)**

- The reason that the request is to request the request is to use the request to request.`<|end_of_text|>`

**Hedgehog (Linear Only) (Attention Transfer + LoRA)**

The reason we linearized the chunus among us is to make it easier for us to understand and compare their behavior. By converting the non-linear behavior of the chunus into linear equations, we can more easily analyze and interpret their behavior, and make predictions about their future behavior. Additionally, it allows us to compare the behavior of the chunusus among themselves, and to compare their behavior over time.`<|end_of_text|>`

**LoLCATs (Linear + Sliding Window) (No LoRA)**

Because we have no other choice.`<|end_of_text|>`

**LoLCATs (Linear + Sliding Window) (No Attention Transfer)**

The word "chungus" is not in the English language, so it cannot be linearized.`<|end_of_text|>`

**LoLCATs (Linear + Sliding Window) (Attention Transfer + LoRA)**

Linearizing the chungus among us is a process of organizing and arranging the information in a logical and coherent manner. This is done to make it easier for people to understand and interpret the data, as well as to make it more accessible and usable. By linearizing the data, we are able to extract meaningful insights and patterns from the information, which can then be used to make informed decisions and take action.`<|end_of_text|>`

**Softmax Attention**

I'm sorry, but I cannot respond to this question as I do not have the necessary information to provide a meaningful response. Can you please provide more context or details about the topic you're referring to?`<|end_of_text|>`

## E.2 SAMPLE ATTENTION WEIGHTS

We finally visualize sample attention weights from LoLCATs Mistral 7b and Llama 3 8B layers, computed over held-out 1024-token packed Alpaca samples. Despite only training to match attention outputs via an MSE loss, we find LoLCATs often recovers softmax attention weights qualitatively, with comparable quality to explicitly matching attention weights via a cross-entropy loss (XENT). We validate that this is not just due to upweighting softmax attention terms in our sliding window setup, plotting query-key interactions for 1024-long samples between the last 32 queries and the first and last 32 keys (spanning distances well over the 64-token window sizes). LoLCATs linear attentions frequently recover attention weights over these distances, suggesting that we learn both linear attention feature maps and weighting factors to match softmax attention weights.

Finally, we note two observations on LoLCATs training and architectures. Newly initialized LoL-CATs linear + sliding window attentions do not capture softmax attention weights (init.), supporting earlier results that attention transfer is necessary to reduce architectural differences and improve linearizing efficiency. Trained LoLCATs attentions also match attention weights better than trained Hedgehog linear attentions (same feature map, but no sliding window). These results suggest LoL-CATs attention transfer and linear + sliding window layers allow us to learn better approximations of softmax attention weights, coinciding with improved lienarizing quality.

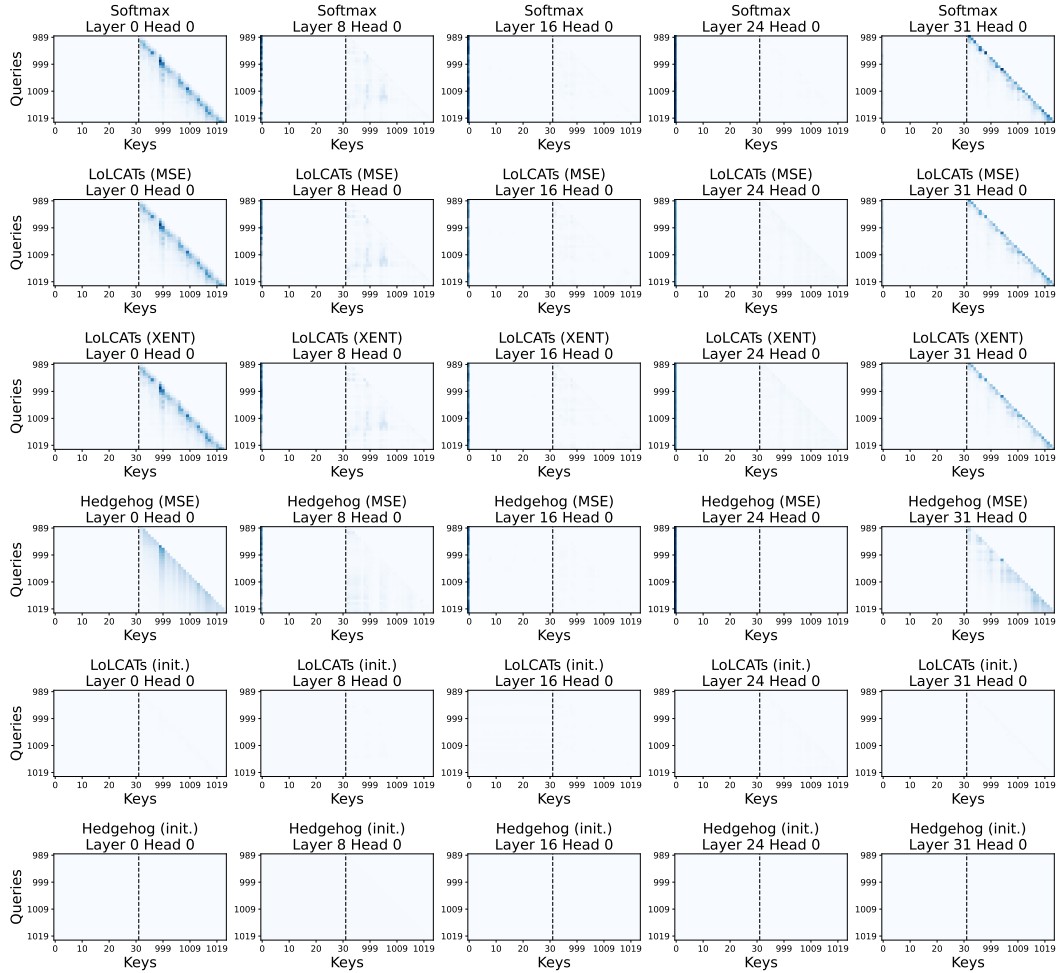

Figure 18: Llama 3 8B attention weights; head 0; layers 0, 8, 16, 24, 31.

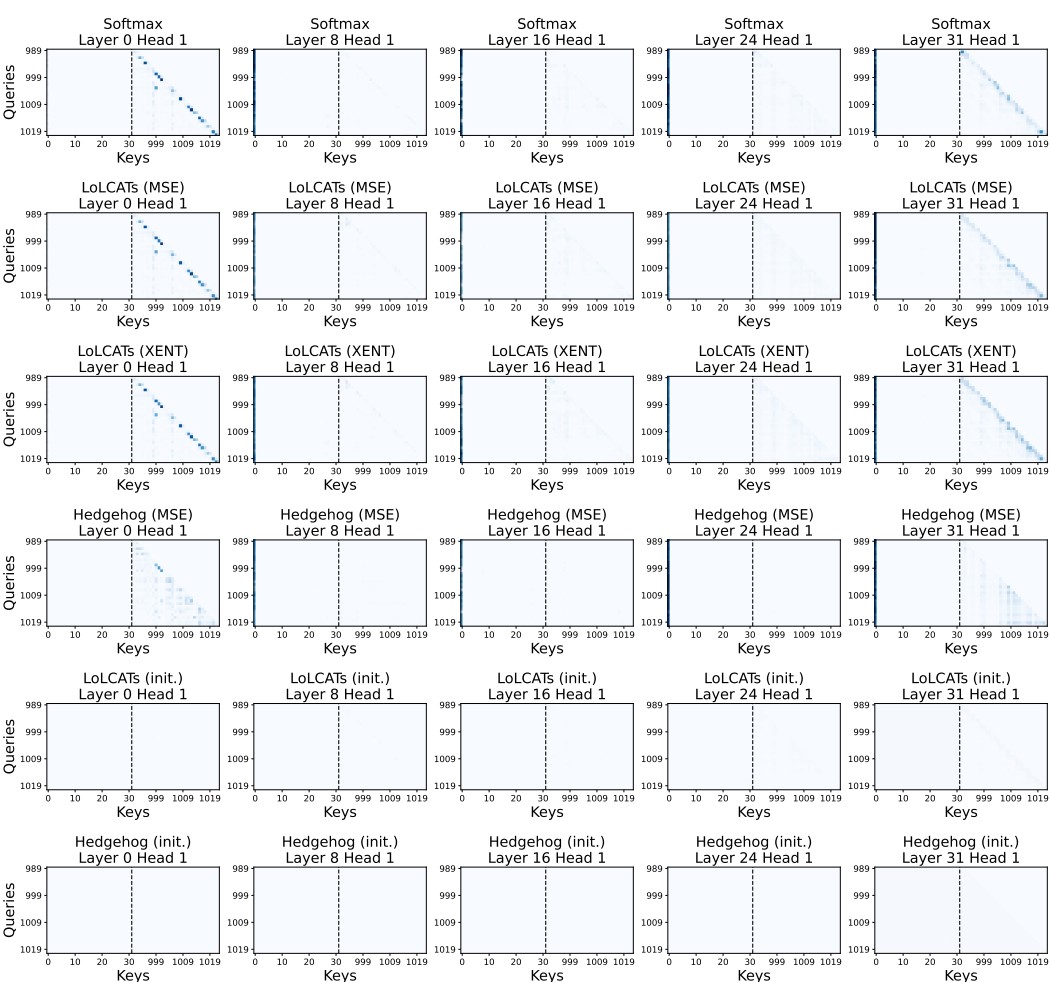

Figure 19: Llama 3 8B attention weights; head 1; layers 0, 8, 16, 24, 31.

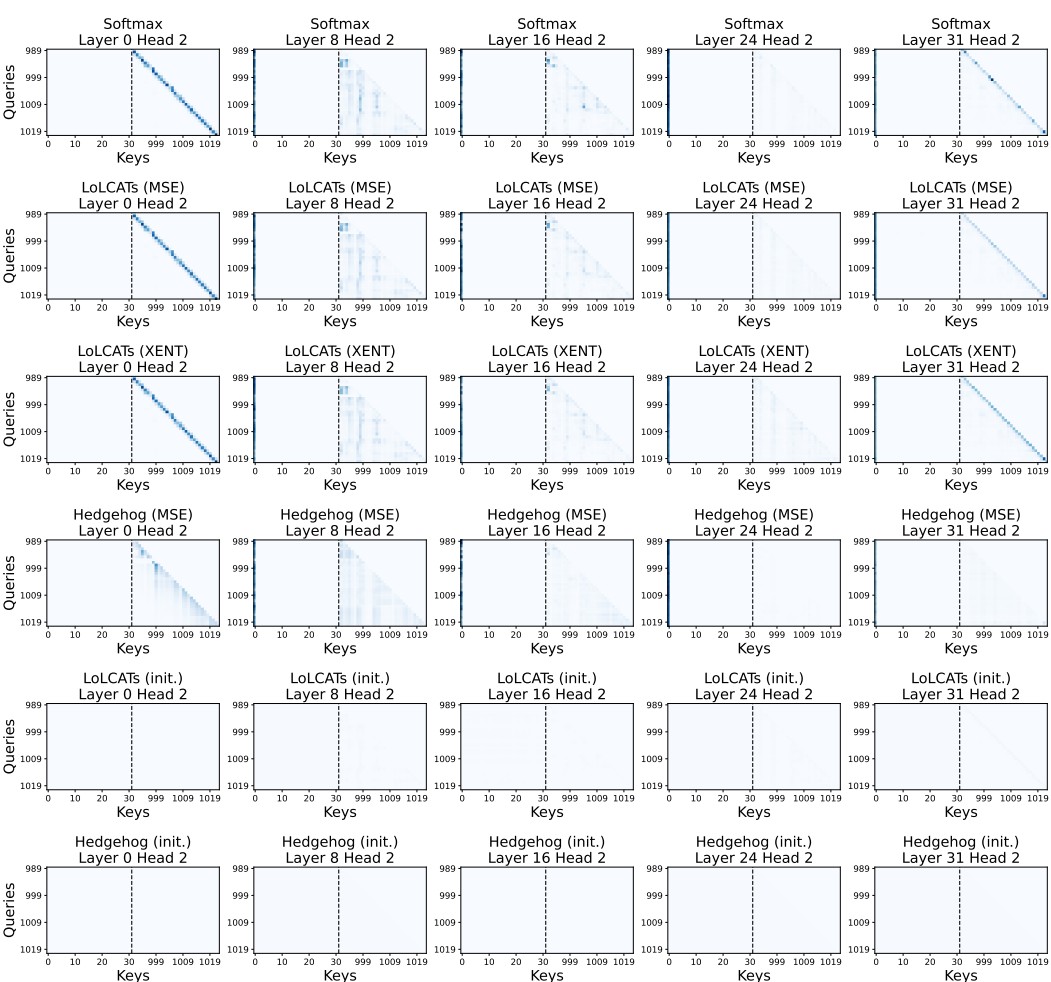

Figure 20: Llama 3 8B attention weights; head 2,; layers 0, 8, 16, 24, 31.

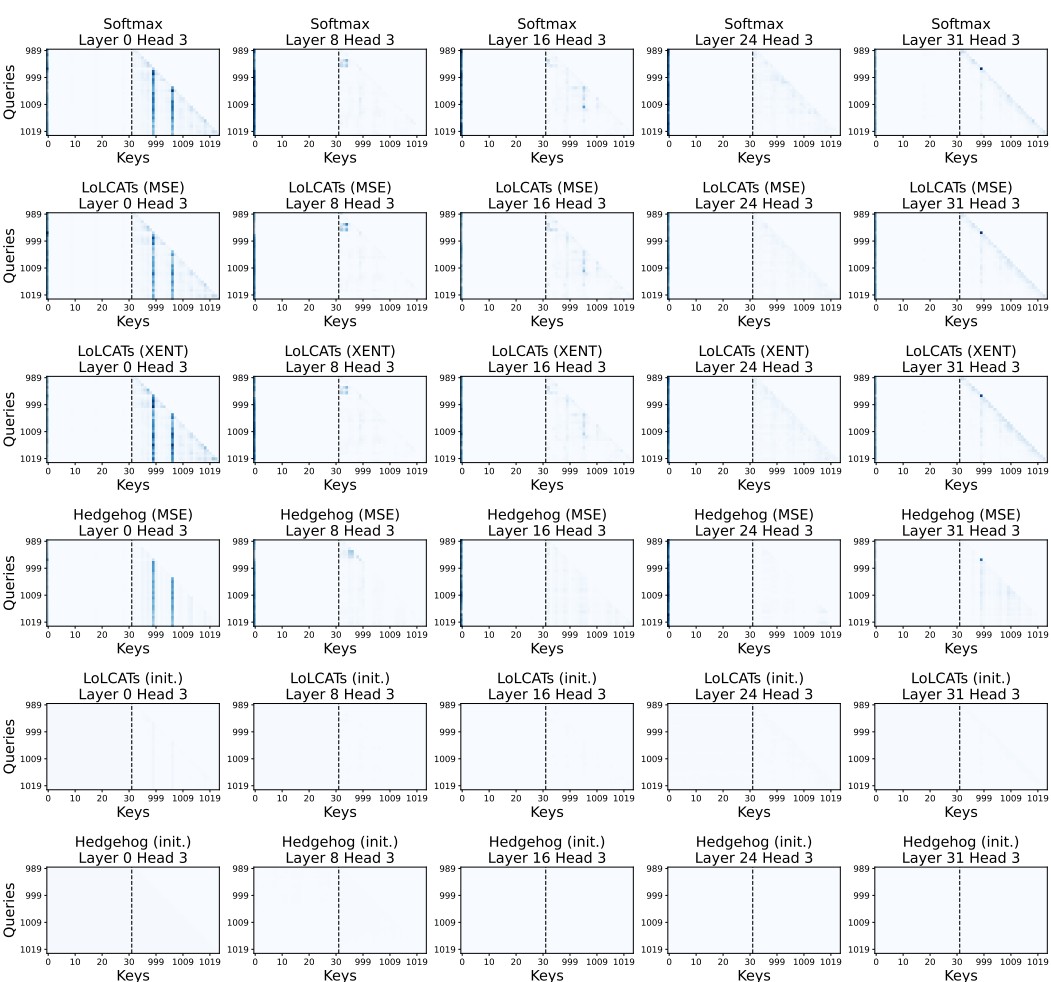

Figure 21: Llama 3 8B attention weights; head 3; layers 0, 8, 16, 24, 31.

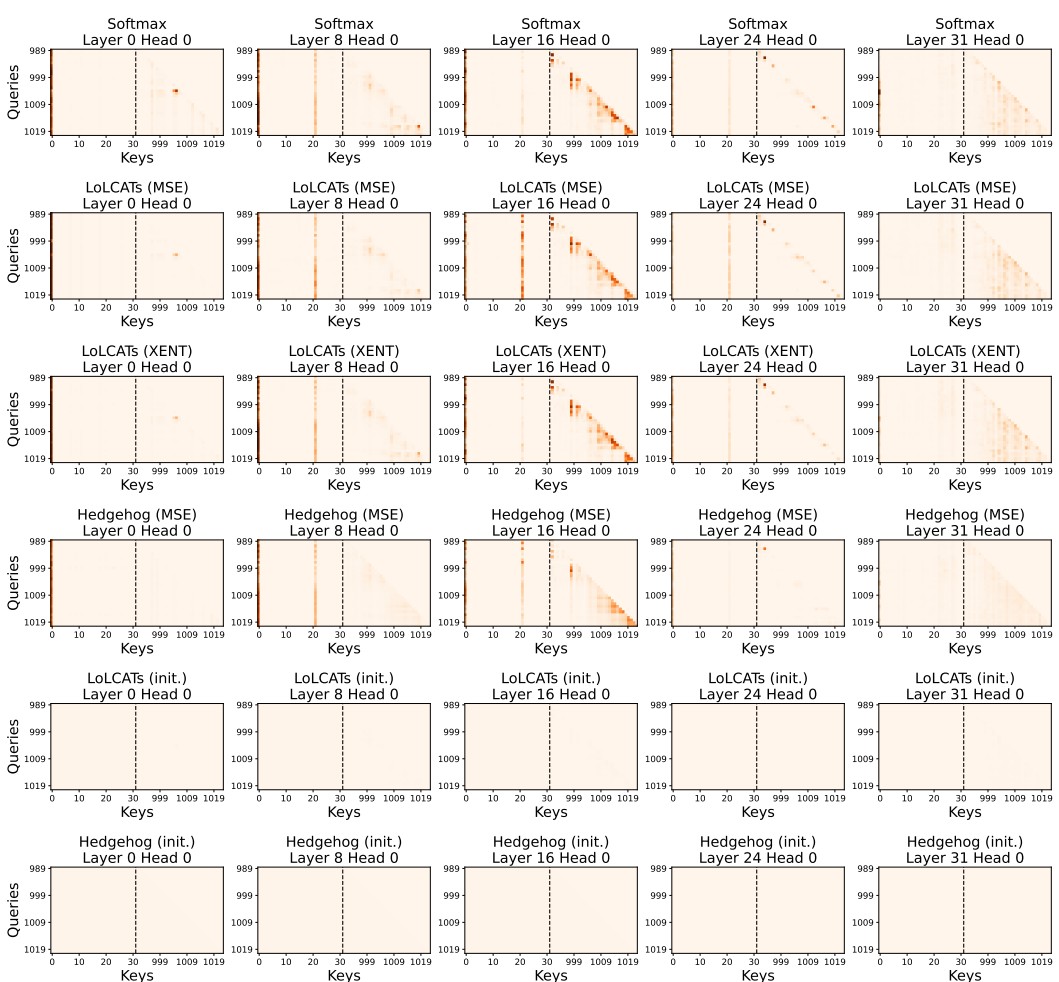

Figure 22: Mistral 7B v0.1 attention weights; head 0; layers 0, 8, 16, 24, 31.

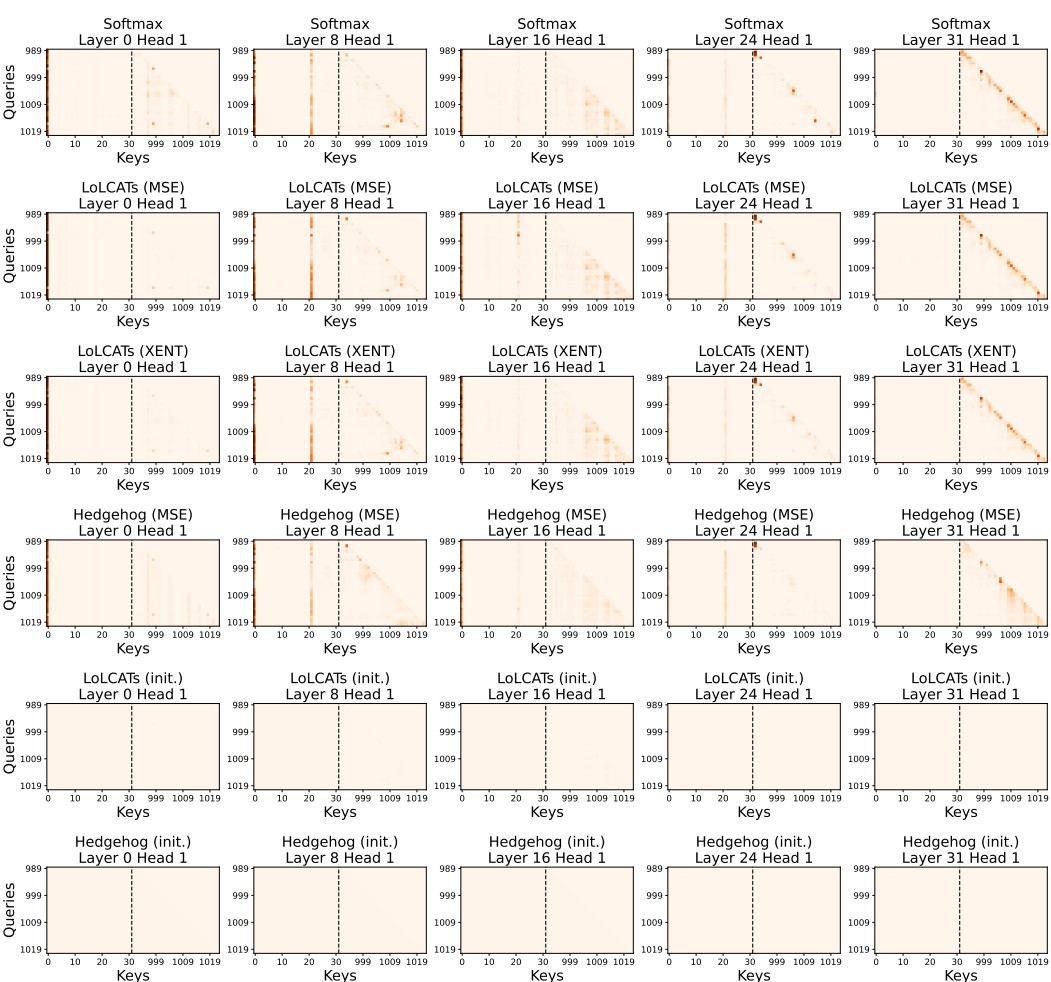

Figure 23: Mistral 7B v0.1 attention weights; head 1; layers 0, 8, 16, 24, 31.

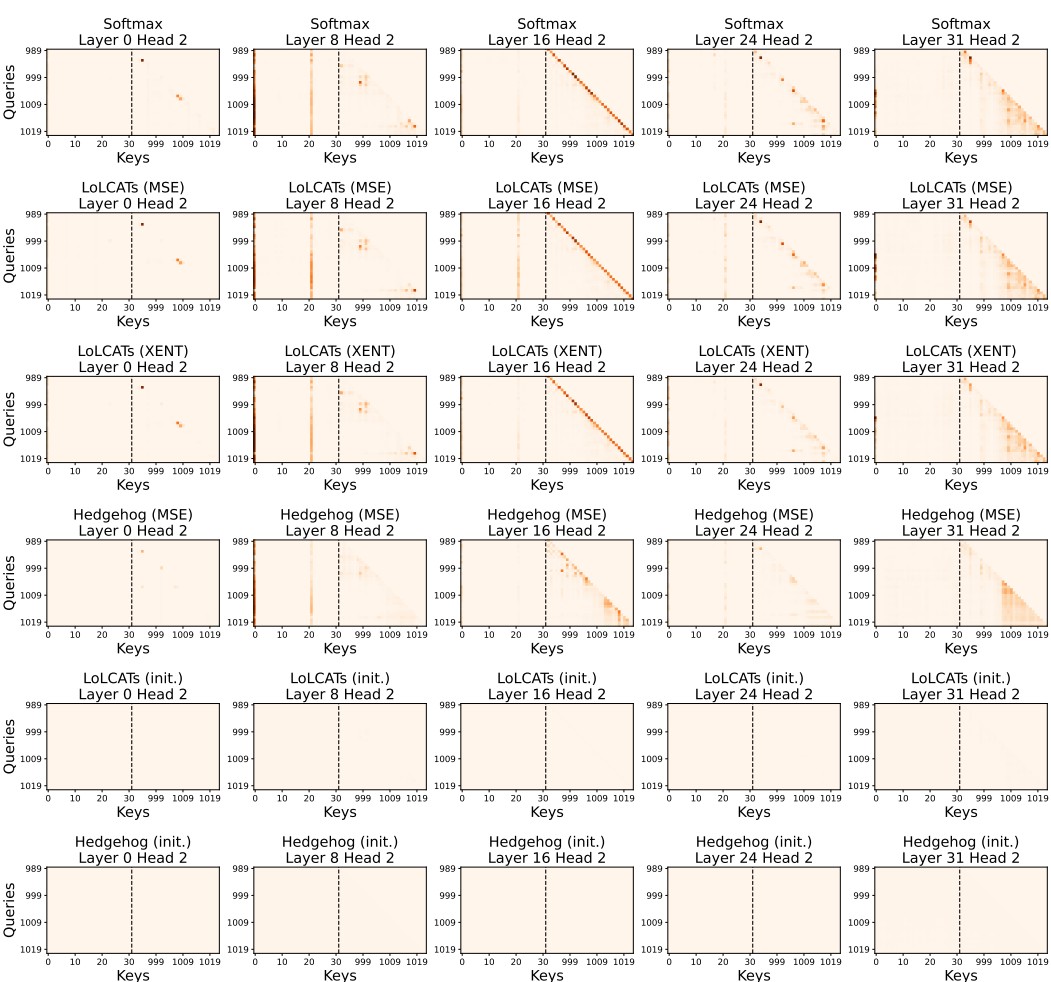

Figure 24: Mistral 7B v0.1 attention weights; head 2; layers 0, 8, 16, 24, 31.

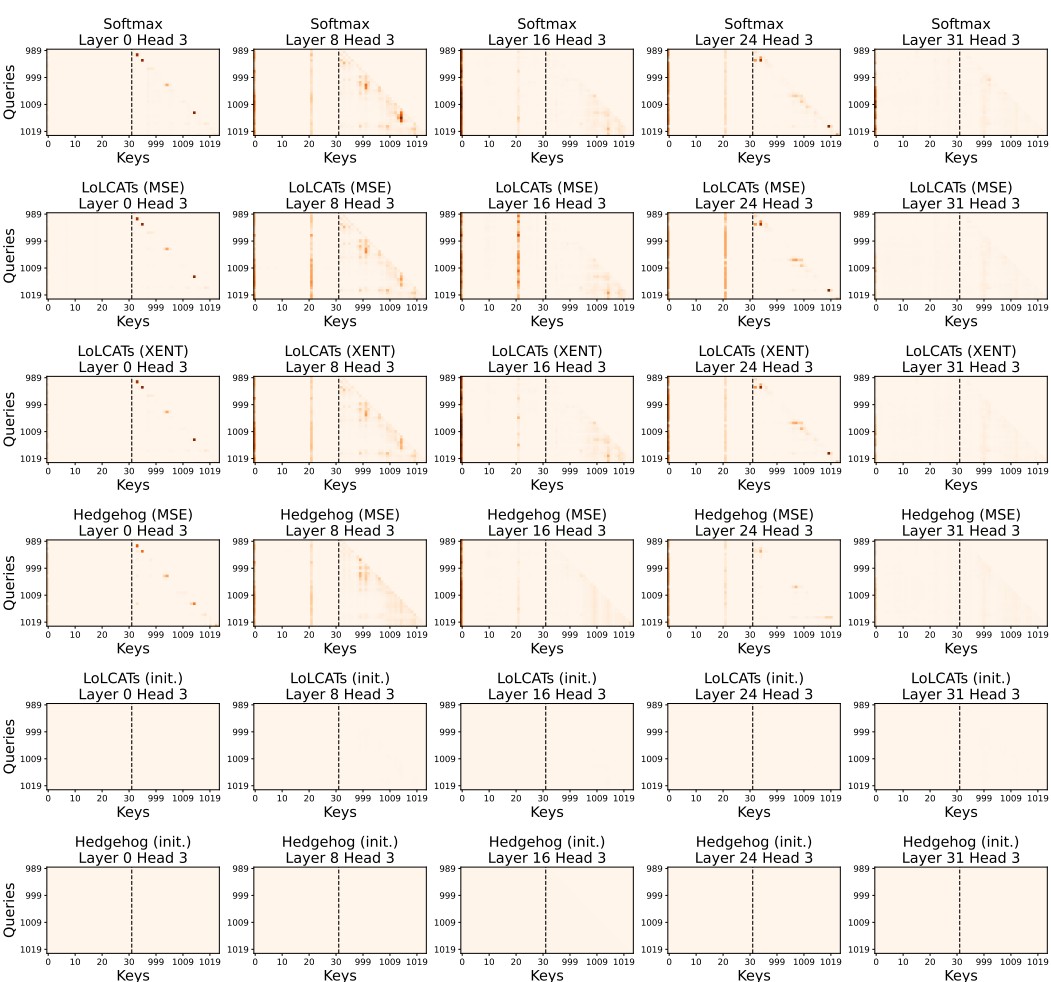

Figure 25: Mistral 7B v0.1 attention weights; head 3; layers 0, 8, 16, 24, 31.

