# OpenReview forum: "LoLCATs: On Low-Rank Linearizing of Large Language Models"
_ICLR.cc/2025/Conference — ICLR 2025 Poster_

### Official Review · Reviewer_povv · 2024-11-03

**Soundness:** 2
**Presentation:** 2
**Contribution:** 2
**Rating:** 5
**Confidence:** 4

**Summary:**

Authors proposed a new method to approximate the quadratic attention operation to semi-linear ones (with a pre-specified quadratic window). The linearized approximation seems to work with PEFT so the computational burden is partially alleviated when accelerating the computation on large models.

**Strengths:**

- The Introduction reads well (but unfortunately not the case for the rest)

- An interesting problem to study.

- Provide the code in the Appendix

**Weaknesses:**

- The way authour present their methods interleaves many previous works, and thus it's difficult to precisely pinpoint what's their contribution.

- Some lacking experiments make me uncertain if the claimed effectiveness is true.

- The above two factors combined make me uncertain what's the real efficacy of the method.

**Questions:**

1. Can authors point out which line of appended code correspond to a. Eq 7?

2. I'd like to confirm the author is doing post-training approximation or training from scratch with this specialized architecture. (It's a bit unclear to me). And for all the baseline methods, are they doing training from scratch or post-training approximation.

3. My understanding is that actually there is no new technical thing proposed. Eq 5 is proposed by previous works, and sliding window attention was also proposed in the previous work. And it seems like combining this 2 idea is the only technical contribution. Is that correct? Or do you have other new proposed approximation that I overlooked?

3. In that regard, adding a sliding window to any of the previous linearization techniques should improve the performance, and this has to be validated. If not, there is a need to investigate why synergy only happens here.

4. Apologize that I really don't get that's the purpose of Table 1, Figure 3 and Figure 5. What's the main point you want to convey here?

5. Is there a study of different window size?

6. For ablation study I only see Table 5. Is there one with your proposed method? Or should I read it as Hedgehog == Eq 5 and your method is nothing butHedgehog + Sliding?  And somehow Hedgehog + sliding ==> 68.78 outperformed your proposed method a lot, which reads weird to me.

7. Also don't know why ablation is only done on MMLU, I am interested in seeing RCV-e and PiQA too.

8. I don't really see what's the help from the linearization to parameter-efficient finetuning. I still don't understand why PEFT can't work on other methods. In addition, full FT should still outweigh PEFT. I believe authors should also show their method with full parameter tuning to see what's the difference.

---

> ### Author Response · Authors · 2024-11-21
> **Response to Reviewer povv (1/2)**
>
> Thank you for your constructive comments and review. We appreciate the clarifying questions, and believe updating the draft in response has improved our presentation (adding algorithm boxes and additional signposting text), allowed us to add additional experimental results and insights, and hopefully resolve any doubts on the claimed effectiveness + contributions.
>
> We are happy to follow up with any questions
>
> ---
> > **Q1: Which line of the appended code corresponds to Eq 7?**
>
> Sorry our initial submission only included the standard linear attention. In our revision, we updated the pseudocode to include Eq. 7’s implementation in Listing 5 (L1837) (now Eq. 6; we removed an earlier redundant line). We also substantially reworked this section (App C.1) to improve the presentation, walking thru each component in PyTorch-like code.
>
> ---
> > **Q2a: I'd like to confirm the author is doing post-training approximation or training from scratch with this specialized architecture**
>
> We are doing post-training approximation. We added Algorithm 1 in the revision to make this clearer (L397). Given an existing softmax attention, we only newly initialize the learnable feature maps $\phi_q$, $\phi_k$ and mixing term $\gamma$, and train these parameters to match the original softmax attention.
>
> > **Q2b: For all baseline methods, are they doing training from scratch or post-training approximation**
>
> We compare against both. Table 3 compares against linearizing or post-training methods. Table 4 compares against subquadratic LLMs trained from scratch.
>
> ---
> > **Q3: Clarifying technical contributions (Eq 5 is proposed by previous works, and sliding window attention was also proposed in the previous work. It seems like combining this 2 idea is the only technical contribution)**
>
> You’re correct that we build on simple and straightforward ideas easily adopted in prior work. However, **we clarify that our technical contribution lies in figuring out + understanding how to make these ideas work together**, i.e., to effectively linearize much larger LLMs (405B, 50x the size of prior) at unprecedentedly accessible training budgets (only using 0.2% of prior method training tokens, 0.2% of their trainable parameters). We note two points here on quality and efficiency:
> * **On quality**, we propose a new sliding window + linear attention layer, but we also propose a new way to linearize LLMs (explicitly with the goal to replicate softmax attentions) as a way to improve quality. Regarding technical contributions, **we contribute various empirical analyses**, studying
>   * Different linear attention feature maps (Hedgehog vs T2R)  (Sec. 3.2)
>   * The effect of different training stages (attention transfer (Eq. 5) vs just swapping attentions + LoRA finetuning),
>
>   * Ablations on how different combinations affect quality (reporting these in the updated revision, e.g., LoRA rank (App. B.3.1), LoRA projection (App. B.3.2), window size (App. B.3.3))
> * **On efficiency**, just computing this sliding window + linear attention in PyTorch is slow compared to kernel-optimized softmax attention implementations such as FlashAttention [1]. In the revision, we clarify that we provide hardware-aware implementations in ThunderKittens [2] to make our method efficient in practice (L359). We expand on details in App. C.2
>
> > **Q3a: Eq 5 is proposed by previous works**
>
> We also point out that while in general a layer-wise MSE loss (Eq. 5) or LoRA finetuning are not new, **we repurpose these components** in new ways to learn softmax-approximating linear attentions and recover language modeling quality in linearized LLMs
> * The most related prior work (Hedgehog) uses a cross-entropy loss over all n^2 attention weights (for n-long samples) to supervise attention approximation (Eq. 12) [3]. This requires O(n^2) attention for training, so [3] cannot use FlashAttention and limits linearizing to smaller n. Instead, we **show for the first time that supervising with just the outputs also works** (see plotted attention weights, Fig. 18-25). Notably, this **reduces training memory from O(n^2) to O(n)**, making LoLCATs much more accessible. By just computing attention outputs, we can use FlashAttention for softmax attentions and Eq. 6 for linear attentions both in O(n) memory.
> * Relatedly, all prior related linearizing works call for full-rank updates [3, 4, 5], making it unclear if LoRA suffices for linearizing. We show for the first time that LoRA works, but much better after first learning the attentions (Table 2, Fig. 3)
> ---
>
> **References**
> [1] FlashAttention: Fast and Memory-Efficient Exact Attention with IO-Awareness, Dao et al., 2024
> [2] ThunderKittens: Simple, Fast, and Adorable AI Kernels, Spector et al., 2024
> [3] The Hedgehog & the Porcupine: Expressive Linear Attentions with Softmax Mimicry, Zhang et al. 2024
> [4] Finetuning Pretrained Transformers into RNNs, Kasai et a., 2021
> [5]  Linearizing Large Language Models, Mercat et al., 2024

---

> > ### Author Response · Authors · 2024-11-21
> > **Response to Reviewer povv (2/2)**
> >
> > > **Q4: Adding a sliding window to any of the previous linearization techniques should improve the performance, and this has to be validated**
> >
> > We agree; in our main paper ablations (Table 5) we show how adding a sliding window impacts performance for both Hedgehog [3] and Transformer-to-RNN (T2R) [4] linear attentions (Table 5; see Table 10 in revision for per-task results). For both, **we validate that sliding windows improves performance across tasks** (Swap & Finetune vs +Sliding Window). We list these below (improvements in parentheses). Adding the first stage of training to approximate softmax attention (+Attention transfer) improves quality further. In general, we can apply these components to any feature map, where new feature maps may further improve linearized quality
> >
> > | Linear attention | Metric | Swap & Finetune | +Sliding window | +Sliding window, +Attention transfer |
> > |---|:---:|:---:|:---:|:---:|
> > | Hedgehog | Avg. LM Eval | 44.20  | 68.78 (+24.58) | 70.66 (+26.46) |
> > | Hedgehog | MMLU | 23.80 | 45.80 (+22.00) | 52.77 (+28.97) |
> > | T2R | Avg. LM Eval | 38.84 | 39.52 (+0.68) | 68.28 (+29.44) |
> > | T2R | MMLU | 23.20 | 23.80 (+0.60) | 40.70 (+17.50) |
> >
> > ---
> >
> > > **Q5: Purpose of Table 1, Fig. 3, Fig. 5**
> >
> > These present results that motivate our later contributions. As part of our method, we propose low-rank linearizing, and study how adapting available linear attentions to this setting perform (all prior works linearize with full parameter finetuning (500x our parameters), making it unclear if LoRA is feasible)
> >
> > The tables and figures thus show:
> > * **An initial contribution**: we show for the first time we can linearize with just LoRA. Furthermore, with attention transfer, we substantially reduce the training tokens needed to reach low PPL (L287, Fig. 3)
> > * **Motivation for why we need LoLCATs architecture**: there is still a quality gap with these models (see Fig. 4 in combination with Table 4)
> > * **Insights for how to improve linearizing**: Fig. 5 suggests we can improve linearized LLM quality by matching softmax attentions more closely (MSE vs PPL)
> >
> > ---
> >
> > > **Q6: Is there a study of different window size?**
> >
> > Yes. In our revision we added these results in App. B.3.3 (Table 17). We ablate window size in {4, 16, 64, 256} and measure linearized LLM quality on LM Eval tasks. We found size 64 best-in-quality.
> >
> > ---
> >
> > > **Q7: Clarification on main paper ablations**
> >
> > We clarify that in Table 5, we organize ablations by linear attention feature map (Hedgehog or T2R), and training steps (use sliding window, use attention transfer). This lets us study how each component individually impacts performance
> >
> > * The LoLCATs default is Hedgehog feature map, +sliding window, +attention transfer. (clarified, L494)
> > * Hedgehog itself refers to the feature map described in Table 1
> > * Hedgehog + Sliding uses this feature map in the Eq. 6 sliding window + linear attention layer (Eq. 7 in the original submission), but without attention transfer (just swapping attention + finetuning) (L512)
> >
> > > **Q7a: somehow Hedgehog + sliding ==> 68.78 outperformed your proposed method a lot**
> >
> > We think this may be a slight misreading. 68.78 is the average LM Eval score. Our method for this metric in the original submission gets 70.6 (corrected to 70.66 in the revision, sorry for the rounding typo)
> >
> > ---
> >
> > > **Q8: Why is ablation only done on MMLU? Interest in ARC-e and PiQA too**
> >
> > For space, we grouped results by MMLU and 0-shot LM Eval tasks, as subquadratic models perform noticeably worse vs Transformers on MMLU. In our revision, we report full results in Table 10 (App. B.1.1). We also add many more ablations with per-task results in App. B (e.g., LoRA rank, Table 15; LoRA projection, Table 16; Window size, Table 17)
> >
> > ---
> >
> > > **Q9: In addition, full FT should still outweigh PEFT. I believe authors should also show their method with full parameter tuning to see what's the difference.**
> >
> > Thanks for this suggestion. We added this in App. B.3.1, comparing full finetuning with different LoRA ranks (r = 4 to 256). We summarize results below (please see results per task in Table 15)
> >
> > | LoRA Rank | 4 | 8 | 16 | 32 | 64 | 128 | 256 | Full FT |
> > |---|:---:|:---:|:---:|:---:|:---:|:---:|:---:|:---:|
> > | MMLU | 50.1 | 52.8 | 48.9 | 51 | 51.7 | **53.4** | 52.1 | 52.1 |
> > | Avg LM Eval | **71.3** | 70.7 | 69.9 | 70.1 | 71.1 | 69.7 | 69.2 | 70.1 |
> >
> > Surprisingly, full finetuning does not lead to best performance, and smaller ranks (r=4, 8) are competitive.
> > * We leave further exploration for future work, but hypothesize that low-rank updates may maintain 0-shot quality by preventing overfitting. To linearize, we need to train over some data. With full parameter updates, we may overfit to this data, introducing potentially harmful updates to pretrained LLM weights and hurting generalization. LoRA caps these updates to low-rank updates and may thus reduce this risk [6]
> >
> > **References**
> > [6] LoRA Learns Less and Forgets Less, Biderman et al., 2024

---

> > ### Comment · Reviewer_povv · 2024-11-23
> > **Additional Questions**
> >
> > Sorry that I don't really have time during these few days and I just scanned through your replies and have rather ad-hoc additional questions.
> >
> > A. For Q2B, you mentioned "Table 3 compares against linearizing or post-training methods". Honestly, I don't know what's the official definition of linearizing method. So "compares against linearizing" is the part I don't get it. Is it training from scartch with linearizing architecture? I think my original question is try to get from scratch vs non from scratch (post training).
> >
> > B. I think the Eq5 I talked about is now Eq 4 in the latest version. So if the major contribution you metnioned here is the layer-wise to output only, I. believe the writing needs to be revised greatly. Following on the current reading, I don't really get that point and I can't really pinpoint what exactly that means. It reads like the difference between the latest Eq5 and Eq8 should be the key factor, but somehow I couldn't really grasp what's the major difference..... It reads to me still the current eq 6 is the major contribution and perhaps I need to read a bit the code (response in Q1) to get it clear. But I'd like to point out things should be written clearly in the main text but not appendix.  A reviewer is not required to read appendix.

---

> ### Author Response · Authors · 2024-11-24
> **Response to Additional Questions (1/2)**
>
> No worries and thanks for your time!
>
> We appreciate the opportunity to improve our paper's clarity. We have also uploaded a new version (further updates in green) to address your questions.
>
> ---
>
> > **A. For Q2B, you mentioned "Table 3 compares against linearizing or post-training methods". Honestly, I don't know what's the official definition of linearizing method. So "compares against linearizing" is the part I don't get it.**
>
> Apologies, we clarify that "linearizing" and "post-training" here are the same thing  (we meant the "or" to mean they are interchangeable). Linearizing means we take a pretrained LLM, swap or change the softmax attentions into linear attentions, and finetune the model (hence a form of "post-training")
>
> * So back to your original question:
> > **Q2b: For all baseline methods, are they doing training from scratch or post-training approximation**
>
> We compare against both baselines trained from scratch (Table 4) and those "post-trained" (Table 3).
> * We feel we've made this clear in all our drafts, defining "linearizing" in the first lines of our intro, e.g., L041-042 (emphasis added):
> > linearizing aims to *start with openly available LLMs*—e.g., those with
> 7B+ parameters pretrained on trillions of tokens (AI, 2024; Jiang et al., 2023)—and (i) swap their
> softmax attentions with subquadratic analogs, before (ii) *further finetuning* to recover quality.
> * This terminology also follows from prior literature e.g., [1]
> ---
>
> > **B. I think the Eq5 I talked about is now Eq 4 in the latest version. So if the major contribution you metnioned here is the layer-wise to output only, I. believe the writing needs to be revised greatly.**
>
> Sorry we confused the Eq 5 in our rebuttal. If your original concern on
>   * > **Q3a: Eq 5 is proposed by previous works**
>
> referred to our use of linear attention $y_n = \sum_{i=1}^n \frac{\phi_q(q_n)^T \phi_k(k_i)}{\sum_{i=1}^n \phi_q(q_n)^T \phi_k(k_i) } v_i$, then we re-emphasize that simply using Eq. 5 (now Eq. 4) is **not our technical contribution**.  We agree many linear attentions already exist.
>
> Rather, we respectfully clarify that our paper describes the following main contributions. We:
> 1. Propose a way to linearize LLMs with much greater training efficiency---by converting LLMs with softmax attention to those with linear attentions (L071 - 078) before LoRA finetuning---and better understand how *existing linear attentions* work in this low-rank linearizing setup (L080-081)
> 2. Figure out how to improve the quality of these linear attentions to get SoTA results, e.g., with Eq. 6 (L093-094)
> 3. Use these advances to scale linearizing up to unprecedentedly large model sizes (L097 - 101)
>
> These contributions are consistent with what's presented in the **main paper** intro (lines above), methods, and results:
>
> | Contribution | Methods Section | Results Section |
> |---|:---:|:---:|
> | 1. Propose efficient linearizing & study existing linear attentions | Sec. 3.1, 3.2 | Sec. 4.1 |
> | 2. Improve quality | Sec. 3.3.1 | Sec. 4.2 |
> | 3. Scale up linearizing to 70B, 405B LLMs | Sec. 3.3.2 | Sec. 4.3 |
> ---
>
> They are also acknowledged in the initial reviews (**Summary** and **Strengths**) of every other reviewer (NsVB, M1AP, Dj9p)
>
> ---
> > **So if the major contribution you mentioned here is the layer-wise to output only, I. believe the writing needs to be revised greatly.**
>
> Similarly, we do not view the layer-wise MSE loss you reference here as one of our 3 major contributions (stated above). Rather, we pointed this out in our response as just another technical contribution and advantage over the prior Hedgehog related work [2].
> * (We also clarify we *are* doing a layer-wise loss, but it's computed over *each layer's attention outputs* (Eq. 5) instead of each layer's *attention weights* like in [2], see the Hedgehog loss in Eq. 12)
> ---
>
> > **Following on the current reading, I don't really get that point and I can't really pinpoint what exactly that means.**
>
> For space we clarify the advantage in our next reply. This relates to improving linearizing training efficiency, and we added a comment on this in Sec 3.1 (**Training footprint and efficiency** paragraph, L234-239).
>
> ---
> > **I'd like to point out things should be written clearly in the main text but not appendix**
>
> Thanks to your comments we believe the current revision reflects this. The key methods, claimed contributions, and differences with prior work are presented in the main paper, with extra details referenced and deferred to the appendix.
>
> ---
> We are happy to follow-up with any additional questions. Given our responses to your earlier concerns, we would greatly appreciate if you could reconsider your score, in light of these clarifications and our paper's demonstrated advances in linearizing LLMs.
>
> ---
>
> **References**
>
> [1] Linearizing Large Language Models, Mercat et al., COLM 2024
> [2] The Hedgehog & the Porcupine: Expressive Linear Attentions with Softmax Mimicry, Zhang et al. 2024

---

> ### Author Response · Authors · 2024-11-24
> **Response to Additional Questions (2/2) (MSE loss advantage in LoLCATs)**
>
> > **Clarification on layer-wise MSE loss advantage in LoLCATs**
>
> This relates to improving linearizing training efficiency, and we added a comment on this in Sec 3.1 (**Training footprint and efficiency** paragraph, L234-239).
>
> * Recall that we train the linear attention to match softmax attention (L197-198).
>    * This requires computing both a "ground-truth" softmax attention and a "predicted" linear attention (Eq. 4) (using some linearizing data, e.g., text samples with $n$ tokens)
>
> ---
>
> * There are multiple ways we can train the linear attention to match the softmax attention. The prior work (Hedgehog) [1] calls for matching the attention weights ("qk dot products", e.g., plotted in Fig. 18 - 25), see [1] or Eq. 12 for the training loss.
>   * But this means we need to compute all $n^2$ weights (e.g., $a_{i, j}$ for query $i$ and key $j$, Eq 1) for both the softmax and linear attentions in each layer. **This makes the attention training procedure in Hedgehog quite memory expensive, scaling quadratically** when using sequences with large $n$ (e.g., why Transformers had limited context lengths before 2022 [2]).
>
> ---
>
> * Fortunately, if we only need to compute the attention outputs $y_i$ (as with the LoLCATs MSE loss over outputs), then we only need these $n$ outputs (**so we can compute the loss with linearly scaling memory**)
> * We now also have ways to compute both the softmax and linear attention outputs in $O(n)$ memory. Following these ways (described below), **LoLCATs then reduces the training memory from $O(n^2)$ to $O(n)$**
>   * For **softmax attention outputs**, we can use **FlashAttention** [2]. This fuses the attention operations (Eq 1) in a CUDA kernel so we can quickly compute the outputs in $O(n)$ memory (see *tiling* and *recomputation* of the attention weights in [2] for more details). Note that with this fusion, simple off-the-shelf implementations only return the outputs and not the intermediate attention weights [3] (which again is fine for LoLCATs, because we just need the outputs. But makes things more complicated to implement with Hedgehog [1], which needs the attention weights)
>   * Meanwhile, for **linear attention outputs**, we can simply use Eq. 2 or Eq. 6 to compute the outputs in $O(n)$ memory.
>
> ---
>
> So to recap, with the LoLCATs MSE loss, we only need to output the $n$ attention outputs for both softmax and linear attentions, instead of the prior $n^2$ attention weights in Hedgehog. This further lets us use modern softmax implementations like FlashAttention to keep training in $O(n)$ memory (when using samples of length $n$). As a result, we are **an order-of-complexity more efficient in memory** than the prior attention learning approach presented in past work [1].
> * Furthermore, despite these memory savings, we note that we can recover similar attention weights (see our visualizations in Figures 18-25)
>
> ---
>
> **References**
>
> [1] The Hedgehog & the Porcupine: Expressive Linear Attentions with Softmax Mimicry, Zhang et al. 2024
> [2] FlashAttention: Fast and Memory-Efficient Exact Attention with IO-Awareness, Dao et al. 2022
> [3] https://github.com/huggingface/transformers/issues/28903

---

### Official Review · Reviewer_Dj9p · 2024-11-04

**Soundness:** 3
**Presentation:** 3
**Contribution:** 3
**Rating:** 8
**Confidence:** 3

**Summary:**

The paper presents a novel method for linearizing large language models (LLMs) to make them more efficient in terms of memory and compute resources. The authors propose a method called Low-rank Linear Conversion via Attention Transfer (LOLCATS), which aims to replace the quadratic attention mechanisms in popular Transformer-based LLMs with subquadratic alternatives, such as linear attention. This approach avoids the expensive pretraining costs associated with traditional LLMs.

**Strengths:**

- The method is shown to scale to unprecedentedly large models (70B and 405B parameters), which was not possible with previous linearization techniques.
- The method is applied to various models, showing its broad applicability and potential impact on the field of natural language processing.

**Weaknesses:**

- There is a lack of an overall summary description of the LOLCATS method. An algorithm description or pseudo code can be added.
- There are some writing errors, such as Line 294: APP.???

**Questions:**

- Why was the proposed method not validated on a smaller model, such as llama1B?

---

> ### Author Response · Authors · 2024-11-21
> **Response to Reviewer ​​Dj9p**
>
> Thank you for your review! We appreciate the attention to detail and have fixed the writing errors and typos in our revision. We also used your comments to improve our manuscript, as described below.
>
> > **W1: Lack of overall summary**
>
> We updated the paper to include algorithm boxes to summarize LoLCATs (Algorithm 1, Algorithm 2; L397 - 409). We also restructured the code in the appendix to be easier to follow as pseudocode for the entire linearizing process (Appendix C.1).
>
> > **Q1: Why was the proposed method not validated on a smaller model, such as llama 1B?**
>
> We initially focused on larger LLMs with at least 7B parameters as this model size more strongly motivates LoLCATs. Among related works that propose new subquadratic architectures, several report results for pretraining 1 or 1.3B parameter models [1, 2, 3, 4, 5, 6]. However, fewer do so at the 7B scale, suggesting a need to develop more cost-effective ways to scale up new architectures for larger LLMs.
>
> However, we agree that linearizing 1B LLMs is also important to validate, especially if we can save training time by starting with available Transformers. We thus added experiments validating LoLCATs on two recent 1B models: Llama 3.2 1B, and Phi 1.5 1.3B, comparing against readily available alternatives (see Table 12, 13, 14 for full results). We find LoLCATs is able to achieve state-of-the-art linearized LLM quality in all evaluations.
>
> ---
> **Llama 3.2 1B Comparison**
> | Model | PiQA | ARC-e | ARC-c (acc. norm) | HellaSwag (acc. norm) | Winogrande | MMLU (5-shot) |
> |---|:---:|:---:|:---:|:---:|:---:|:---:|
> | Llama 3.2 1B | 74.4 | 65.5 | 35.8 | 63.7 | 60.5 | 31.9 |
> | -> T2R | 69.2 | 58.2 | 29.9 | 42.6 | 54.1 | 23.3 |
> | -> Hedgehog | 70.1 | 55.8 | 29.8 | 47.7 | 50.7 | 23.0 |
> | -> LoLCATs (Ours) | **74.6** | **63.0** | **35.1** | **63.7** | **61.5** | **27.3** |
>
> ---
> ---
>
> **Phi 1.5 1.3B Comparison**
> | Model | PiQA | ARC-e | ARC-c (acc. norm) | HellaSwag (acc. norm) | Winogrande | MMLU (5-shot) |
> |---|:---:|:---:|:---:|:---:|:---:|:---:|
> | **Transformer** |  |  |  |  |  |  |
> | Phi 1.5 1.3B (Our run) | 76.6 | 76.1 | 47.6 | 62.6 | 72.8 | 43.6 |
> | Phi 1.5 1.3B (from [7]) | 76.6 | 75.6 | 48.0 | 62.6 | 73.4 | - |
> | **Linearized** |  |  |  |  |  |  |
> | Phi-Mamba 1.5 [7] | 75.5 | 74.0 | 44.1 | 60.2 | 71.7 | - |
> | Hybrid Phi-Mamba 1.5 [7] | 76.5 | 75.3 | 45.8 | 60.6 | 72.0 | - |
> | Phi 1.5 1.3B T2R | 71.0 | 69.1 | 36.6 | 46.2 | 53.6 | 24.3 |
> | Phi 1.5 1.3B Hedgehog | 72.7 | 70.9 | 38.0 | 49.4 | 54.1 | 23.5 |
> | Phi 1.5 1B LoLCATs (Ours) | **76.9** | **77.0** | **46.9** | **62.3** | **72.7** | **39.2** |
>
> ---
>
> Similar to the 7B scale, we find LoLCATs gets state-of-the-art linearized LLM quality when compared against available linearizing alternatives (Table 12, 13), while also resulting in strong performance against 1.3B subquadratic LLMs pretrained from scratch (Table 14). This is all despite only using 40M training tokens, or 1.3% of the next best reported linearizing method for Phi 1.5 1.3B, and parameter efficient training (only updating feature maps in step 1, and using LoRA in step 2).
>
> ---
>
> **References**
>
> [1] xLSTM: Extended Long Short-Term Memory, Beck et al., 2024
> [2] Gated Linear Attention Transformers with Hardware-Efficient Training, Yang et al., 2023
> [3] Parallelizing Linear Transformers with the Delta Rule over Sequence Length, Yang et al., 2024
> [4] Simple linear attention language models balance the recall-throughput tradeoff, Arora et al., 2024
> [5] Mamba: Linear-Time Sequence Modeling with Selective State Spaces, Gu and Dao, 2024
> [6] Transformers are SSMs: Generalized Models and Efficient Algorithms Through Structured State Space Duality, Dao and Gu, 2024
> [7] Transformers to SSMs: Distilling Quadratic Knowledge to Subquadratic Models, Bick et al., 2024

---

> > ### Comment · Reviewer_Dj9p · 2024-11-22
> >
> > I thank the authors for their rebuttal. The explanation of Q1 makes sense. I have updated my scores from 6 to 8. And considering the addition of algorithm boxes, I have updated my Presentation scores from 2 to 3.

---

> > > ### Author Response · Authors · 2024-11-22
> > >
> > > Thanks again for your review and paper suggestions! We appreciate the score update.

---

### Official Review · Reviewer_M1ap · 2024-11-04

**Soundness:** 3
**Presentation:** 4
**Contribution:** 3
**Rating:** 6
**Confidence:** 4

**Summary:**

The paper addresses the efficiency and scalability challenges of large language models (LLMs) caused by the quadratic complexity of traditional Transformer models. To overcome these limitations, it introduces LOLCATS (Low-rank Linear Conversion with Attention TranSfer), a novel method that linearizes attention mechanisms to reduce computational and memory demands. LOLCATS uses an “attention transfer” phase to approximate softmax attention efficiently and a low-rank adaptation (LoRA) to correct errors. This approach enables scalable training of LLMs with up to 405B parameters—50 times larger than previous models—while maintaining high performance. Experimental results show LOLCATS outperforms existing methods and opens new avenues for scaling LLMs further.

**Strengths:**

1. The LOLCATS approach significantly reduces computational and memory costs through a two-step process involving "attention transfer" and low-rank adaptation (LoRA), effectively lowering the complexity of large models.

2. LOLCATS effectively retains the performance of the original self-attention model. Experimental results show that this method can recover much of the original model's quality after linearization, using only a small portion of the parameters and training data.

3. LOLCATS is the first to successfully apply linearization to large models with 70B and 405B parameters, expanding the applicability of linearization techniques.

**Weaknesses:**

1. The paper mentions the differences in errors across layers and their impact on model performance, but it does not sufficiently discuss the underlying reasons, such as why lower soft-attention entropy leads to higher errors.

2. There is insufficient explanation for new terms, such as "attention transfer," which may lead to misunderstandings regarding specific implementation details and processes. Clearly defining key concepts within the paper would improve overall clarity.

3. The paper does not provide averages, variances, or confidence intervals from multiple experiments, making it difficult for readers to assess whether performance differences in the model are statistically significant and robust.

4. As indicated in [1], there is a notable difference between Linear Attention and Softmax Attention in retrieval tasks, particularly in "needle-in-a-haystack" scenarios. Therefore, assessing the performance of models on such tasks before and after the application of LoLCAT would provide a more comprehensive validation of the method's effectiveness.

[1] Xuyang Shen, Dong Li, Ruitao Leng, Zhen Qin, Weigao Sun, & Yiran Zhong. (2024). Scaling Laws for Linear Complexity Language Models.

**Questions:**

1. I would like the author to supplement and polish the article based on the weaknesses.

2. The authors mention that LoRA effectively reduces approximation errors. Have they considered conducting a specific error propagation analysis? This is crucial for understanding how low-rank adaptation accumulates errors across different layers in deeper models, as it is vital for controlling cumulative errors.

3. In the LoRA method, the choice of the rank parameter is critical for the model's approximation effectiveness. How have the authors taken into account the impact of low-rank parameters on performance across different layers of the model?

4. Does this approach extend to other variants of Linear Attention, such as Mamba 1/2 and Hgrn 1/2?

---

> ### Author Response · Authors · 2024-11-21
> **Response to Reviewer ​​M1ap (1/2)**
>
> Thank you for your review and constructive comments! We have updated the paper following your feedback. Here we try to:
> * Better discuss the limitations and differences in MSE error that motivate our technical contributions (L300-308).
> * Define terms such as “attention transfer” (L199) (**W2**)
> * Report results over multiple seeds (means and standard deviations) (Table 11)
> * Expand our experiments with new results on needle-in-a-haystack retrieval tasks (App. B.4.2, L1296), layer-wise error and LoRA analysis (App. B.6.1, B.6.2), and LoRA rank + projection layer (App. B.3.1, B.3.2)
>
> Please see responses to your questions and comments on the above revisions below.
>
> ---
>
> > **W1: Adding discussion on why attention approximation errors impact linearized model performance**
>
> We updated the draft in several places to better discuss this. However, we acknowledge understanding the exact mechanisms is still an open question and interesting for further study.
> * First, we add that **prior works suggest low-entropy softmax attentions are difficult to approximate with standard linear attentions** [1] (L302 - 307), resulting in larger attention errors. We then show in Figure 5 the strong correspondence between attention error (high MSE) and poor final LLM quality (high perplexity)
>   * This motivates our choice to incorporate some sliding window softmax attention to better approximate the softmax attentions.
> * In our updated appendix, we added results to aid in our understanding of how these errors impact downstream linearized quality.
>   * In App. B.6.2, we **explore the connection between larger layer-wise MSEs and LoRA training dynamics** (Figure 16). We find LoRA updates with linear attentions poorly approximate softmax attention lead to noticeably different trajectories than those with softmax attention. This can lead to potential divergences in linearized model quality.
> * In Fig. 18-25, we find more evidence that pure linear attentions struggle to approximate low-entropy softmax attentions. We **visualize layer and head attention weights**, where prior Hedgehog linear attentions often fail to match the softmax attention weights in low-entropy samples. This results in larger output MSEs, and worse quality overall (again referencing Figure 5).
>
> ---
>
> > **W3: The paper does not provide averages, variances, or confidence intervals from multiple experiments**
>
> In our revision, we added Table 11 to include averages and standard deviations (SD) across 3 random seeds for our main LM Eval tasks, comparing LoLCATs with the prior Hedgehog linearizing method.
>
> |  | PiQA | ARC-e | ARC-c (norm) | HellaSwag (norm) | Wino- grande | MMLU (5-shot) | Avg. | Avg. (no MMLU) |
> |---|:---:|:---:|:---:|:---:|:---:|:---:|:---:|:---:|
> | Hedgehog | 76.86 (0.32) | 73.27 (0.67) | 40.76 (0.69) | 65.77 (0.38) | 53.42 (0.22) | 24.22 (0.62) | 55.72 (0.35) | 62.02 (0.35) |
> | LoLCATs | 80.79 (0.11) | 81.62 (0.41) | 54.73 (0.41) | 79.48 (0.07) | 72.92 (1.02) | 52.74 (0.64) | 70.38 (0.33) | 73.91 (0.29) |
>
> Out of convention, in our main tables we reported the results from related works on the LM Eval tasks, which all only include the absolute accuracies for these tasks [2, 3, 4]. However, we find the SDs to be quite low relative to the reported accuracies in general (c.f., Table 3, 4)
>
> ---
>
> > **W4: Assessing performance in "needle-in-a-haystack" scenarios**
>
> Thanks for this suggestion; we added these evals (App. B.4.2, Table 20, Figure 10). We use the passkey retrieval task [5, 6, 7] (see Listing 1, L1296 for an example) and Llama 3 8B. Given Llama 3 8B’s pretrained context length of 8192, we test whether LLMs can retrieve 5-digit passkeys hidden inside 8192-token-long texts.
>
> As a potential drawback of LoLCATs, if we simply use the model linearized with Alpaca data already, the model fails to retrieve the passkey correctly. However, by linearizing with passkey retrieval samples, we are able to recover softmax attention performance.
>
> | Placement | 0-10% | 10-20% | 20-30% | 30-40% | 40-50% | 50-60% | 60-70% | 70-80% | 80-90% | 90-100% |
> |---|:---:|:---:|:---:|:---:|:---:|:---:|:---:|:---:|:---:|:---:|
> | Llama 3 8B (Alpaca) | 100.00 | 100.00 | 100.00 | 100.00 | 100.00 | 100.00 | 100.00 | 100.00 | 100.00 | 100.00 |
> | LoLCATs Llama 3 8B (Alpaca) | 0.00 | 0.00 | 0.00 | 0.00 | 0.00 | 0.00 | 0.00 | 0.00 | 0.00 | 0.00 |
> | LoLCATs Llama 3 8B (Passkey) | 100.00 | 100.00 | 100.00 | 100.00 | 100.00 | 100.00 | 100.00 | 100.00 | 100.00 | 100.00 |
>
> In Figure 10, we show that when linearizing with retrieval data, the LoLCATs LLM retrieval is robust to various context lengths in a similar way to standard Transformers. We finally note that retrieving over 8192-long sequences is 4x our sliding window “receptive field” (32 layers * 64 window size = 2048), suggesting that we are not only relying on the softmax attention. Instead, we can learn subquadratic approximators that recover softmax-attention-like retrieval.

---

> ### Author Response · Authors · 2024-11-21
> **Response to Reviewer ​​M1ap (2/2)**
>
> > **Q2: The authors mention that LoRA effectively reduces approximation errors. Have they considered conducting a specific error propagation analysis?**
>
> We think this may be a slight misunderstanding, and are happy to clarify. In our main setup, we only do LoRA *after* we learn attentions, training the model with LoRA only on next-token prediction to recover language modeling quality (L219; Algorithm 2, lines 9-11).
> * This is because the attentions we learn layer-wise in stage 1 may be imperfect, and we need to update the original model parameters to adjust to these imperfect approximations (e.g., see generations in Appendix E.1).
> * This was simpler than further trying to match the original Transformer—e.g., via knowledge distillation on LLM outputs—while still obtaining state-of-the-art results (Table 3, 4).
>
>
> **Error propagation**. Furthermore, when we *are* learning to match attentions, we actually reduce error propagation by “teacher-forcing” (Fig 1 middle). We pass the true softmax attention outputs to the next layer, and thus prevent earlier approximation errors from propagating to the latter (we clarify this in the revision with Algorithm 1; also L1916 pseudocode, L232 discussion).
>
> **Layer-wise analysis**. However, we did track the layer-specific attention output MSEs in Figure 6b and 7 (right), where exactly as you point out, we found larger MSEs in the later layers after attention transfer. These are magnified with larger model size (comparing the 70B and 405B MSEs in Table 24 and 25). This motivated our block-wise training (Section 3.3.2).
>
> **Extra experiments**.  Finally, to potentially better address your question on the impact of LoRA for adjusting to layer-wise MSE differences, we ran additional experiments in App. B.6. Here we study:
> * How LoRA further reduces MSE per-layer when explicitly trained to match the original Transformer (App. B.6.1, see Figure 14a)
> * How LoRA layers with larger starting MSEs “cover more ground” and reduce MSE more than those with smaller MSEs (Figure 14b)
> * How LoRA training dynamics (in the form of cumulative weight updates) differ when LoRA finetuning a softmax attention Transformer, vs linearized models with different levels of attention approximation quality (App. B.6.2). Here we report these dynamics per LoRA projection (Figure 15) and layer (Figure 16, 17).
> ---
>
> > **Q3: On choice of LoRA rank parameter**
>
> We studied this as an ablation in App. B.3.1, where we **compare linearized LLM 0- and 5-shot performance over rank in {4, 8, 16, 32, 64, 128, 256}**. We surprisingly found that smaller rank 4 lead to best zero-shot performance (Table 15). We think this may be due to larger r allowing models to overfit to linearizing data--hurting pretrained quality--but need to study this further as an interesting question for future work.
>
> This **question also motivates our added ablation on LoRA projection target** (i.e., subset of Q, K, V, O proj) in App B.3.2. Here we interestingly see that LoRA on V and O proj (i.e., those not involved in the attention weight computation) often substantially improves LLM quality (Table 16) over LoRA subsets without either.
>
> ---
>
> > **Q4: Does this approach extend to other variants of Linear Attention, such as Mamba 1/2 and Hgrn 1/2?**
>
> We believe so! LoLCATs applies directly to any architecture that can be viewed as a linear attention, i.e., we can map the attention’s query, key, value projections to equivalents in the target architecture. We compare against concurrent works that explore this connection with linearizing Transformers into Mamba [3, 4] (Table 3, 13). Mamba-2 also discusses the architectural similarities to linear attention [7], which we may be able to exploit further.
>
> We are also happy to try LoLCATs on HGRN. **While we used simple linear attentions as a first step in this work, we are excited about how more modern and expressive architectures** (e.g., with state-expansion [8], delta updates [9]) could improve linearized performance further, and how LoLCATs can help scale up new architectures.
>
> ---
> **References**
> [1] The Hedgehog & the Porcupine: Expressive Linear Attentions with Softmax Mimicry, Zhang et al. 2024
> [2]  Linearizing Large Language Models, Mercat et al., 2024
> [3]  Transformers to SSMs: Distilling Quadratic Knowledge to Subquadratic Models, Bick et al., 2024
> [4]  The Mamba in the Llama: Distilling and Accelerating Hybrid Models, Wang et al., 2024
> [5]  Landmark Attention: Random-Access Infinite Context Length for Transformers, Mohtashami and Jaggi, 2023
> [6]  Extending Context Window of Large Language Models via Positional Interpolation, Chen et al., 2023
> [7] Transformers are SSMs: Generalized Models and Efficient Algorithms Through Structured State Space Duality, Dao and Gu, 2024.
> [8]  HGRN2: Gated Linear RNNs with State Expansion, Qin et al., 2024
> [9] Parallelizing Linear Transformers with the Delta Rule over Sequence Length, Yang et al., 2024

---

> > ### Author Response · Authors · 2024-11-26
> > **Checking in**
> >
> > Dear Reviewer M1ap,
> >
> > Thank you again for your time and reviewing our work. We especially appreciated the constructive feedback (eg on defining terms, better discussing our observations, including error bars, studying retrieval, and studying LoRA), and believe your suggestions have helped us supplement and polish the submission.
> >
> > As the last day to upload a revised PDF is coming up (11/27), we just wanted to check if you had any additional questions, and if you found our responses and revision helpful?
> >
> > Please let us know and thanks again for your review!

---

### Official Review · Reviewer_NsVB · 2024-11-04

**Soundness:** 3
**Presentation:** 4
**Contribution:** 3
**Rating:** 8
**Confidence:** 3

**Summary:**

This paper presents LoLCATs, a method for converting large language models (LLMs) with quadratic attention complexity into models with linear complexity while maintaining model quality. The key innovation is a two-step approach: (1) attention transfer - training linear attention layers to directly approximate the original softmax attention outputs, and (2) low-rank adaptation (LoRA) fine-tuning to adjust for approximation errors. The authors demonstrate that LoLCATs can effectively linearize models up to 405B parameters with significantly less compute and data compared to previous methods, while better preserving model capabilities.

**Strengths:**

- I believe this is a good work. The paper presents a novel method for linearizing LLMs that addresses key limitations of existing techniques. By focusing on approximating softmax attention outputs and using low-rank adjustments, the authors offer a fresh perspective on reducing computational complexity without sacrificing model quality.
- The LOLCATS method reduces the amount of training required, both in terms of model parameters updated and training data used. This efficiency makes the method practical for widespread use, especially in environments with limited computational resources.
- Demonstrating the ability to linearize large models up to 405B parameters is a notable achievement. The scalability of the method suggests it can be applied to future, even larger models.  The authors provide extensive experimental results, including comparisons with existing methods on multiple benchmarks.
- The paper delves into the reasons why previous linear attentions struggled to approximate softmax attention effectively. By identifying issues like attention entropy and layer-wise errors, the authors provide valuable insights that inform their improved architecture.

**Weaknesses:**

- The introduction and analysis of the preliminaries are too lengthy, with the core improvements in LOLCATs not appearing until the seventh page, which makes the reading experience somewhat disjointed.
- The experimental results are promising given the training budgets. However, I notice that even though previous works perform significantly worse for more challenging benchmarks (like MMLU in the setting), LOLCAT still considerably underperforms the original models. What will happen when the benchmarks become more challenging? (e.g., complex reasoning).
- While the authors claim improvements in inference efficiency, quantitative metrics such as actual speedup factors, memory utilization during inference, or comparisons of throughput are not extensively reported.

**Questions:**

- How were the hyperparameters for the attention transfer and low-rank adaptation chosen?

---

> ### Author Response · Authors · 2024-11-21
> **Response to Reviewer NsVB (1/1)**
>
> Thank you for your time and review! We appreciate your questions and comments, and hope to address them below.
>
> We also appreciate the feedback on paper presentation (**W1**). In our revision, we remove some lines on linear attention preliminaries, and clarify that our methods section:
> * First proposes the attention transfer + low-rank approach (page 4)
> * Then identifies issues with simply adapting existing linear attentions to this setting, which then
> * Finally motivates our final subsection on additional technical contributions to improve linearizing quality.
>
> If there are any additional suggestions, we would be happy to incorporate them.
>
> ---
>
> > #### **W2: Linearized performance is considerably worse on challenging benchmarks (MMLU), what will happen when benchmarks become more challenging?**
>
> We think we can still push linearizing quality further, and study this in the updated revision from the perspective of both linearizing data and architecture.
>
> First, on **data**, we found linearizing data choice can impact downstream task quality, where **linearizing with even a small amount of samples that match downstream task can help**.
>
> In our revised App. B.4.2, we study this for MMLU. Based on MMLU’s 5-shot multiple choice setup, we consider also linearizing with 10k samples of another multiple-choice dataset (CommonsenseQA (CQA) [1]). In combination with the 50k Alpaca samples, this results in a ~2 point boost (Table 19, L1294, also below). While a modest gain, for higher-level reasoning tasks, it can be helpful to linearize with a combination of pretraining and reasoning samples (e.g., chain-of-thought traces).
>
> | Alpaca | Alpaca + CQA | CQA only | Llama 3 8B |
> |:---:|:---:|:---:|:---:|
> | 52.8 | **54.5** | 43.9 | 66.6 |
>
> In addition, we can also **improve the linearizing architecture to better match softmax attention**. In Table 5, we saw significant improvement by adding small sliding windows of softmax attention (23.8 vs 52.8).
>
> In the same direction, one simple approach is to **keep some entire layers as softmax attention**. This trades efficiency for quality, but may be necessary for more complex tasks. As a preliminary result, when we kept the first half of layers as softmax attention and only linearized last half for Llama 3 8B, with just attention transfer over Alpaca (see Table 7 for full details) we were able to **substantially close the 5-shot MMLU gap (65.8% vs 66.6%)**.
>
> | Softmax Attn. Layers | PiQA | ARC-E | ARC-C | HellaSwag | WinoGrande | MMLU |
> |---|:---:|:---:|:---:|:---:|:---:|:---:|
> | All (Llama 3 8B baseline) | 79.9  | 80.1 | 53.3 | 79.1 | 73.1 | 66.6 |
> | 0-15 (LoLCATS 50%, Just Attn Transfer) | 79.5 | 80.2 | 53.4 | **79.2** | 73.6 | **65.8** |
> | None (LoLCATs, Attn Transfer + LoRA) | **80.9** | **81.7** | **54.9** | 79.0 | **74.1** | 52.8 |
>
> ---
>
> > #### **W3: Inference efficiency quantitative metrics**
>
> We reported this in Section 4.2 (“Subquadratic Generation Throughput and Memory”, Figure 8), but are happy to conduct any further requested benchmarking.
>
> ---
>
> > #### **Q1: How were hyperparameters chosen?**
>
> These were done through a hyperparameter sweep based on validation metrics (MSE during attention transfer, perplexity during LoRA adjusting). We clarify this in our revision on L842-847, and add additional experimental details in Appendix A.
> * For learning rates, we did an initial sweep over {1e-2, 1e-3, 1e-4}, checkpointing with early stopping.
> * We did not tune batch size or choice of optimizer, and used default values informed by prior work for other design parameters such as sliding window size [4], LoRA rank, and LoRA projection layers [5].
> * In our revision, we explored different LoRA ranks, LoRA projection layers, and window sizes as ablations (Appendix B.3.1, B.3.2, B.3.3)
>
> ---
>
> **References**
> [1] CommonsenseQA: A Question Answering Challenge Targeting Commonsense Knowledge, Talmor et al., 2019
>
> [2] Gated Linear Attention Transformers with Hardware-Efficient Training, Yang et al., 2023
>
> [3] HGRN2: Gated Linear RNNs with State Expansion, Qin et al., 2024
>
> [4] Simple Linear Attention Language Models Balance the Recall-Throughput Tradeoff, Arora et al., 2024
>
> [5] LoRA: Low-rank Adaptation of Large Language Models, Hu et al., 2021

---

> > ### Comment · Reviewer_NsVB · 2024-11-26
> >
> > Thanks, I will keep the rating.

---

### Author Response · Authors · 2024-11-21
**General Response (1/1)**

Many thanks to all reviewers for their helpful feedback and thoughtful comments, and to the ACs for chairing. Below we recap our paper + reviewer comments, and go over updates in our revision.

---
### **Recap**
To recap, towards obtaining LLMs with subquadratic efficiency, we study how to convert or “linearize” modern Transformer-based LLMs into linear attention variants with state-of-the-art quality, training and model parameter efficiency, and scalability.

As highlights, our method (called LoLCATs):
* Gets state-of-the-art quality on popular LM Eval tasks, outperforming prior linearizing methods and subquadratic LLMs trained from scratch
* Uses only 40M training tokens (prior linearizing methods use 2500x the tokens [1]), while only training 0.2% of LLM parameter counts via LoRA
* Scales up linearizing to 70B and 405B LLMs for the first time

We appreciate that reviewers consistently acknowledged our method's **effectiveness and potential impact**, noting that our work:
* Offers a “**fresh perspective on reducing computational complexity without sacrificing model quality**”, is “practical for widespread use, especially in environments with limited computational resources”, and provides “valuable insights that inform their improved architecture” (NsVB)
* “**Opens new avenues for scaling LLMs further**”, “expanding the applicability of linearization techniques” (M1ap)

* “Scales to unprecedentedly large models (70B and 405B parameters), which was not possible with previous linearization techniques,” and “is applied to various models, showing its broad applicability and **potential impact on the field of natural language processing**” (Dj9p)
* Provides an “interesting problem to study” (povv)

---
### **Revision**
Thanks to reviewer feedback, we uploaded a revision with updates highlighted in blue. We group these updates into two themes by reviewers:

**Writing clarity + presentation**
* We update the methods section to better present our overall method & first contribution (NsVB, povv):
  * We study + show for the first time that we can use LoRA to convert Transformer LLMs into viable linear attention variants. All prior methods use full model training, making it unclear before our work if our proposed LoRA linearizing is feasible [1, 2, 3] (povv)
* We add discussion on prior linear attentions' limitations in this low-rank setup (L294-308) (M1ap)
* We clarify how these results motivate our additional technical contributions to improve quality (Section 3.3) (povv)
* We explicitly define terms like “attention transfer” (L199) (M1ap)
* We summarize with algorithms 1 and 2 (L397 - 409) (DJjp), and update pseudocode to walk thru all components (App. C.1) (DJjp, povv)

**Expanded experimental analysis**
* *Additional study + evaluation*. We:
  * Find LoLCATs also gets **state-of-the-art quality when linearizing 1B LLMs** (Llama 3.2 1B, Table 12; Phi 1.5 1.3B, Table 13), outperforming other linearizing methods by 0.2-1.7 points, while only training 0.22% of their parameters (Dj9p)
  * Provide standard deviations and means over three seeds (Table 11) (M1ap)
  * Study how to improve quality on challenging tasks like MMLU* (App. B.42, L1275) (NsVB)

  * Include needle-in-a-haystack / passkey retrieval evals: we recover softmax attention-like recall by linearizing with retrieval data (App. B.42, L1323) (M1ap)
  * Conduct additional layer-wise analysis, tracking MSE error (App. B.6.1) and LoRA weight training dynamics (A and B low-rank matrices, App. B.6.2) (M1ap)

* *Expanded ablations*. We add ablations on:
  * LoRA rank (including full finetuning) (Table 15) (M1ap, povv)
  * Linearizing attention sliding window size (Table 17) (povv)
  * Attention transfer, sliding window, and feature map results on all zero-shot LM Eval tasks (Table 10) (povv)
We supplement these reviewer-requested ablations with more experiments on LoRA projection layer (Table 16), linearizing data (App. B.4.1), and training token budgets (App. B.5)

*In late-breaking results, by leaving some layers as softmax attention (50% like in prior work [3]), we substantially close the MMLU gap for Llama 3 8B:

|  | Mamba2-Llama (50% softmax attn) [3]   | LoLCATs (0% softmax attn)   | LoLCATs (50% softmax attn) | Llama 3 8B |
|---|:---:|:---:|:---:|:---:|
| 5-shot MMLU % | 55.7 | 52.8 | 65.8 | 66.6 |
---

We thank all reviewers again for their constructive comments, which we believe have strengthened the paper’s presentation. They also led to many more experiments and findings to improve overall content.

Please find our responses for individual reviewer comments below. We are happy to follow up with any questions.

**References**
[1]  Linearizing Large Language Models, Mercat et al., 2024
[2]  Transformers to SSMs: Distilling Quadratic Knowledge to Subquadratic Models, Bick and Li et al., 2024
[3]  The Mamba in the Llama: Distilling and Accelerating Hybrid Models, Wang and Paliotta et al., 2024

---

### Meta-Review · Area_Chair_q7R3 · 2024-12-22

**Metareview:**

## Summary

The paper introduces LoLCATs, a method proposed for improving the efficiency and scalability of large language models (LLMs) by replacing quadratic softmax attention with subquadratic linear attention. LoLCATs use a two-step process: attention transfer, where linear attention approximates softmax attention with minimal error, and low-rank adaptation (LoRA) to refine the model’s quality. This method significantly reduces computational requirements while maintaining competitive performance. LoLCATs enable training larger models, including the first linearized 70B and 405B parameter LLMs, with reduced memory and token costs. Experiments show notable quality improvements over previous approaches, narrowing the performance gap between linearized and original LLMs.

## Decision

The proposed idea is novel and relevant to improving the efficiency of frontier models. The method is shown to scale to large llama3 models (70B and 405B parameters), which was not possible with the previous linearization techniques. The method is applied to various models, showing its broad applicability and potential impact on natural language processing. The results are convincing, and I believe that publishing this paper would be beneficial for the LLM community.

**Additional Comments On Reviewer Discussion:**

The reviewers have raised some concerns and provided some feedback on the paper. Overall, I think the authors did a good job addressing most of them and clarifying some confusion caused by the writing. The authors have run more evaluations and ablations, which were requested by the reviewers as a result of the rebuttal. At the end of the rebuttal, I think the point of this paper is much clearer right now. I recommend the authors to incorporate all the suggested changes in the camera-ready version of the paper.

---

### Decision · Program_Chairs · 2025-01-22

Accept (Poster)